# Large-scale experimental evidence of carbon-mediated N and P co-amplification in proglacial soils

Hongyang Sun [1] ✉, Dong Yu [2] ✉, Jun Zhou [3] & Yang Chen[3]

Understanding N-P interactions is crucial for co-limitation in ecosystems but remains unclear due to oversimplified frameworks. Here, a large-scale N and P addition experiment in early soils revealed four key processes driving nutrient coupling, emphasizing carbon's central role. N addition significantly increases P availability. P availability drives biological N fixation. Both nutrients stimulate carbon fixation, with N having a stronger effect. N addition also intensifies ammonia nitrification, further reducing pH; P has no significant effect on nitrification. Based on these findings and prior research, we establish two models: one depicting P-driven N availability, the other N-driven P availability. These models highlight carbon-related processes as central to N-P coupling, enhancing nutrient availability in early soils. The results reveal a synergistic pathway where nutrient enrichment boosts co-availability rather than intensifying limitation. This framework clarifies the driving mechanisms of N-P bioavailability interactions that support positive ecosystem succession during soil development.

The interaction between nitrogen (N) and phosphorus (P) cycling processes has long been a focal point in the field of biogeochemistry. It is universally acknowledged that global terrestrial ecosystems are constrained by both N and P availability[1–7]. Empirical evidence demonstrates that across various ecosystems, plant growth exhibits comparable increments following the addition of either N or P, which suggests the prevalence of N and P co-limitation[5,8]. Moreover, given the fact that organisms require elements in stoichiometric proportions and are expected to take up and recycle nutrients in ratios that maintain balanced nutrition[9,10], some scholars[2] considered that such co-limitation profoundly suggests an intrinsic coupling of N and P through soil biological processes, highlighting the interconnected dynamics of these nutrient cycles within terrestrial environments. To elucidate the interaction mechanisms between them, extensive research efforts[11–14] have focused on two critical processes: P mobilization (encompassing phosphate solubilization and organic P mineralization) and N fixation. These investigations aim to clarify the regulatory effects of N and P availability on key parameters (such as N

fixation activity[12,13], phosphatase activity[14], and mineralization rates[11]) within these processes, thereby advancing our understanding of the coupling mechanisms underlying N and P cycling processes. However, due to the complex intrinsic properties of these processes that hinder the incorporation of other relevant processes in such studies, and the lack of ideal study areas to simplify these complexities, no research has fully elucidated the complete mechanism of the interconnection or coupling between N and P availability. This has led to difficulties in reaching unified conclusions among existing studies (see discussion below). To elucidate the coupling relationship and its underlying mechanisms between N and P cycles, previous research endeavors have predominantly adopted two analytical perspectives to gain deeper insight.

One analytical perspective focuses on the impact of N on indicators related to P processes, usually through N amendment experiments. Organic phosphorus (Po) mineralization and inorganic phosphorus (Pi) dissolution are two critical P processes for soil P mobilization and activation, with their relative predominance

[1]Sichuan Zoige Alpine Wetland Ecosystem National Observation and Research Station, Southwest Minzu University, Chengdu, China. [2]Sichuan Provincial Grassland Technology Research and Extension Center, Chengdu, China. [3]Institute of Mountain Hazards and Environment, Chinese Academy of Sciences, Chengdu, China. ✉e-mail: sunhongyang2009@sina.cn; yu_dong2013@sina.cn

influenced by the conditions of the soil matrix. For instance, in mature soils during later stages of soil formation, Po (Po mineralization) dominates within the organic layer matrix, whereas, in early-stage soils, primary mineral phosphorus (Pi dissolution) is more prevalent[15,16]. The mechanisms through which these two P processes mobilize P are entirely different. Therefore, adding N to different soils is likely to influence P availability through distinct pathways. Regarding the process of Po mineralization, N should logically be a significant positive factor for this process, as it is fundamentally linked to the process through one basic pathway: N is a key component of phosphatase (a typical protein, with an N content of 8%-32%)[17], which is the core component driving Po mineralization. However, over the past decade, meta-analyses encompassing dozens of global N addition experiments have reached varying conclusions: Marklein and Houlton[2] found that N addition substantially increased soil phosphatase activity (by 46%), while Gou et al.[18] concluded that N addition slightly enhanced acid phosphatase activity but inhibited alkaline phosphatase activity. By contrast, Chen et al.[19] reported that N addition had no significant effect on rhizosphere phosphatases but inhibited non-rhizosphere phosphatase activity. These discrepancies highlight a shared limitation among these studies: reliance on single-process indicators (e.g., sole focus on N fertilization rates or phosphatase activity) without considering other related processes. For example, carbon (C) processes external to the N and P cycles might be essential for explaining nitrogen's mechanism in Po mineralization, given research indicating Po mineralization could be driven by microbial demand for C[20,21]. Additionally, the acidification caused by the nitrification process[22] can lead to the Pi solubilization, which potentially inhibits phosphatase activity. However, the impact of the nitrification process on the mineralization of organic phosphorus is rarely evaluated.

In contrast to the well-studied effects of N addition on Po mineralization, there seems to be a lack of attention towards its effects on the phosphate dissolution process. Insights can be gleaned from related studies, such as our team's laboratory results suggesting that nitrogen's acidification effect may promote phosphate dissolution and that the production of organic acids involved in dissolving phosphate is closely linked to C processes[23]. Additionally, findings from pot experiments[24,25] on the role of C allocation in plant strategies for phosphorus acquisition (i.e., root exudation of organic acids and forming mycorrhizal symbioses) underscore the importance of C supply in the phosphate dissolution process. Thus, elucidating the mechanisms of nitrogen's effect on both Po mineralization and phosphate dissolution likely requires consideration of other related processes. Integrating these related processes into an N–P coupled framework is crucial for interpreting environment-dependent responses and resolving discrepancies in conclusions, a concept overlooked in previous nutrient addition experiments.

The other analytical perspective examines the influence of P on indicators related to N processes, typically through P addition experiments. Many studies have shown that exogenous P inputs can enhance diazotrophic activity and the N fixation rate[26–28], especially in P-deficient ecosystems[29,30], indicating the regulatory role of P in soil N availability and its driving role in the coupling of N and P. It is generally accepted that a direct mechanism underlying these enhancing effects involves the role of P in N fixation metabolism and nitrogenase synthesis[31,32]. However, this mechanism does not account for the study demonstrating that adding P to 24 forest soils failed to stimulate N fixation and, instead, resulted in an average 31% decrease in N fixation[33]. Moreover, recent meta-analyses[12] have concluded that P addition weakens N fixation overall, with a particular emphasis on inhibiting free-living N fixation and promoting symbiotic N fixation. Considering that symbiotic N fixation has a stable energy source (i.e., plants as hosts), whereas free-living N fixation lacks one[34], and given that N fixation reactions are highly energy-consuming[35], these findings imply that the responses of N fixation to

P addition are closely related to the supply of energy substances. Soil organic carbon (SOC), as a representative of soil energy substances, has been found to be a good predictor of biological N fixation in a comprehensive meta-analysis of field measurements[36]. A recent meta-analysis[37] focusing on the energy sources of N fixation also showed that negative responses of terrestrial N fixation to N addition weaken across increased SOC levels, and that SOC was the most important predictor regarding the responses of N fixation to N addition. These findings imply that C processes may also play a role in the impact of P addition on N fixation. Apparently, the above discussion indicates that existing studies have neglected to integrate C processes into the N-P coupling framework, which is likely a significant reason for the current lack of consensus in conclusions.

The aforementioned studies also indicate that while extensive research has been conducted on the N–P interactions in mature ecosystems, these studies often encounter methodological limitations due to intricate plant-soil feedback loops. Such ecological complexity obscures the fundamental mechanisms driving biogeochemical coupling between these essential nutrients. The uniqueness of primary succession environments—particularly newly exposed substrates following glacial retreat—provides an ideal experimental alternative. These pioneer systems exhibit simplified soil biogeochemistry due to no vegetation establishment, effectively decoupling nutrient dynamics from plant-induced disturbances. This environmental simplification allows clearer isolation of core N–P interaction mechanisms compared to mature ecosystems.

Traditional theory[2] posits that organisms are expected to allocate their resource reserves toward strategies that increase the acquisition of the most limiting resources—thus moving toward a state where all resources simultaneously limit productivity and growth. The observed N–P co-limitation patterns in developed ecosystems appear consistent with this framework. However, this paradigm fails to explain the successional trajectory of nascent soil systems: despite an initial severe lack of available nutrients, these systems progressively evolve into productive forests[15,38]. This evolution suggests the presence of positive feedback mechanisms, indicating a directional development that is incompatible with static co-limitation models.

Here, we have identified early soils in the most recently deglaciated area of the Hailuogou Glacier and established large-scale N and P addition experiments on these soils. Our aim is to integrate additional soil processes, such as the C process and nitrification process, into the theoretical framework of N–P coupling, beyond the processes of N fixation and P dissolution. Based on this approach, we seek to explore and reveal the positive feedback mechanisms among nutrient elements during the early stages of soil formation, providing empirical insights for the development of a unified theory of nitrogen–phosphorus coupling.

## Results

### Natural state of early soil in the most recently deglaciated area of the Hailuogou Glacier

In natural conditions, the early-stage soil is alkaline, with relatively consistent pH values at different sampling locations, ranging from 7.97 to 8.80, and an average pH of 8.43 (Table 1). As expected, total organic C (TOC) and total N (TN) in the early soils are very low (2.11 mg/kg for TOC mean, 0.09 mg/kg for TN mean), and both TOC and TN exhibit high heterogeneity, with the maximum TOC value being approximately 5 times the minimum value, and the maximum TN value being approximately 3 times the minimum value. Despite the total P (TP) being abundant (1.38 g/kg for TP mean) and exhibiting low heterogeneity (max/min < 1.5), the available P (AP) concentrations (2.3 mg/kg for AP mean) are very low and show high heterogeneity (max/min > 7.5). Based on the soil particle size distribution analysis, the early soil is more like rock debris (Table 1, Coarse grains > 40%). Moreover, the soil bulk density is very high, with a mean value of 1.82 g/cm³.

Regression analysis indicated there is very strong statistical evidence (i.e., $p < 0.001$) to support a linear relationship between available N and AP (Fig. 1A, B). However, linear relationships with weak statistical evidence ($p > 0.05$) were observed between TP and AP (Fig. 1C), TN and available N (Fig. 1D, E), or TN and TP (Fig. 1F).

## Effects of N and P addition on soil processes

The results of this study are based on experimental treatments with large sample sizes (90 replicates per treatment), thereby ensuring good statistical precision and robustness. The following sections will present the response of early-stage soils to N and P additions through

**Table 1 | Early-stage soil properties in the deglaciated area of the Hailuogou Glacier under natural conditions**

| Indices for the intact soil | Value |
| --- | --- |
| Soil pH | 8.43 ± 0.17 |
| TOC (g/kg) | 2.11 ± 0.60 |
| TN (g/kg) | 0.09 ± 0.02 |
| TP (g/kg) | 1.38 ± 0.08 |
| Fe (g/kg) | 30.87 ± 3.09 |
| Al (g/kg) | 67.67 ± 3.27 |
| Soil moisture (%) | 9.05 ± 0.16 |
| Soil temperature (°C) | 11.8 ± 1.4 |
| Bulk density (g/cm³) | 1.82 ± 0.12 |
| Coarse grains (2–0.2 mm, %) | 42.53 ± 3.02 |
| Fine grains (0.2–0.02 mm, %) | 45.04 ± 1.87 |
| Silt + clay (<0.02 mm, %) | 12.42 ± 3.83 |

The sample size for the determination of soil pH, total organic carbon (TOC), total nitrogen (TN) and total phosphorus (TP) is all 90, while the sample size for the determination of the remaining indicators is 10.

four soil processes: P solubilization, C fixation, ammonia nitrification, and N fixation.

In the early-stage soil, the P solubilization process was stimulated by N addition, yet showed no significant response to P addition (Fig. 2A). For parameters characterizing the P solubilization process, N addition treatments negatively affected pH (causing a significant decrease), but positively affected AP concentrations, low molecular weight organic acids levels (LMWOA), and the abundance of acid secretion genes (*gcd*), all of which showed significant increases. Although the results showed an increase in AP under P addition treatments, this did not mean that P addition promoted the P solubilization process; clearly, this was a result of the directly added AP in those treatments. This also indicated that this added AP was not entirely removed by rainfall and leaching, and that the P addition treatment indeed achieved an increase in the soil AP.

For the C fixation process in the early-stage soil, the dissolved organic C (DOC), TOC, *cbbL* gene and mRNA-*cbbL* gene related to C fixation were measured (Fig. 2B). Compared to the control treatment, both N and P additions increased the concentration of DOC, TOC and the copy number of the *cbbL* gene and mRNA-*cbbL* gene. Notably, N addition had a greater enhancing effect on these parameters than P addition.

For N fixation processes, our study does not merely rely on the abundance data of the nitrogenase gene (*nifH*), but also makes use of ammonium nitrogen ($NH_4^+$-N) and nitrogenase activity to more accurately reflect the characteristics of the actual nitrogen fixation process (Fig. 2C). Compared to the control check (CK) treatment, P addition significantly increased $NH_4^+$-N concentration, nitrogenase activity, and the abundance of *nifH* gene in early-stage soil (Fig. 2C), indicating that P availability drove the N fixation process during this stage. In contrast, N addition did not have a significant effect on nitrogenase activity or the abundance of the *nifH* gene, except for the increase in $NH_4^+$-N concentration. The increase in $NH_4^+$-N concentration under N

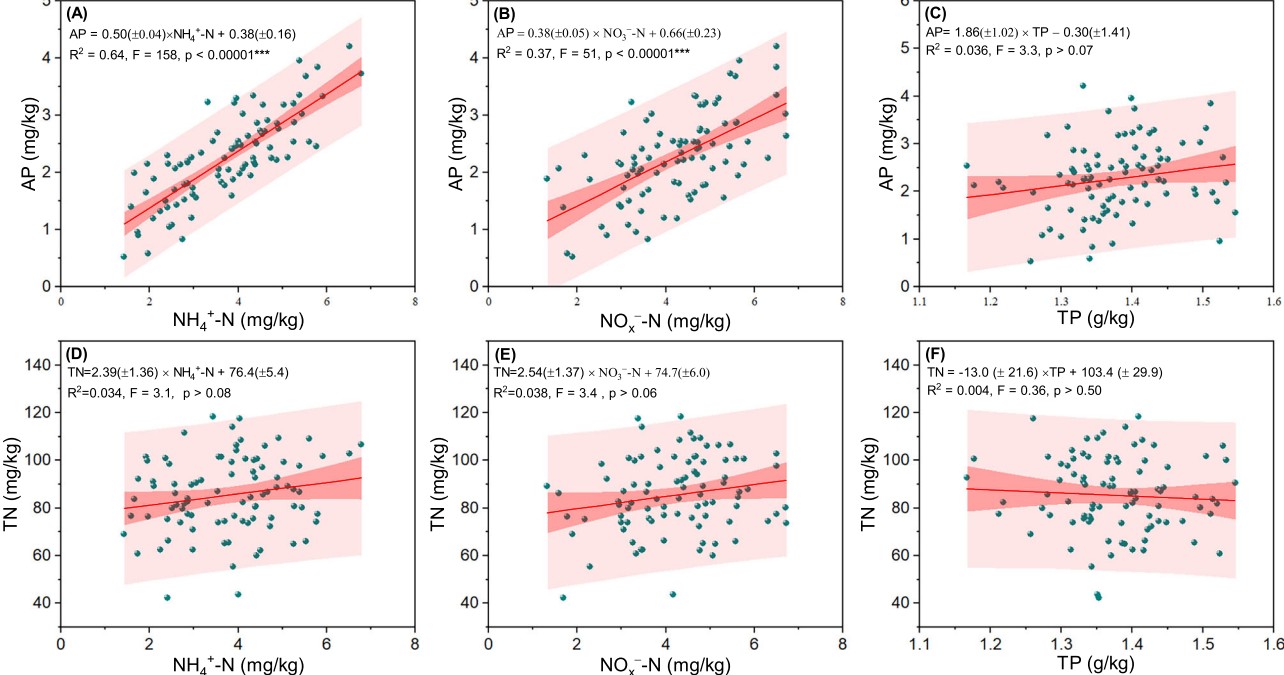

**Fig. 1 | Relationships between N and P in the early-stage soil under natural conditions (without N or P addition, *n* = 90 plot replications per treatment).** The red line represents the linear regression between two variables, the red shaded region represents the 95% confidence band for the regression, and the light red shaded region represents the 95% prediction band for the regression. **A** Linear regression between available phosphorus (AP) and ammonium nitrogen ($NH_4^+$-N). **B** Linear regression between AP and nitrate nitrogen ($NO_x^-$-N). **C** Linear regression between AP and total phosphorus (TP). **D** Linear regression between total nitrogen (TN) and $NH_4^+$-N. **E** Linear regression between TN and $NO_x^-$-N. **F** Linear regression between TN and TP.

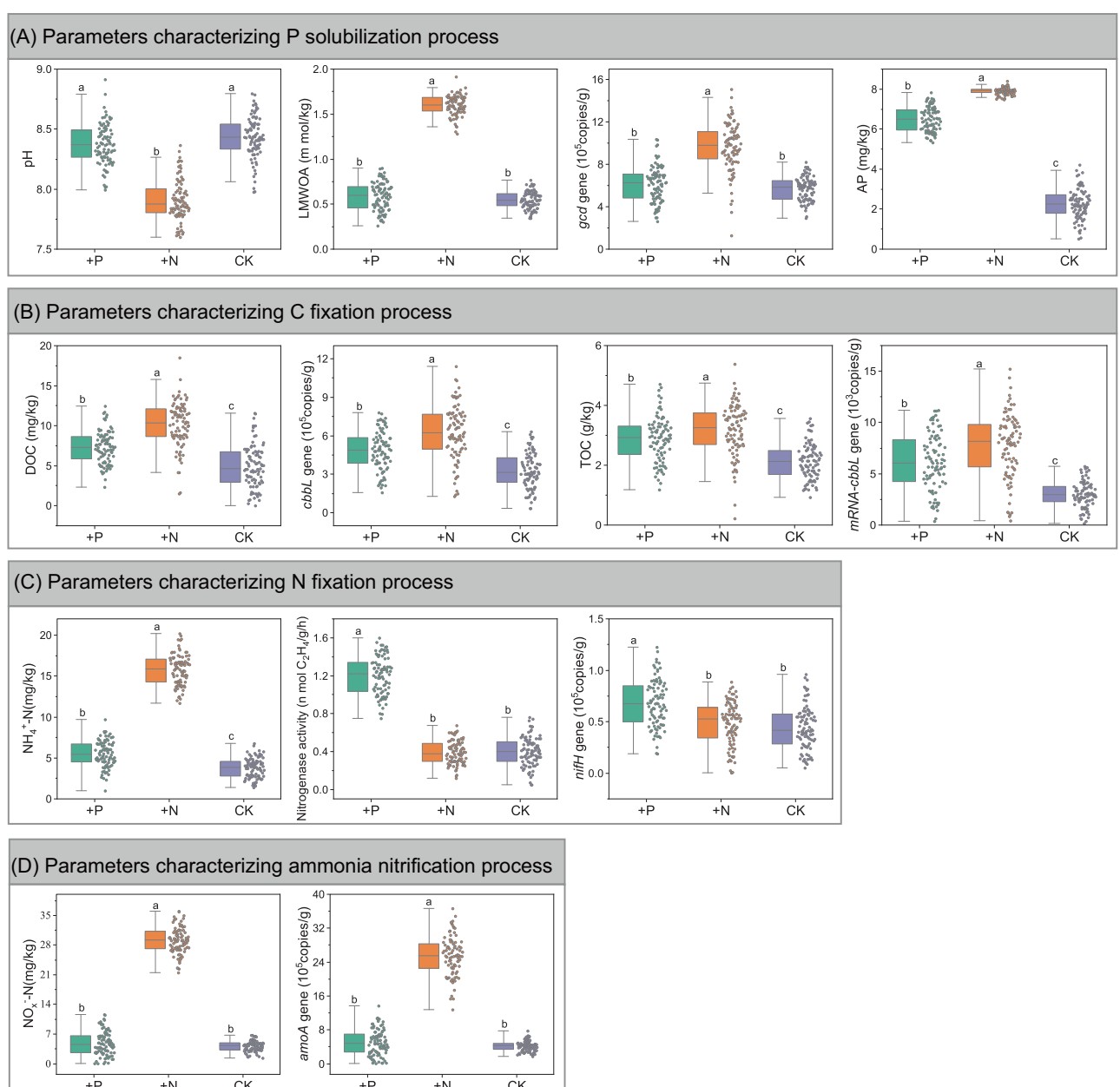

**Fig. 2 | The effects of N and P additions on soil processes in the early-stage soils over a three-year period ($n = 90$ plot replications per treatment).** The band in the middle of each box represents the median, and the top and bottom of the box represent the first and third quartiles, respectively. The error bars show 1.5× inter-quartile range. Data values of indicators are indicated by the dots on the right side of the box. Different lowercase letters over the error bars indicate significant differences ($p < 0.05$) among different treatments. The ' +P': P addition treatments; The ' +N': N addition treatments; The ' CK': control treatments. **A** Effects of N and P additions on parameters characterizing the P solubilization process. **B** Effects of N and P additions on parameters characterizing the C fixation process. **C** Effects of N and P additions on parameters characterizing the N fixation process. **D** Effects of N and P additions on parameters characterizing the ammonia nitrification process.

addition treatments is unsurprising, as it is a direct result of the added $NH_4^+$–N and not due to biological N fixation. Similarly, the increase in $NH_4^+$–N concentration also indicated that the added $NH_4^+$–N was not completely removed by precipitation and soil leaching during the experimental period, confirming that the N addition treatments were effective and indeed achieved their aim of significantly increasing soil available N.

Additionally, we speculated that pH regulation in early-stage soils is linked to the ammonia nitrification process; thus, we measured nitrate N ($NO_x^-$–N) concentration and the copy number of the ammonia oxidation gene (*amoA*) (Fig. 2D). The results showed that N addition significantly increased $NO_x^-$–N concentration and the copy number of the *amoA* gene in early-stage soils. It suggested that N

addition can lower soil pH by intensifying the ammonia nitrification process. In contrast, P addition had no significant effect on these parameters.

## Linking major soil processes to nitrogen-related and phosphorus-related processes

Based on the above results of soil processes triggered by N (or P) addition, it can be determined that the increase in available N (or available P) in the N (or P) addition treatment is the cause of the changes in soil P (or N). Therefore, it is reasonable to construct a structural equation model using available N (or available P) as the driver to explore the causal relationships of the driving effects between N and P. This approach can help elucidate the coupling

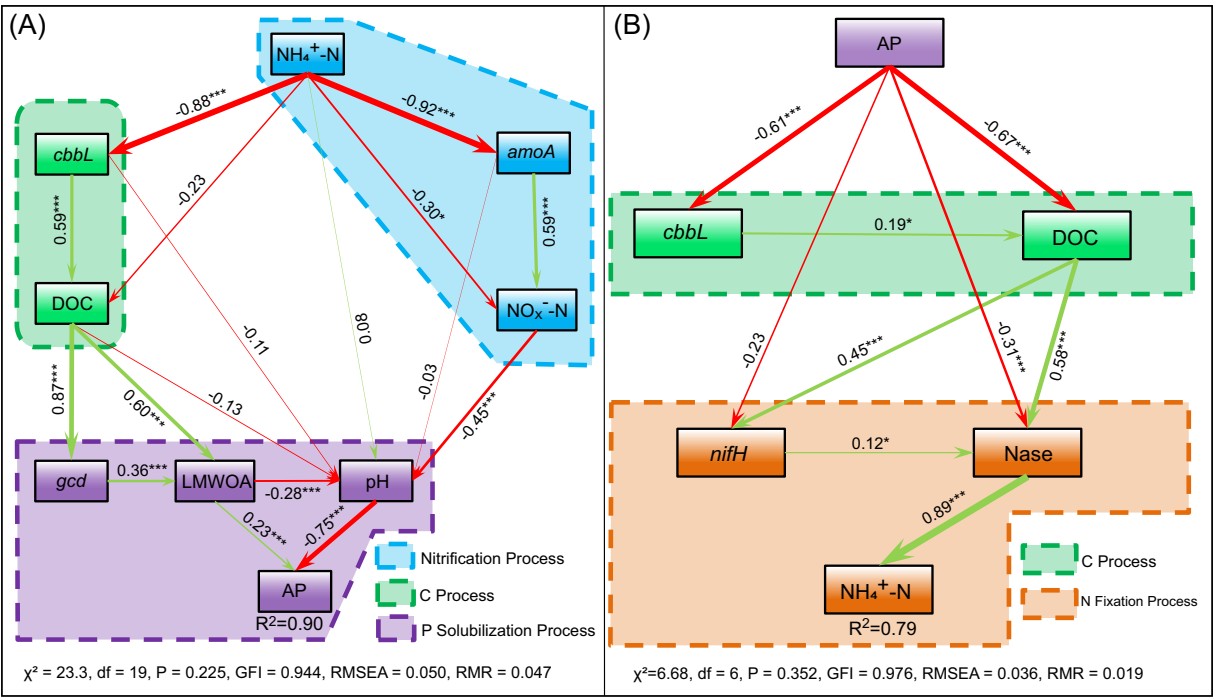

**Fig. 3 | Structural equation model of the N-driven mode and the P-driven mode based on additional treatments (sample size: $n$ = 90).** Red lines indicate positive path coefficients, and green lines indicate negative path coefficients. Arrow widths indicate the strength of standardized path coefficients. The numbers associated with the arrows indicate the path coefficients. $R^2$ values indicate the proportion of variance explained by variables. $p < 0.001$: ***, $p < 0.05$: *. NH$_4^+$–N: ammonium N; $cbbL$: C fixation genes $cbbL$; amoA: $amoA$ genes−ammonia oxidation genes associated with nitrification process; DOC: dissolved organic C; (NO$_x^-$–N: nitrate and nitrite N; $gcd$: $gcd$ genes associated with acid production process; LMWOA: low-molecular-weight organic acids; AP: available P; $nifH$: nitrogen fixing gene $nifH$; Nase: nitrogenase. **A** Structural equation model of the N-driven mode based N addition treatments. **B** Structural equation model of the P-driven mode based P addition treatments.

pathways between N and P in the early-stage soil. Overall, in early-stage soils, available N influences P availability through C fixation and nitrification processes (Fig. 3A), while AP affects N availability through C fixation and N fixation processes (Fig. 3B). Although the structural equation model showed a negative correlation between available N and the parameters characterizing C fixation (and ammonia nitrification), this indicated that added N had a stimulatory effect on the C fixation process (and ammonia nitrification), as the structural equation model was based on the results of N addition experiments (Fig. 3A), and thus the negative correlation indicates that the added N is being consumed for other soil processes. Similarly, in the structural equation model based on P addition experiments (Fig. 3B), the negative correlation associated with AP also indicated a stimulatory effect of AP on the corresponding soil processes. In addition, the structural equation model based on P addition experiments showed that the NH$_4^+$–N concentration was mainly directly influenced by nitrogenase activity. The structural equation model based on N addition experiments showed that the AP concentration was mainly directly influenced by pH and LMWOA. Both $gcd$ and $nifH$ genes were strongly correlated with DOC.

## Discussion

### The mechanism of the effect of available N on P availability in the early-stage soil

The availability of N greatly influences the microbial transformation of soil P, particularly in early soils characterized by severe N deficiency. However, the exact mechanism through which N stimulates microbial processes to release available P from these early soils remains unclear. Our field experiments have demonstrated that elevated N availability significantly enhances the P availability in the early soils. For the

mechanisms underlying the nitrogen-driven P availability, previous studies based on mature soil with vegetation have concluded that higher availability of N enhances the activity of phosphatase[39,40], resulting in an augmented hydrolysis of ester-phosphate bonds in soil organic P and ultimately improving soil P availability[17]. However, in the early soils characterized by no vegetation, the levels of soil organic P are very low (even below the detection limit), and the phosphate minerals (i.e., inorganic P) are practically equivalent to the total P in the soil[23]. Therefore, it is unlikely that the mechanism based on phosphatase activity can explain the improvement of P availability under N addition for these early soils. Our results indicated that the mechanisms underlying nitrogen-driven P availability in the early-stage soil are different from the mechanisms proposed in the previous studies[39,40]. Instead, nitrification processes and C fixation processes explained the driving effect of N on the soil P availability for the early-stage soil. On the one hand, our data showed that promoting nitrification processes is one important way that N drives the P availability through acidifying the soil environment (Fig. S3). The surge in soil NO$_x^-$–N and ammonia oxidation genes ($amoA$ genes) (Fig. 2D), in the plots with N addition treatments, clearly indicated significant nitrification processes had occurred. The nitrification processes are an important acidogenic reaction that contributes to soil acidification by releasing protons and producing nitrous acid[41]. Theoretically, during the process of nitrification, 1 mole of ammonium ions is oxidized to nitrate ions, releasing 2 moles of hydrogen ions into the environment[22]. For ammonia-oxidizing archaea (AOA), they are even able to produce 4 moles of hydrogen ions by oxidizing 1 mole of ammonium ions[42]. Hence, soil nitrification stimulated by the addition of NH$_4^+$–N is one important reason for the decrease in the soil pH. Actually, we also observed a significant decrease in soil pH in the plots with N addition treatments

(Fig. 2A). On the other hand, our results showed that stimulating the C fixation process is another way for N to improve soil P availability. In our study, compared to those in the control plots, higher levels of DOC, TOC, and C fixation genes (*cbbL* genes) (Fig. 2B) in the plots with N addition treatments indicated that the C fixation was stimulated by the N addition. Additionally, an increase in LMWOA concentration and elevated expression of acid-producing genes (*gcd* genes) were detected. These findings demonstrate that the C fixation process promotes the production of organic acids. The organic acids not only contribute to acidifying the soil environment but also chelate the cations that are bound to phosphate through their hydroxyl and carboxyl groups and thus transform rock phosphates into soluble forms[43]. RDA analysis showed that $NO_x^-$–N (as a measure of the nitrification process) and LMWOA explained 89% of the variation in soil pH (Fig. S4). Based on the above results, a structural equation model (Fig. 3A) was constructed to visualize the mechanism of N-driven P availability in the early soils. In the structural equation model, the significant negative correlations between $NH_4^+$–N and key process parameters (e.g., *amoA*, *cbbL* and $NO_x^-$–N) indicated that the added N was consumed and transformed, to drive nitrification processes and C fixation processes (Fig. 3A), and subsequently influenced the availability of P. Therefore, in this early-stage soil, the C processes and the nitrification process mediate the promoting effect of available N on P availability.

## Organic C mediates the promoting effect of P on the N availability in the early-stage soil

Our results showed that P addition significantly increased the availability of N in early-stage soil. This result is contrary to previous studies conducted on mature soils, which showed that P addition reduced soil available N[44,45]. These different results indicate that the mechanism of P-driven N availability differs between mature soil and early soil. For the majority of mature soils, plants are the dominant players in the geochemical processes of soil elements. Thus, P addition first stimulates plant growth and the uptake of mineral N from the soil[44], diminishing the available N in the soil and limiting soil microorganism production of enzymes to mineralize organic N[45]. Some studies have also proposed another mechanism, which is that when inorganic P is scarce and organic matter (containing organic P) is abundant (such as in mature forest soil), soil microorganisms may produce cellulase to degrade the organic matter containing organic P to improve P availability[46], and this process could also promote the degradation of N-containing organic matter to improve N availability. Conversely, when available P becomes sufficient (e.g., P addition), it would decrease microbial investment in organic matter decomposition enzymes and impede the release of available N from the organic matter[47,48]. Hence, P addition in the mature soil often leads to a decrease in available N. However, the early soil lacked plant colonization and rich organic matter. Those environmental conditions, which are different from mature soil, indicate that the above mechanisms cannot be used to explain the driving effect of P on N availability in the early soil. In such pristine deglaciated forefield, the early soil is an environment dominated by microbial community, and their activity is thought to be responsible for the initial accumulation of organic C and nutrients[49]. Additionally, a considerable amount of the *cbbL* gene is involved in C fixation from microorganisms ($3.4 \times 10^4$–$1.1 \times 10^6$ copies/ g in Fig. 2B, compared to the study by Tahon et al.[50]). Moreover, a significant increase in both DOC and nitrogenase activity was observed with P addition treatments. Therefore, in our opinion, the C process is a pivotal link in the mechanism of P-driven N availability for the early soil. Another important reason supporting this perspective is that N fixation requires substantial energy input. As demonstrated in previous studies[51–53], organic C serves as a crucial energy source for N fixation processes, and this organic C is primarily derived from C fixation pathways, such as photosynthesis. However, low P availability may reduce rates of photosynthesis, which in turn may inhibit

nitrogenase by reducing photosynthate supplies and, in particular, the supply of ATP[54]. On the contrary, when available P is sufficient, for example, it is added into the early soil lacking N, and the situation in this soil was very similar to eutrophication in aquatic environments. Especially, the increase in DOC, $NH_4^+$–N and bacterial genes indicated that "eutrophication" occurred in the early soil. To elucidate the mechanism by which P drives N availability in early-stage soils, we constructed a structural equation model (SEM) (Fig. 3B). In this model, whereas no significant relationship was observed between AP and *nifH* gene abundance, several significant pathways (such as the associations between AP and nitrogenase activity, AP and DOC, DOC and the *nifH* gene, DOC and nitrogenase activity) combined with the findings that AP addition treatment (Fig. 2B) significantly increased DOC concentration, TOC concentration, and *cbbL* RNA reverse-transcribed copy numbers, collectively indicate that organic C derived from C fixation processes serves as the critical link between AP and N availability. Besides, these significant pathways also indicate that AP may preferentially regulate nitrogenase activity and energy acquisition (i.e., organic C production through C fixation) rather than simply increasing *nifH* gene copy numbers. Moreover, previous studies[51–53] have established organic C as a crucial energy source for N fixation. Therefore, our findings emphasize that AP in early-stage soils enhances N availability by stimulating organic C production through C fixation processes, thereby promoting N fixation. The results and analyses above indicate that integrating C-related processes into the interaction framework of the N-P cycle plays a key role in refining the existing N–P coupling theory.

## Implications for the driving mechanisms of primary succession

Our study site is located at the beginning of a soil chronosequence (with soil ages ranging from 0 to approximately 130 years) that features a complete primary vegetation succession[23,55], and the soils at our site exhibit very low N and P availability (Fig. 2B). According to traditional views[2], plants (and microbes) are expected to allocate their resource reserves toward strategies that increase the acquisition of the most limiting resources—thus moving toward a state where all resources simultaneously limit productivity and growth. Despite the low availability of nutrients in early soils limiting ecosystem development, the forward succession of ecosystems on the soil chronosequence does not align with the above views. This forward succession, progressing from bare ground to a forest, and the gradually increasing nutrient availability suggest that nutrient limitation is improving rather than worsening. This involves the origins of the establishment of nutrient provisioning capacities in subsequent, more developed ecosystems, and it implies that the early soil processes serve as the starting point for the formation of mechanisms that couple elements such as N and P. This primary succession sequence provides macro-scale evidence of the mutual enhancement of N and P availability in early soils. Finally, we used a conceptual diagram (Fig. 4) to illustrate the process of mutual promotion of N and P availability in the early soil. In conjunction with the findings of this study regarding the improved availability of nutrient elements, it strongly suggests that such inter-element coupling interactions are a significant fertility driver for primary vegetation succession on early-stage soils.

For plant-dominated soil ecosystems, previous studies have indicated that increased N deposition exacerbates P limitation[56,57], which in turn affects plant growth and ecosystem C uptake[58–60]. However, for early-stage soil ecosystems devoid of plants, our results show that N addition not only improves the availability of soil P but also increases the content of DOC and TOC. This clearly indicates that N deposition does not exacerbate P limitation in early-stage soils; rather, it enhances P nutrition and promotes C fixation. This also suggests that the method of using N-to-P ratios to assess N (or P) limitation in soil ecosystems has limitations and should be applied with caution. This perspective is also supported by previous case studies[61].

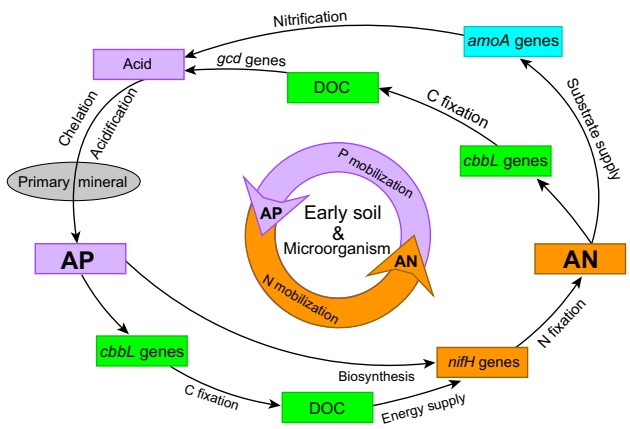

**Fig. 4 | A conceptual model diagram illustrating the process of mutual promotion of N and P availability in the early soil.** AP available P, AN available N, DOC dissolved organic C, *amoA* genes ammonia oxidation genes associated with the nitrification process, *cbbL* genes C fixation genes *cbbL*, *nifH* genes nitrogen fixing gene *nifH*.

In conclusion, this study demonstrated that in the earliest stages of soil development, the enrichment of one nutrient can improve the availability of other nutrients and drive positive succession in ecosystems, rather than exacerbating nutrient limitation as traditionally believed. This strongly implies that such element inter-coupling interactions represent an important fertility driver for primary vegetation succession on early-stage soils. Furthermore, our study demonstrated C-mediated co-amplification of N and P mobilization in early-stage soils. These mediating processes are conceptualized within structural equation models as the P-driven mode and the N-driven mode, respectively. These models provide an integrated prototype for integrating C-related processes into the interaction framework of the N–P cycle to refine the existing N–P coupling theory.

## Methods
### Study area
The study location is situated within the most recently deglaciated region of the Hailuogou Glacier (coordinates: 29° 34′ 08″ N, 101° 59′ 41″ E, Fig. S1), which lies in the transitional area between the Tibetan Plateau and the Sichuan Basin in southwestern China. The study site is at an elevation of 2990 meters above sea level (a.s.l.). The annual precipitation at this location varies between 1586 and 2175 mm. The soil temperature, measured at a depth of 10 cm, fluctuates throughout the year from a minimum of −3 °C to a maximum of 21 °C. The soil within the study area has been exposed for a period not exceeding 18 months and is characterized by an absence of vegetation. Examination of the soil property data reveals that, in precise terms, the soil in question more closely resembles rock debris, notwithstanding the presence of a minor quantity of organic matter. According to previous studies conducted by our team[62,63], the dominant P mineral in the study area is apatite, accounting for approximately 2% of the total soil mineral mass. Additional soil characteristics are delineated in Table 1.

### Field in situ experiments and soil sample collection
We conducted a large-scale, three-year in situ nitrogen (N) and phosphorus (P) addition experiment on early-stage soils in the Hailuogou glacier retreat area, with over 270 plots established. This experiment is particularly significant (beyond its focus on "early-stage soils") because the early soils in this study area can develop into a soil ecosystem with approximately 25% pioneer plant cover within about 6 years[51]. Consequently, for the early-stage soil ecosystems in this region—characterized by the absence of vegetation and rapid development—a three-year

study period can be regarded as a "long-term" study. Thus, our experiment can be defined as a "long-term in situ N and P addition experiment targeting early-stage soils."

In May 2019, a substantial number of plots (>270) were established in the study area to conduct the in situ N and P addition experiments. Each plot measured 0.3 m × 0.3 m, with spacing between plots exceeding 4 m. The experiment included three treatments: N addition (+N), P addition (+P), and CK (Table S1). Given the complex microtopography, frequent geological disturbances (e.g., small-scale debris flows and landslides), and significant microenvironmental heterogeneity in the study area, to enhance the accuracy of experimental data and mitigate data loss from potential plot damage caused by natural disasters, we randomly allocated over 90 plots per treatment as replicates (i.e., the total number of plots assigned to the three treatments exceeded 270). This design also ensured the collection of 90 replicate samples per treatment (a total of 270 samples for the three treatments) by the end of the experiment. For each of the over 270 plots, the experimental setup involved the following steps:

Soil column extraction: Using an undisturbed soil sampler (inner diameter: 10 cm), we extracted an intact soil column (dimensions: 15 cm height × 10 cm diameter, preserving its natural structure) from each plot. The soil column was placed into a tubular polyethylene bag to ensure thorough interaction between subsequently added N (or P) and the soil column, prevent lateral leaching of N (or P) and avoid cross-contamination between plots.

Reinstallation: The soil column was then returned to its original location (Fig. S1).

Nutrient addition protocol:

Given the high permeability of early-stage soils and referencing prior studies[64–66], N and P were not applied in a single dose. Instead, the following protocol was implemented:

N addition: Starting in May 2019, N was applied to +N treatment plots at a rate of 10 g N m⁻² year⁻¹. Specifically, $NH_4Cl$ solution ($NH_4Cl$ concentration: 10.00 mg/ml) was evenly sprayed onto the soil column surface (area: 78.5 cm²) using a sprayer (brand: Lizhen; capacity: 100 ml; spray volume per press: 0.132 ml). Each application involved 5 ml of solution (achieved via 38 presses), repeated every two months.

P addition: Similarly, +P treatment plots received 10 g P m⁻² year⁻¹ via $NaH_2PO_4$ solution (concentration: 10.13 mg/ml), applied identically using the same sprayer (5 ml per application, 38 presses).

CK treatment: For CK plots, pure water was sprayed using the same method.

Additionally, as a reference, the combined nitrogen and phosphorus addition treatment was also implemented (Fig. S2). In summary, the above-mentioned in-situ experimental method ensured that in July 2022, three years later, we collected a sufficient number of soil samples (i.e., 90 replicates for N addition treatments, 90 replicates for P addition treatments, and 90 replicates for CK treatments, totaling 270 soil samples). The in-situ experiment was concluded in July 2022, and soil columns (0–5 cm) in each plot were collected as soil samples. The collection method was as follows: For each plot, the tubular polyethylene bag enclosing the intact soil core was excavated as a whole, and the soil at a depth of 0–5 cm of this core was collected and placed into a sterile sample bag as one soil sample. Altogether, we collected 270 fresh soil samples. Each of them was sieved through a 2-mm mesh before being divided into two subsamples. One subsample was stored at 4 °C for the measurement of soil physicochemical characteristics, and the other subsample was stored at −80 °C for gene analysis.

### Quantification of functional genes
Soil total DNA was extracted from soil samples using a FastDNA® Kit for Soil (MP Biomedicals, Ohio, USA) following the manufacturer's instructions with minor modifications as necessary. Briefly, approximately 0.25 g of soil was mixed with lysis buffer and glass beads in a microcentrifuge tube. The mixture was vortexed vigorously for several

minutes to lyse the cells. Following centrifugation to pellet debris, the supernatant containing the released DNA was transferred to a new tube and processed through a series of washes and elutions according to the kit protocol. The purified DNA was eluted in 100 µL of elution buffer provided with the kit and stored at −20 °C until further use. The concentration and purity of the extracted DNA were determined using a Nanodrop® ND-1000 UV–Vis spectrophotometer (NanoDrop Technologies, Delaware, USA). The extracted DNA will be used as template DNA for the quantification analysis of functional genes.

Real-time quantitative polymerase chain reaction (qPCR) was conducted using a CFX96 optical real-time detection system (Bio-Rad, Hercules, CA, USA) to quantify the abundances of soil functional genes. The PCR reactions were performed in a total volume of 25 µL, comprising 1× qPCR-SYBR Premix Ex Taq (a premix of dNTPs, Taq DNA polymerase, PCR buffers, and SYBR Green; Takara Biotech, Dalian, Liaoning, China), 5 µM of each gene-specific primer (as listed in Tables S2), and 1 µL of template DNA. The templates for the standard curve were prepared by performing 10-fold serial dilutions of linearized recombinant plasmids that contained the amplicon amplified from soil DNA. These recombinant plasmids, which harbored one copy of the corresponding gene fragment, were diluted from $1:10^1$ to $1:10^{10}$ and used as targets for constructing the qPCR standard curves. For the functional genes, qPCR was performed using soil DNA as the template, with four different dilutions ranging from $1:10^1$ to $1:10^4$, under optimized conditions (as detailed in Table S2). At the conclusion of the qPCR cycles, a melt curve analysis was conducted to verify the specificity of the PCR amplification. The experiments were independently repeated three times, and the amplified products were subjected to agarose gel electrophoresis to confirm the expected sizes. The qPCR data were analyzed using the comparative cycle threshold value ($C_T$) method, and the copy number of the targeted gene was determined[67].

Total RNA was extracted from 1.0 g of soil samples using the FastRNA® Pro Soil-Direct Kit (MP Biomedicals, Inc., CA), following the manufacturer's protocol. To ensure complete removal of residual DNA contamination, the extracted RNA was treated with the TURBO DNA-free™ Kit according to the manufacturer's instructions. Following the methodology described by Crépeau et al.[68], prior to purification using the RNeasy minikit (QIAGEN™), EDTA was added to the extracted products to achieve a final concentration of 15 mM, followed by incubation at 65 °C for 10 min to terminate the DNA digestion reaction. The completeness of DNA removal was verified through direct PCR amplification. For soil samples exhibiting low RNA concentrations, additional aliquots were processed through repeated extraction procedures. The resulting extracts from the same soil sample were pooled and concentrated through vacuum freeze-drying at −55 °C to obtain RNA of sufficient concentration and total yield for subsequent analyses.

The construction method of the standard curve for quantitative RNA is as follows: The DNA extracted from the strain of cyanobacterium Calothrix sp. PCC 7716 was used to amplify the cbbL gene with primers. The specific restriction enzyme cleavage sites EcoRI/BamHI were introduced through PCR. Subsequently, the gene fragment was cloned into the multiple cloning site of the pGEM-3Z vector (Promega Corp., Madison, Wis.). The recombinant plasmid was formed through the digestion of restriction enzymes and the reaction of ligase. The recombinant plasmid was transformed into competent Escherichia coli cells. Positive clones were screened using ampicillin resistance, and the correctness of the inserted gene was verified by colony PCR and sequencing. After the verified plasmid was linearly processed with restriction enzymes, it was used as a template, and cbbL mRNA was transcribed using the MAXIscript in vitro transcription kit (Ambion). The transcription product was treated with DNase I to remove the template DNA, and impurities were removed by the Dynabeads™ RNA Purification Kit (Thermo Fisher Scientific Inc.). The concentration of the purified RNA was determined using a DNA fluorometer to ensure its precise quantification for constructing the RNA standard curve.

According to the manufacturer's instructions, the above sample RNA and standard RNA were reverse-transcribed into cDNA using the High-Capacity cDNA Reverse Transcription Kit (Thermo Fisher Scientific), followed by DNA amplification.

Based on the primers (primer K: 5′-GCGAATTCAA(AG)CC(TA)AA(AG)(TC)TAGG(TG)(CT)T(AT)TC-3′; primers V: 5′-AGGGATCC(TC)TC(TC)A(AG)(TC)TTACC(AT)AC(GAT)AC-3′)[69] (Xu and Tabita, 1996), qPCR was performed on the cbbL cDNA using the QuantiTect SYBR Green PCR Kit (Qiagen) according to the manufacturer's instructions.

## The determination of soil properties

The pH value of the soil, prepared with a soil-to-water ratio of 1:2.5, was ascertained with a pH meter. Soil moisture content (SM) was quantified through the gravimetric method, involving the oven-drying of soil samples to a state of constant weight at a temperature of 105 °C. Soil temperature (ST) measurements were recorded at a depth of 10 cm using a thermometer at the site of sample collection. The bulk density (BD) of the soil was determined using the following method: natural soil samples were collected with a ring sampler (volume 100 cm³), and subsequently oven-dried until reaching a constant weight. The BD was then calculated as the ratio of the oven-dried soil weight to its known volume (100 cm³). The available P (AP) concentration in the soil was extracted using a 0.5 M NaHCO₃ solution (pH 8.5) at a soil-to-extractant ratio of 1:60 and shaken on a reciprocal shaker at 280 rpm and 25 °C for 30 min. The AP concentration was then determined using the molybdenum blue colorimetric method[70]. TOC and TN in the soil were measured using an elemental analyzer (Elementar Vario EL). Specifically, prior to determining the TOC, soil samples were soaked for 48 h in diluted hydrochloric acid to remove carbonates.

DOC in the soil was measured by extracting with deionized water (soil:water ratio = 1:4) through shaking at 280 rpm for 1 h, followed by centrifugation at 10,000 rpm for 5 min, and filtration through 0.45 µm Millipore filters. The extraction solution was acidified prior to measuring the DOC concentration using a total organic carbon analyzer (Elementar Vario TOC). The concentrations of ammonia N ($NH_4^+–N$) and nitrate N ($NO_x^-–N$) were analyzed using a flow autoanalyzer (SEAL Analytical Auto Analyzer 3) following extraction with 2 M KCl. Soil nitrogenase (Nase) activity was assessed by measuring the potential N fixation rate using the acetylene reduction assay[53,71]. The LMWOAs in the soil were extracted using a 0.1% $H_3PO_4$ solution (soil:extractant ratio = 1:1) through shaking for 1 min, followed by centrifugation at 10,000 rpm for 5 min, and filtration through 0.45 µm Millipore filters. The total concentrations of LMWOAs were then measured by high-performance liquid chromatography (HPLC, Shimadzu, Japan)[72]. Soil samples (0.5 g) were digested using an HCl–HF–HNO₃–HClO₄ mixture, and the concentrations of Fe, Al, and TP were determined using an inductively coupled plasma atomic emission spectrometer (ICP-AES, American Leeman Labs Profile).

## Data analyses

Data were tested for normal distribution using the Shapiro–Wilk test and for homogeneity of variances using Levene's test. Where data did not meet the assumption of normality, appropriate transformations were applied. One-way analysis of variance (ANOVA) followed by Tukey's honestly significant difference (HSD) test was used to assess the effects of different treatments on soil properties, functional gene abundances, and enzymatic activities. Especially for data with unequal variances, the Games-Howell test was applied. Linear regression models were utilized to explore the relationships between N fractions (or N fractions) and among them. The normality of residuals of the linear regression was checked by Q–Q plots. Structural equation modeling (SEM) was employed to quantify the relative importance of direct and indirect pathways influencing soil processes. ANOVA and

linear regression analyses were conducted using IBM SPSS Statistics 21 and Origin 2022, respectively. SEM was performed using AMOS 21.0. In this study, significant differences were considered at a statistical level of $p < 0.05$.

## Reporting summary

Further information on research design is available in the Nature Portfolio Reporting Summary linked to this article.

## Data availability

Source data are provided with this paper. They are available as data files in the Supplementary_Information.pdf and Source Data file.xlsx.

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

## Acknowledgements

This work was supported by the National Natural Science Foundation of China [grant numbers: 42271073 (H.S.), 42477329 (J.Z.) and 42311540015 (J.Z.)] and supported by the Southwest Minzu University Research Startup Funds[RQD2024012 (H.S.)] and Qinghai-Tibet Plateau unique grass germplasm resources and grassland ecological restoration innovation team project [2024CXTD11 (H.S.)]. The funders had no role in study design, data collection and interpretation, or the decision to submit the work for publication.

## Author contributions

H.S. and D.Y. conceived and designed the study. H.S., D.Y., J.Z. and Y.C. provided the methodology. H.S., D.Y. and J.Z. designed and performed N and P addition experiments. H.S. and J.Z. isolated DNA from soil

samples. H.S., D.Y. and J.Z. measured the physical and chemical properties of soil samples. H.S., D.Y., J.Z. and Y.C. completed quantification of genes. H.S. and D.Y. did the data analysis. H.S. and D.Y. wrote the original draft of the paper. J.Z. reviewed and edited the paper. D.Y. administered the project. H.S. and D.Y. acquired funds. All authors approved the paper before submission.

## Competing interests

The authors declare no competing interests.
