## [Peer Review file · Nature Communications]

Large-scale experimental evidence of carbon-mediated N and P co-amplification in proglacial soils

Corresponding Author: Dr Hongyang Sun

Version 0:

Reviewer comments:

Reviewer #2

(Remarks to the Author)

The title : Insights into the driving effects between nitrogen and phosphorus in early soils of a glacier retreat area: a massive in situ nitrogen and phosphorus addition experiment

The work is interesting, and the manuscript is well written and structured. Overall, this is an excellent study that provides significant value in enhancing our understanding of the coupling mechanisms of nitrogen and phosphorus in soils. Firstly, I greatly appreciate the research perspective presented in this manuscript. Rather than focusing on mature ecosystems as is common in previous studies of N and P cycling mechanisms, this study explores these mechanisms from the vantage point of early soils, which are the origins of mature ecosystems and involve relatively simpler processes. This approach is particularly advantageous for elucidating the mechanisms that are obscured by complex interactions in more developed systems. For instance, the experiments conducted in this study reveal that the enrichment of one nutrient in the soil can enhance the availability of other nutrients, a finding that aligns well with my expectations and starkly contrasts with the traditional view that the abundance of one nutrient would lead to limiting effects on others. Therefore, I find this research to be highly innovative. Secondly, for a field experimental study, the volume of data utilized in this research is remarkably large, even comparable to that used in some meta-analyses. I am confident that the reliability of the data analysis results from this study surpasses that of many similar studies.

However, the manuscript also has several issues that need addressing, and therefore I recommend publication after major revisions. The specific issues are as follows:

Line 10: "N and P" should be used as abbreviations for nitrogen and phosphorus.

Line 40-42: "In soil environments, the bioavailability of N and P is governed by a suite of intricate soil processes, including N fixation, phosphate dissolution, organic matter mineralization, and biological uptake." These processes should also include soil leaching.

Line 43: "...the impacts of N or P availability...." The word "or" fits better in this context than "and".

Line 61: please provide a reference for "These processes are influenced by N availability"

Line 63: please provide a reference for "...as the N element is an important component of phosphatase proteins.

Line 67, 74: nitrogen > N

Line 64-76 : In fact, these meta-analyses with large datasets also highlight the importance of case studies that utilize large data volumes

Line 112-115: The authors have not fully elucidated why "the nutrient limitation in early soil ecosystems is distinctive.

Line 353: please specify why 10 g m⁻² year⁻¹ was adopted?

Line 131: AP? Write out the full name

Line 125-135: The author provides a description of the soil physicochemical properties under natural conditions for the early soils in the study area, which is excellent. However, I believe that information on phosphorus minerals in the soil of the study area (such as apatite) should also be provided. This information is crucial for extending the research findings to mature ecosystems.

Line 136: "Highly significant?" Other sections also contain similar phrasing. I believe it is not appropriate to describe the statistical p-values of research results from highly variable environments using binary terms such as 'significant' or 'not significant.' This approach can mislead readers about the precision of the study results. I recommend adopting the 'Evidence language' method proposed by Stefanie Muff et al., 2022.

Line 140-141: In this study, each experimental treatment includes 90 replicates, which is quite remarkable for a field experiment. To justify the necessity of such a large number of replicates, I recommend demonstrating the variability of the data in the supplementary materials. This would provide readers with a reference for evaluating the credibility of previous field experiments, as many typically do not exceed 10 replicates. I believe this approach would greatly assist readers in assessing the reliability of past studies.

Line 153: DOC? Write out the full name

Line 192: LMWOA? Write out the full name

Line 194: The Discussion section could be organized into several subsections with headings for clarity. For example, lines 195-236 could be headed 'Mechanism Underlying Nitrogen-Driven Phosphorus Availability in Early-Stage Soils'. Additionally, in the discussion, I recommend extending the discussion to speculate on the mechanisms related to nitrogen and phosphorus in mature soils (or vegetation-dominated soils) based on your data results. This would significantly enhance the depth of the discussion in this manuscript.

Line 223: CK? Write out the full name

Line 240, 290, 296: mechanism > mechanisms; nitrogen and phosphorus > N and P; nitrogen addition > N addition

Line 314-316: The content of this sentence should be moved to the Discussion section. It should correspond with the description of phosphorus minerals in the Supplementary Materials.

Line 350-356: Present the experimental design in tabular form and include it in the Supplementary Materials.

Reviewer #3

(Remarks to the Author)

The manuscript entitled "Insights into the driving effects between nitrogen and phosphorus in early soils of a glacier retreat area: a massive in situ nitrogen and phosphorus addition experiment" focuses on important nitrogen (N) and phosphorus (P) processes in the early soils of a glacier retreat area, which lack vegetation. The authors conducted a three-year experiment involving 270 N or P addition treatments and presented intriguing results, showing that ammonium addition enhanced P solubilization, while P addition increased N₂ fixation. They concluded that N and P processes exhibit a positive coupling relationship in early soils, which contrasts with mature soils covered by vegetation.

However, I believe the manuscript still have several notable flaws and could be improved in several aspects:

1. Lines 125-135 and Table 1: This section describes the natural state of the early soils. However, key details are missing, such as when and how the soils were sampled, the number of replicates, and whether the sampling occurred before, during, or after the three-year experiment. It is also unclear whether the natural state of the soils remained consistent over the three-year period without N and P addition.
2. Lines 153-156 and Figure 2B: The *cbbL* gene copy number showed only minor changes, with slight differences between N and P additions, despite statistical significance. However, *cbbL* gene copy number does not always correlate with C fixation activity. Direct measurements of C fixation rates or qPCR analysis of *cbbL* RNA transcripts would provide more robust evidence. Given the large variation in this study, small changes in *cbbL* gene abundance, and the lack of C fixation rate measurements, the conclusion that N and P additions increased C fixation may not be fully supported.
3. Organic Carbon Data: Figure 2 and many places in the manuscript present only dissolved organic carbon (DOC) data, excluding total organic carbon (TOC). Since DOC represents only a fraction of the organic carbon fixed by microorganisms, TOC is a critical parameter. Without both TOC and DOC data, it is difficult to draw right conclusions.
4. Lines 157-167: Nitrogenase activity is a more reliable indicator of N₂ fixation than *nifH* gene copy number. It is well known that many bacteria/proteobacteria, possess the *nifH* gene but do not actively fix N₂. Some lack the complete set of genes required for nitrogenase function, while others, despite having the full gene set, exhibit no detectable N₂ fixation activity. I would suggest that gene abundance, such *cbbL* and *nifH* gene, may not be good indicators in this particular study.
5. Lines 168-170, 219-220: The data show a decrease in pH alongside increased nitrate concentration and *amoA* gene abundance. To conclude that nitrification significantly contributed to the pH decrease in ammonium addition groups, additional experiments are needed to quantify the pH change due to nitrate generation from ammonium via nitrification in this study.
6. Lines 174-193: The two models presented are somewhat confusing due to numerous negative correlations. Ammonium-N in the N addition group represents residual ammonium in the soil, and available P (AP) represents residual P. Including these residual concentrations in the models may not be appropriate and could complicate interpretation.
7. Lines 227-230: The manuscript discusses organic acids but does not quantify their contribution to pH reduction.

8. Lines 257-258: The claim of “a large number of C fixation genes” lacks comparative data, making it difficult to assess the validity of this statement.

9. Lines 261-262: The manuscript incorrectly states that C fixation provides energy for N₂ fixation. Photosynthesis converts light energy into chemical energy, which can support cellular processes, including N₂ fixation. However, C fixation itself refers to the conversion of inorganic carbon (e.g., CO₂) to organic carbon. This distinction should be clarified.

10. Lines 267-276: The proposed link between AP and available N via C fixation may not be valid. As noted earlier, C fixation does not provide energy for N₂ fixation, and the evidence for increased C fixation is weak (see comments on cbbL gene copy numbers and TOC). Additionally, P addition might promote N₂-fixing microorganisms that do not depend on available N, while other microorganisms remain N-limited. It should be noted that some N₂-fixing microorganisms, such as cyanobacteria, can perform both photosynthesis and N₂ fixation. Therefore, AP directly contributes to N₂ fixation and C fixation may not be the link between AP and N₂ fixation

Reviewer #4

(Remarks to the Author)

Sun et al. Insights into the driving effects between nitrogen and phosphorus in early soils of a glacier retreat area: a massive in situ nitrogen and phosphorus addition experiment. Nature Communications

Sun et al. perform a nutrient addition experiment (N,P) in early seral habitat in an attempt to understand available N & P dynamics. If I understand their experimental design correctly then they tested the effect of three treatments on various soil properties and N cycling functional genes with 30 randomly assigned plots per treatment. Each plot replication contained a soil core incubated in a plastic bag to ensure that additions of nutrients were kept in contact with the soil and didn't leach away. Post-incubation the soil cores were assayed. N&P additions were provided as a solution periodically sprayed onto the top of each bagged soil core. It isn't clear that a similar amount of water was sprayed onto the control cores to control for moisture additions (which might possibly affect organic matter if water limited). The authors probably need to elaborate on how the nutrient addition amounts were carefully controlled with a sprayer (e.g. make and model of sprayer, calibration info. etc.) in place of the more conventional approach of pouring a fixed amount of solution into bags.

In the Introduction the authors cite numerous meta-studies on related topics and appear to use these meta-studies as motivation for the current study. This is somewhat confusing. A central problem with the study's presentation is that the authors don't clearly distinguish their study from prior work or adequately highlight the novelty of their study relative to prior work. Their study is likely somewhat novel because they applied nutrients to an area undergoing primary succession unlike most studies but exactly why that is needed or necessary isn't clear. Their study appeared to mainly be motivated by the fact that three similar meta-studies reached divergent conclusions. However, this isn't convincing motivation for doing yet another nutrient addition field experiment. In fact, the reality that there are now multiple meta-studies on the topic is an indication that the topic is well studied. So the motivation for the present study was a bit unclear and was not framed well. Further, this had me questioning why the authors submitted the current study to a journal that expects studies to make important advances within a field. Nutrient addition studies are possibly the most common field manipulation in ecological studies.

Many of the topics presented in the paper on N and especially P cycling were only superficially explored in the Introduction. I expected these topics to be well developed and that they would lead the reader toward a researchable problem. Sentences often lacked citations or mainly cited meta-studies. Overall, their study appeared to be mainly inspired by meta-studies.

The writing was often vague and topics were not well developed. Below are many suggestions to help improve the writing.

Additional comments-

L1 “Insights into the driving effects between nitrogen and phosphorus”- wordy & vague

L7 “But, due to the complexity of processes in mature ecosystems, a variety of inconsistent conclusions have been drawn.” – Authors then go on to mention how studying primary successional systems may help to improve understanding of mature ecosystems. This seems like a long shot. I think the authors are trying to say, “look scientists have spent a lot of time studying mature ecosystems and barely understand what is going on. So we are motivated to study these processes during primary succession.” However, it seems more direct to say while mature ecosystems have been extensively studied little is known about X in newly developing ecosystems. [or something similar] I think you are mostly trying to identify a gap in understanding. And primarily citing meta-studies fails to show that problem.

L10 “initial processes of nitrogen and phosphorus and their interactions”- unclear. N & P aren't processes... Also, it isn't clear what you measured or why?

L12 “N & P interactions”- This sounds like you want to study how the biogeochemistry of these two things interact. I suspect you actually have a different aim that isn't being described well here.

L13 Another nutrient addition experiment... Is there a hypothesis?

L14 This sounds like a fishing expedition without a testable hypothesis.

L15-16 This seems detached from logic (add stuff there should be more not less right) and readers aren't presented with counter point for mature systems in Abstract introduction... Please rework.

L17 "soil carbon (C) fixation processes" do you mean "soil carbon sequestration"?

L20 "P solubilization" do you mean "P addition"? Also, I wonder if this is backwards. You added nutrients so isn't possible that nutrient additions accelerate various processes that might have been limited by initial soil nutrient levels? What about co-limitation?

L22 Unclear the basis for this interpretation or modes of what? Passage feels like a word salad.

L23 same

L29 "The mobilization of nitrogen (N) and phosphorus (P) constitutes a fundamental and critical soil process, with the bioavailability dynamics of these elements being among the most ecologically significant outcomes in this process." Wordy and redundant.

L36 I would probably replace citations 1,2 with (Fay et al. 2015, Du et al. 2020).

L36 "Most notably, in terms of mechanism, the prevalence of N and P co-limitation hinted strongly the couplings between these two nutrients in ecosystems2."- wordy & confusing

L38 "The establishment and maintenance of this coupling relationship cannot be separated from the most active components (i.e., available N and P)." – This doesn't make sense to me. My impression is that N is highly labile while available P is not and sorbs to numerous soil components (clay, Ca, Calcium carbonate, Fe, Al). The next sentence hints at some of these issues but seems to ignore how fundamentally different these two nutrient behave in the soil solution and interact with other things.

L41 This statement feels a bit superficial and doesn't include citations. P is incredibly immobile (Jungk 2002) and that should probably be noted and possibly differentiated from N. Also, it seems like 10x more research has been conducted on N.

L44 phosphatase activity concerns mineralization of organic matter. In early seral environments, there is little OM. I would suspect that other processes (weathering, P solubilizing bacteria) are likely more important for P mobilization.

L44-45 It seems most of your citations are for nutrient additions. Are these realistic and/or good ways for understanding processing affecting nutrient availability in most natural systems? It seems like a crude tool, especially for understanding P availability which can be sorbed to various soil particles (Jones et al. 2013b).

L47 I think only one of the supporting sentences in this paragraph included citations...

L48 "interrelated"- I don't follow how N and P availability is interrelated.

L53-55 I don't follow the logic of this statement. Occurring plants are known to use different strategies to acquire P (e.g. Raven et al. 2018).

L55-57 I don't follow the logic of this statement either. Poor word choice. What two analytical perspectives?

L58 perspective on what? Poor word choice and poor starting sentence of a paragraph (which is often the paragraph's topical sentence).

L59 Again, nutrient addition seems like a rather crude tool to build understanding.

L60 Again, this feels superficial as there is no mention of the conditions that might cause one to predominate over the other. These are some relevant studies to look over (e.g. Ryan et al. 2012, Reinhart et al. 2024 and citations therein).

L62 Don't all enzymes require N?

L64 "conclusions" on what?

L63-7 wordy

L77 Meta-analysis tailor their study one questions with sufficient data to justify tests. They are effectively hamstrung by the prelevance of data (often collected by other people).

L79 Good point but content seems like something that should have been mentioned earlier.

Comment- The authors seem to be using the inconsistent results of three global meta-analyses as motivation for the current study. This seems like a weak motivation for an empirical study. For example, the presence of meta-analyses alone indicate that there is already a lot of work on the topic. How is your work unique. If you are simply conducting a replication of prior studies then that work should be targeted to journals like PLoS One; Agrosystems, Geosciences & Environment; etc..

L80 This seems unlikely. Many soil scientists have studied the effects of liming (to counter act soil acidification with fertilization) on available P and effects of fertilizer on soil pH (which can regulate available P). It is well known that N additions acidify soil and this can increase soil P depending on the soil's starting pH.

L88 "promoting effect of P is always significant." – vague & poor word choice. "This" refers to what?

L98 good point

L109 You should clearly (re-)state the research gap that is being addressed.

L113 citation #2- I think you mean optimal allocation theory (Bloom et al. 1985).

L116 sentence has similar problem to L109.

L146 poorly worded sentence. Try- X positively affected Y and negatively affected Z.

L151 leaching- What about sorption? If the sorption capacity of the soil is high that there may be no (or a very temporary) increase in AP (Jones et al. 2013a, Jones et al. 2013b).

L173 Okay but pH was affected by N addition right (Fig. 2)?

Discussion (L194)

L340 It is somewhat curious that with 90 replicate pots (+N, +P, and control) that you didn't include an N&P addition and reduce your replication number per treatment.

L348 "to prevent lateral leaching of N or P"- I sort of understand the decision to bag and reinsert cores but this also adds layers of disturbance (coring) and abstraction (bagged soil that prevents leaching) which makes the results a bit unrealistic compared to the more common practice of applying fertilizer to entire plots.

L360-1 So each sample (270) were separately assayed for physicochemical properties and genes? Were any samples lost or destroyed [I think that was motivation for the large number of plots]?

L390 You probably need to spell out the samples size for the various measurements (ST, BD) that were not based on the 270 soil cores.

L589 Fig. 2 header should mention the number of plot (or core) replications per treatment (e.g. n= 90 plot replications per treatment)

Fig. S1 This figure is somewhat helpful but I was expecting to see something with more detail on where the plots were positioned or plot layout diagram or photos. The two pictures of the bags fail to show plots. Ex/ How big is the area where the 90 plots (per treatment) were laid out (also not shown on the map)?

Excel data file doesn't appear to include information on bulk density, soil temperature, and moisture.

References

- Bloom, A. J., F. S. Chapin, and H. A. Mooney. 1985. Resource limitation in plants--An economic analogy. *Annual Review of Ecology and Systematics* 16:363-392.
- Du, E., C. Terrer, A. F. A. Pellegrini, A. Ahlström, C. J. van Lissa, X. Zhao, N. Xia, et al. 2020. Global patterns of terrestrial nitrogen and phosphorus limitation. *Nature Geoscience* 13:221-226.
- Fay, P. A., S. M. Prober, W. S. Harpole, J. M. H. Knops, J. D. Bakker, E. T. Borer, E. M. Lind, et al. 2015. Grassland productivity limited by multiple nutrients. *Nature Plants* 1:15080.
- Jones, M. P., B. L. Webb, V. D. Jolley, B. G. Hopkins, and D. A. Cook. 2013a. Evaluating Nutrient Availability in Semi-arid Soils with Resin Capsules and Conventional Soil Tests, I: Native Plant Bioavailability under Glasshouse Conditions. *Communications in Soil Science and Plant Analysis* 44:971-986.
- Jones, M. P., B. L. Webb, V. D. Jolley, M. D. Vickery, R. L. Buck, and B. G. Hopkins. 2013b. Evaluating nutrient availability in semi-arid soils with resin capsules and conventional soil tests, II: Field studies. *Communications in Soil Science and Plant Analysis* 44:1764-1775.
- Jungk, A. O. 2002. Dynamics of nutrient movement at the soil-root interface. Pages the hidden half (Ed.3): 587-616 in Y. Waisel, A. Eshel, and U. Kafkafi, editors. *Plant Roots: The hidden half*. Marcel Dekker, Inc., New York, USA.
- Raven, J. A., H. Lambers, S. E. Smith, and M. Westoby. 2018. Costs of acquiring phosphorus by vascular land plants: Patterns and implications for plant coexistence. *New Phytologist* 217:1420-1427.
- Reinhart, K. O., L. T. Vermeire, C. J. Penn, and Y. Lekberg. 2024. Experimental evidence that poor soil phosphorus (P) solubility typical of drylands due to calcium co-precipitation favors autonomous plant P acquisition over collaboration with mycorrhizal fungi. *Soil Biology and Biochemistry* 199:109605.
- Ryan, M. H., M. Tibbett, T. Edmonds-Tibbett, L. D. B. Suriyagoda, H. Lambers, G. R. Cawthray, and J. Pang. 2012. Carbon

trading for phosphorus gain: the balance between rhizosphere carboxylates and arbuscular mycorrhizal symbiosis in plant phosphorus acquisition. *Plant, Cell & Environment* 35:2170-2180.

Version 1:

Reviewer comments:

Reviewer #2

(Remarks to the Author)

Based on the authors' response and the revised manuscript, I note their diligent efforts in addressing the revisions. The authors have supplemented substantial new data and literature, while further exploring the novelty of their original work, thereby strengthening the study's persuasiveness and highlighting its innovative contributions. All concerns I previously raised have been properly addressed. I have no further issues regarding this manuscript and therefore recommend it for publication.

Reviewer #4

(Remarks to the Author)

This is the second time that I have reviewed the paper. I am satisfied with the changes made by the authors. Nice experiment and interesting results that help to improve our understanding of nutrient cycling and succession in undeveloped (& probably little developed soils)!

There are a few editorial suggestions to help improve your paper.

L236 "CK plots" why not just write "control plots"?

L238 "stimulating" to "stimulated"

L250 nice job!

L258 "to produce enzymes" to "production of enzymes"

L263 "availability;" to "availability."

L279 "with in turn" to "which in turn"

L298 Good point!

L325 Good point! You might want to add a citation. This is one that I can easily think of (Craine, Morrow & Stock 2008) but there might be better.

L372, 378- Did you mean to indent?

L377 extra spaces

L738 Nice conceptual figure.

References

Craine, J.M., Morrow, C. & Stock, W.D. (2008) Nutrient concentration ratios and co-limitation in South African grasslands. *New Phytologist*, 179, 829-836.

Response to Referees Letter

Dear Referees:

We have studied the valuable comments from you carefully, and tried our best to revise the manuscript. In response to the valuable suggestions from you, we have undertaken systematic revisions to the manuscript. In the introduction section, through a comprehensive review of research advances in the field, we identified significant discrepancies in existing literature regarding the conclusions of nitrogen-phosphorus interactions. Additionally, the long-neglected driving role of carbon in nitrogen and phosphorus cycling processes was highlighted. Based on these findings, we propose integrating carbon-related processes into the theoretical framework of nitrogen-phosphorus coupling, which is critical for elucidating the environmental-dependent response mechanisms of nitrogen-phosphorus interactions and bridging discrepancies in research conclusions across different studies.

At the experimental data level, we have supplemented thousands of key data points as recommended by reviewers, specifically including: 1) measurements of total organic carbon (TOC) content and *cbbL* gene mRNA transcription abundance in 450 soil samples; 2) experimental data from treatments with simultaneous nitrogen and phosphorus additions; 3) detailed results of physicochemical property tests for natural soils; and 4) observed data on pH dynamics in nitrification experiments with nitrogen additions. These newly added data provide more robust empirical support for our research conclusions.

We believe that these revisions—through the innovative construction of the theoretical framework and the refinement of the data system—not only strengthen the scientific rigor of our conclusions but also make a substantial contribution to the development of nitrogen-phosphorus coupling theory.

The point to point replies to the referees' comments are listed as following:

Reply to Reviewer #2

➤ Comment 1:

The work is interesting, and the manuscript is well written and structured. Overall, this is an excellent study that provides significant value in enhancing our understanding of the coupling mechanisms of nitrogen and phosphorus in soils. Firstly, I greatly appreciate the research perspective presented in this manuscript. Rather than focusing on mature ecosystems as is common in previous studies of N and P cycling mechanisms, this study explores these mechanisms from the vantage point of early soils, which are the origins of mature ecosystems and involve relatively simpler processes. This approach is particularly advantageous for elucidating the mechanisms that are obscured by complex interactions in more developed systems. For instance, the experiments conducted in this study reveal that the enrichment of one nutrient in the soil can enhance the availability of other nutrients, a finding that aligns well with my expectations and starkly contrasts with the traditional view that the abundance of one nutrient would lead to limiting effects on others. Therefore, I find this research to be highly innovative. Secondly, for a field experimental study, the volume of data utilized in this research is remarkably large, even comparable to that used in some meta-analyses. I am confident that the reliability of the data analysis results from this study surpasses that of many similar studies.

Reply:

We sincerely appreciate the insightful comments and valuable suggestions provided by you. Your recognition of the novelty and methodological rigor of our work is particularly encouraging. We have carefully addressed all points raised and believe these revisions have significantly enhanced the manuscript.

➤ Comment 2:

Line 10: "N and P" should be used as abbreviations for nitrogen and phosphorus.

Reply:

Based on your comments, we have corrected these irregular words. Moreover, by incorporating the suggestions of other reviewers, we have rewritten the abstract of this manuscript.(see line 30-51 in Manuscript text (Revised Document with Track Changes))

➤ Comment 3:

Line 40-42: "In soil environments, the bioavailability of N and P is governed by a suite of intricate soil processes, including N fixation, phosphate dissolution, organic matter mineralization, and biological uptake." These processes should also include soil leaching.

Reply:

In accordance with your comments and by incorporating the opinions of other reviewers, we have rewritten the introduction of this manuscript. (see line 146-250 in Manuscript text (Revised Document with Track Changes))

➤ Comment 4:

Line 43: "...the impacts of N or P availability..." The word "or" fits better in this context than "and".

Reply:

Based on your comments, we have corrected these irregular words. Additionally, taking into account the comments of other reviewers, we have rewritten the introduction of this manuscript. (see line 146-250 in Manuscript text (Revised Document with Track Changes))

➤ Comment 5:

Line 61: please provide a reference for "These processes are influenced by N availability".

Reply:

According to your comments, I have added the relevant references. Moreover, through your comments and the opinions of other reviewers(see line 168-204 in Manuscript text (Revised Document with Track Changes)), we realized that the introduction of the manuscript needed a thorough improvement, so we have rewritten the introduction of this manuscript. (see line 146-250)

➤ Comment 6:

Line 63: please provide a reference for "...as the N element is an important component of phosphatase proteins.

Reply:

In the revised manuscript, we have provided the references that support this argument (see

line 176-179 in Manuscript text (Revised Document with Track Changes)). This reference indicates that the nitrogen content in the phosphatase molecule is between 8% and 32%, which is sufficient to demonstrate that "the N element is an important component of phosphatase proteins". The references supporting this argument are as follows:

Treseder, K.K., Vitousek, P.M., 2001. Effects of soil nutrient availability on investment in acquisition of N and P in Hawaiian rain forests. *Ecology* 82, 946–954.

➤ **Comment 7:**

Line67, 74: nitrogen > N

Reply:

Based on your comments, we have corrected these irregular words.

➤ **Comment 8:**

Line64-76: In fact, these meta-analyses with large datasets also highlight the importance of case studies that utilize large data volumes.

Reply:

Yes, we highly agree with your comment. The viewpoint of this comment is also one of the bases for our research work to conduct experiments with a large sample size.

➤ **Comment 9:**

Line112-115: The authors have not fully elucidated why 'the nutrient limitation in early soil ecosystems is distinctive.

Reply:

The elaboration on this has been added to the revised manuscript text(see line 236-243 in Manuscript text (Revised Document with Track Changes)), as follows:

Traditional theory(Marklein and Houlton, 2012) posits that organisms are expected to allocate their resource reserves toward strategies that increase the acquisition of the most limiting resources — thus moving toward a state where all resources simultaneously limit productivity and growth. The observed N-P co-limitation patterns in developed ecosystems appear consistent with this framework. However, this paradigm fails to explain the successional trajectory of nascent soil systems: Despite an initial severe lack of available nutrients, these systems progressively evolve into productive forests(Zhou et al., 2013; Buma et al., 2017). This evolution suggests the presence of positive feedback mechanisms, indicating a directional development that is incompatible with static co-limitation models.

Based on your comments and in combination with the opinions of other reviewers, we have improved the writing idea of this introduction and rewritten this part of the manuscript.

References:

- Buma, B., Bisbing, S., Krapek, J., Wright, G., 2017. A foundation of ecology rediscovered: 100 years of succession on the William S. Cooper plots in Glacier Bay, Alaska. *Ecology* 98, 1513–1523. doi:10.1002/ecy.1848
- Marklein, A.R., Houlton, B.Z., 2012. Nitrogen inputs accelerate phosphorus cycling rates across a wide variety of terrestrial ecosystems. *New Phytologist* 193, 696–704. doi:10.1111/j.1469-8137.2011.03967.x
- Zhou, J., Wu, Y., Priezel, J., Bing, H., Yu, D., Sun, S., Luo, J., Sun, H., 2013. Changes of soil phosphorus speciation along a 120-year soil chronosequence in the Hailuoguo Glacier retreat area (Gongga Mountain, SW China). *Geoderma* 195–196, 251–259. doi:https://doi.org/10.1016/j.geoderma.2012.12.010

➤ **Comment 10:**

Line 353: please specify why 10 g m⁻² year⁻¹ was adopted?

Reply:

The reason for using this addition amount in the experiment is as follows: Taking into account the high permeability of the soil in the early stage and referring to the amounts of N and P added in the research conducted by Zhao et al. (2014),; Li et al. (2015),; and Tian et al.(2016).

References:

- Li, J., Li, Z., Wang, F., Zou, B., Chen, Y., Zhao, J., Mo, Q., Li, Y., Li, X., Xia, H., 2015. Effects of nitrogen and phosphorus addition on soil microbial community in a secondary tropical forest of China. *BIOLOGY AND FERTILITY OF SOILS* 51, 207–215. doi:10.1007/s00374-014-0964-1
- Tian, J., Wei, K., Condron, L.M., Chen, Z., Xu, Z., Chen, L., 2016. Impact of land use and nutrient addition on phosphatase activities and their relationships with organic phosphorus turnover in semi-arid grassland soils. *Biology and Fertility of Soils* 52, 675–683. doi:10.1007/s00374-016-1110-z
- Zhao, J., Wang, F., Li, J., Zou, B., Wang, X., Li, Z., Fu, S., 2014. Effects of experimental nitrogen and/or phosphorus additions on soil nematode communities in a secondary tropical forest. *SOIL BIOLOGY & BIOCHEMISTRY* 75, 1–10. doi:10.1016/j.soilbio.2014.03.019

➤ **Comment 11:**

Line131: AP? Write out the full name.

Reply:

AP is the abbreviation of "available phosphorus". We have supplemented the full name of AP (see line 259 in Manuscript text (Revised Document with Track Changes)).

➤ **Comment 12:**

Line125-135: The author provides a description of the soil physicochemical properties under natural conditions for the early soils in the study area, which is excellent. However, I believe that information on phosphorus minerals in the soil of the study area (such as apatite) should also be provided. This information is crucial for extending the research findings to mature ecosystems.

Reply:

We would like to express our sincere gratitude for your guidance. Based on your instructions, we have incorporated information on phosphorus minerals in the research area's soil into the manuscript (see line 503-504 in Manuscript text (Revised Document with Track Changes)), which are detailed as follows: According to previous studies conducted by our team, the dominant phosphorus mineral in the study area is apatite, accounting for approximately 2% of the total soil mineral mass(Zhou et al., 2016, 2018).

References:

- Zhou, J., Bing, H., Wu, Y., Sun, H., Wang, J., 2018. Weathering of primary mineral phosphate in the early stages of ecosystem development in the Hailuoguo Glacier foreland chronosequence. *European Journal of Soil Science* 69, 450–461. doi:10.1111/ejss.12536
- Zhou, J., Bing, H., Wu, Y., Yang, Z., Wang, J., Sun, H., Luo, J., Liang, J., 2016. Rapid weathering processes of a 120-year-old chronosequence in the Hailuoguo Glacier foreland, Mt. Gongga, SW China. *GEODERMA* 267, 78–91. doi:10.1016/j.geoderma.2015.12.024

➤ **Comment 13:**

Line136: “Highly significant?” Other sections also contain similar phrasing. I believe it is not appropriate to describe the statistical p-values of research results from highly variable environments using binary terms such as 'significant' or 'not significant.' This approach can mislead readers about the precision of the study results. I recommend adopting the 'Evidence language' method proposed by Stefanie Muff et al., 2022.

Reply:

We sincerely appreciate your suggestions. Following your advice, we have reformulated the degree of difference between the datasets according to the method proposed by Stefanie Muff et al.(2022). (see line 265-267 in Manuscript text (Revised Document with Track Changes))

References:

Muff, S., Nilsen, E.B., O’Hara, R.B., Nater, C.R., 2022. Rewriting results sections in the language of evidence. *Trends in Ecology & Evolution* 37, 203–210. doi:10.1016/j.tree.2021.10.009

➤ **Comment 14:**

Line140-141: In this study, each experimental treatment includes 90 replicates, which is quite remarkable for a field experiment. To justify the necessity of such a large number of replicates, I recommend demonstrating the variability of the data in the supplementary materials. This would provide readers with a reference for evaluating the credibility of previous field experiments, as many typically do not exceed 10 replicates. I believe this approach would greatly assist readers in assessing the reliability of past studies.

Reply:

Thank you for your appreciation of our experimental method. We also believe that this experiment will greatly assist readers in evaluating the reliability of previous studies in terms of sample size. According to your suggestion, we have included the coefficient of variation (CV) for the dataset in the supplementary materials (see Table S8 the statistical characteristics of the dataset.xlsx).

➤ **Comment 15:**

Line153: DOC? Write out the full name

Line192:LMWOA? Write out the full name

Reply:

We have supplemented these full name (see line 286, line 279 in Manuscript text (Revised Document with Track Changes)).

➤ **Comment 16:**

Line194: The Discussion section could be organized into several subsections with headings for clarity. For example, lines 195-236 could be headed 'Mechanism Underlying Nitrogen-Driven Phosphorus Availability in Early-Stage Soils'. Additionally, in the discussion, I recommend extending the discussion to speculate on the mechanisms related to nitrogen and phosphorus in mature soils (or vegetation-dominated soils) based on your data results. This would significantly enhance the depth of the discussion in this manuscript.

Reply:

Following your advice, we have organized the Discussion section into several subsections

with headings. In fact, we have extended the discussion to speculate on the mechanisms related to nitrogen and phosphorus in mature soils (see line 339-349, line 382-398, line 460-467 in Manuscript text (Revised Document with Track Changes)).

➤ **Comment 17:**

Line223:CK? Write out the full name

Line 240, 290, 296: mechanism > mechanisms; nitrogen and phosphorus > N and P; nitrogen addition > N addition.

Reply:

Based on your comments, we have corrected these irregular words.

➤ **Comment 18:**

Line314-316: The content of this sentence should be moved to the Discussion section. It should correspond with the description of phosphorus minerals in the Supplementary Materials.

Reply:

This sentence have been moved to the Discussion section.

➤ **Comment 19:**

Line350-356: Present the experimental design in tabular form and include it in the Supplementary Materials.

Reply:

Following your advice, we have showed the experimental design in tabular form and include it in the Supplementary Materials (see Table S1 in Supplementary materials).

Reply to Reviewer #3:

➤ Comment 1:

1. Lines 125-135 and Table 1: This section describes the natural state of the early soils. However, key details are missing, such as when and how the soils were sampled, the number of replicates, and whether the sampling occurred before, during, or after the three-year experiment. It is also unclear whether the natural state of the soils remained consistent over the three-year period without N and P addition.

Reply:

Thank you for pointing out the issue of insufficient detail in the description of sample information in our manuscript. According to your comments, we have supplemented the key details you pointed out in the "Methods" section of the manuscript (see line 572-582 in Manuscript text (Revised Document with Track Changes)). The added content is as follows:

In summary, the above-mentioned in-situ experimental method ensured that in July 2022, three years later, we collected a sufficient number of soil samples (i.e., 90 replicates for N addition treatments, 90 replicates for P addition treatments, and 90 replicates for CK treatments, totaling 270 soil samples). The in-situ experiment was concluded in July 2022, and soil columns (0 - 5 cm) in each plot were collected as soil samples. The collection method was as follows: For each plot, the tubular polyethylene bag enclosing the intact soil core was excavated as a whole, and the soil at a depth of 0 - 5 cm of this core was collected and placed into a sterile sample bag as one soil sample. Altogether, we collected 270 fresh soil samples. Each of them was sieved through a 2-mm mesh before being divided into two subsamples. One subsample was stored at 4 °C for the measurement of soil physicochemical characteristics, and the other subsample was stored at -80 °C for genes analysis.

The above supplementary content indicates the following: the time of soil sampling was June 2022; the method of sampling was to excavate the tube-shaped polyethylene bag enclosing the intact soil core for each plot, and collect soil from the 0-5 cm depth of this core into a sterile sample bag as one soil sample; the number of replicates was 90 for each treatment; and the sampling occurred three years after the experiment began, i.e., in June 2022.

Regarding the comment, "It is also unclear whether the natural state of the soils remained consistent over the three-year period without N and P addition," our response is as follows: In fact, we also collected natural soil samples (soils without any treatment, with 90 replicates) corresponding to the experimental treatments. However, these natural soil samples showed no significant differences in various metrics compared to the CK treatment samples. Considering that the CK treatment samples more effectively exclude the influence of other factors and more convincingly demonstrate the effects of the experimental treatments, and to reduce data redundancy in the manuscript for a more focused presentation of the research results, we did not include the natural soil data in the main text. Nevertheless, to address your concern and fully substantiate our findings, the natural soil data have been added to the supplementary materials (see Table S6 source data for natural soil.xlsx and Figure S2).

➤ Comment 2:

2. Lines 153-156 and Figure 2B: The cbbL gene copy number showed only minor changes, with slight differences between N and P additions, despite statistical significance. However, cbbL

gene copy number does not always correlate with C fixation activity. Direct measurements of C fixation rates or qPCR analysis of *cbbL* RNA transcripts would provide more robust evidence. Given the large variation in this study, small changes in *cbbL* gene abundance, and the lack of C fixation rate measurements, the conclusion that N and P additions increased C fixation may not be fully supported.

Reply:

Thank you for your valuable feedback on our research. We understand your perspective regarding the magnitude of changes in the *cbbL* gene copy numbers and have conducted an in-depth analysis accordingly. Indeed, according to our experimental data, the average *cbbL* gene copy number increased by 90.1% under N addition treatments and by 48.2% under P addition treatments compared to the CK treatment. Such increases are statistically significant and represent relatively large changes from both biological and large sample size (90 samples per treatment) perspectives, especially concerning soil microbial activity.

However, we also acknowledge that the actual environmental impacts and ecological significance of these increases may require more direct evidence for support, such as measurements of carbon fixation rates or qPCR analysis of *cbbL* RNA transcripts. We agree with your viewpoint that changes in gene copy numbers alone are insufficient to fully assess their potential impact on carbon fixation. To more accurately reflect the importance of this finding, we will further discuss the implications of these data and conduct additional experiments, namely measuring TOC and performing qPCR analysis of *cbbL* RNA transcripts (see Figure 2B, Table S3 source data for N addition treatment, Table S4 source data for CK treatment, Table S5 source data for P addition treatment), to confirm the significance of these changes.

➤ **Comment 3:**

Organic Carbon Data: Figure 2 and many places in the manuscript present only dissolved organic carbon (DOC) data, excluding total organic carbon (TOC). Since DOC represents only a fraction of the organic carbon fixed by microorganisms, TOC is a critical parameter. Without both TOC and DOC data, it is difficult to draw right conclusions.

Reply:

Thank you for your valuable feedback on our research. In accordance with your comments, we have measured the soil total organic carbon (TOC) and included the results in the revised manuscript(see Figure 2B, Table S3 source data for N addition treatment, Table S4 source data for CK treatment, Table S5 source data for P addition treatment).

➤ **Comment 4:**

4. Lines 157-167: Nitrogenase activity is a more reliable indicator of N₂ fixation than *nifH* gene copy number. It is well known that many bacteria/proteobacteria, possess the *nifH* gene but do not actively fix N₂. Some lack the complete set of genes required for nitrogenase function, while others, despite having the full gene set, exhibit no detectable N₂ fixation activity. I would suggest that gene abundance, such *cbbL* and *nifH* gene, may not be good indicators in this particular study.

Reply:

Thank you for your valuable comments and insightful suggestions on our paper. We fully agree with your viewpoint that nitrogenase activity is a more reliable indicator of nitrogen fixation

compared to the *nifH* gene copy number. We also concur with your opinion that not all bacteria/proteobacteria possessing the *nifH* gene can effectively carry out nitrogen fixation, which may be due to either lacking the complete set of genes required for nitrogenase function or showing no detectable nitrogen fixation activity despite having the full gene set. Indeed, our data (Figure 2C)—showing no significant changes in nitrogenase activity or *nifH* gene abundance under nitrogen addition treatments compared to the CK treatment—further support this viewpoint.

Regarding your suggestions, we would like to provide some clarification: In the research design of this study, we did adopt a multi-level assessment system for nitrogen fixation activity. In addition to analyzing the abundance of the *nifH* gene, we characterized nitrogen fixation activity through direct indicators such as ammonium nitrogen concentration and nitrogenase activity.

The combined use of these indicators ensures the reliability of our research findings. We acknowledge the limitations associated with *nifH* gene abundance, and thus, in our data analysis, we mainly used it as a supplementary reference indicator while focusing on the results obtained from direct measurements. We will further emphasize this point in the revised manuscript (see line 292-295 in Manuscript text (Revised Document with Track Changes)), clearly stating that Our study does not merely rely on the abundance data of the nitrogenase gene (*nifH*), but also makes use of ammonium nitrogen ($\text{NH}_4^+\text{-N}$) and nitrogenase activity to more accurately reflect the characteristics of the actual nitrogen fixation process (Fig. 2 C). Thank you again for your suggestions, which will help us improve the quality of our paper..

➤ **Comment 5:**

5. Lines 168-170, 219-220: The data show a decrease in pH alongside increased nitrate concentration and amoA gene abundance. To conclude that nitrification significantly contributed to the pH decrease in ammonium addition groups, additional experiments are needed to quantify the pH change due to nitrate generation from ammonium via nitrification in this study.

Reply:

In another manuscript of ours focusing on the soil nitrification processes in this study area (not yet published), we have already quantified the changes in soil pH resulting from the conversion of ammonium to nitrate via nitrification through incubation experiments. Please refer to the supplementary material (“Nitrification experiment” section) of the revised manuscript for details.

➤ **Comment 6:**

6. Lines 174-193: The two models presented are somewhat confusing due to numerous negative correlations. Ammonium-N in the N addition group represents residual ammonium in the soil, and available P (AP) represents residual P. Including these residual concentrations in the models may not be appropriate and could complicate interpretation.

Reply:

We are extremely grateful that you have carefully reviewed our paper in the midst of your busy schedule and put forward highly valuable comments. We attach great importance to the two key issues you pointed out: the confusion in understanding caused by the numerous negative correlations in the structural equation model, and the possible complication of interpretation due to the inclusion of the “residual” concentrations of ammonium-N and available phosphorus (AP) in the model. Here, we would like to provide you with a detailed and in - depth explanation.

First of all, regarding the numerous negative correlations in the model, we fully understand your doubts. In the general perception, if the absorption and release of available nutrient elements (e.g. ammonium-N and AP) in soil processes reach a dynamic equilibrium state, the relationships in the model are likely to be positive correlations. However, our research system has its uniqueness: From a data perspective, the results of sample analysis clearly show that the concentration of the added elements in the addition treatment is much higher than that in the control (CK) treatment (Figure 2A, C), indicating that the absorption and release of the added elements (ammonium-N or AP) by the soil processes have not reached a dynamic equilibrium. In terms of the experimental method, we adopt the approach of periodically adding equal amounts of nutrient elements to each replicate plot, which makes it highly unlikely for the soil processes in the plots to reach the above - mentioned dynamic equilibrium state. Since equal amounts of corresponding nutrient elements are added to the corresponding plots, and there is heterogeneity among the replicate plots (the reason why we set 90 replicates for each experimental treatment is to make full use of this heterogeneity), the rates at which the soil processes in each plot absorb and assimilate the added nutrient elements to produce "functional components" (such as the *cbbL* gene, nitrogenase, etc.) cannot be the same. Specifically, the soil in the plots with faster assimilation will produce more "functional components", and correspondingly, the remaining nutrient elements will be less; while the soil in the plots with slower assimilation will produce fewer "functional components", and the remaining nutrient elements will be relatively more. This difference is ultimately manifested as a negative correlation in the structural equation model we constructed, which is actually a scientific reflection of the real experimental situation. Therefore, in the context of our experiment, these negative correlations do not imply an inhibitory relationship between the indicators.

In addition, if we measure the relevant indicators after the soil processes in our experiment reach a dynamic equilibrium state, although the positive correlations presented between the indicators seem intuitive, when analyzing the driving effects of available nutrients on soil processes, it will become more complex and difficult to distinguish. Because a positive correlation means that the indicators drive each other, and it is very difficult to clearly identify which one drives the other under the background of dynamic equilibrium, which is extremely unfavorable for our in - depth exploration of the nitrogen - phosphorus coupling mechanism in the soil and the driving effects of available nutrients.

Secondly, regarding the issue of including the "residual" concentrations of ammonium nitrogen and available phosphorus in the model. In the field of soil ecosystem research, these "residual" concentrations are key indicators for measuring the soil nutrient status, and they intuitively reflect the nutrient content in the soil that is actually available for organisms to absorb and utilize and participate in soil biochemical reactions. In our research, the "residual" ammonium nitrogen and available phosphorus in the soil are not only the result manifestations after the experimental treatment but also important driving factors for subsequent soil processes. For example, the structural equation model based on the phosphorus - addition experiment shows that the ammonium nitrogen concentration is mainly directly affected by nitrogenase activity, indicating that the "residual" ammonium nitrogen in the soil is closely related to the nitrogen - fixation process in the soil; while the structural equation model based on the nitrogen - addition experiment shows that the available phosphorus concentration is mainly directly affected by pH and low-molecular-weight organic acids (LMWOA), which reflects the internal connection

between the “residual” available phosphorus in the soil and the soil chemical properties. Therefore, including them in the model helps us to more comprehensively and accurately reveal the coupling pathways between nitrogen and phosphorus in the soil and the driving mechanisms of soil processes.

We always regard the scientific nature and rigor of the paper as of utmost importance. Once again, thank you for your attention and suggestions. Your comments have been of great help to us in further improving the scientific nature and readability of the paper. We sincerely look forward to your further guidance so that we can present our research results to readers more accurately and clearly.

➤ **Comment 7:**

7. Lines 227-230: The manuscript discusses organic acids but does not quantify their contribution to pH reduction.

Reply:

According to your comment, we quantified the contributions of low molecular weight organic acids (LMWOA) and the ammonia - oxidation process (measured by NO_3^- -N, the product of the ammonia - oxidation process) to pH using partial Redundancy Analysis (partial RDA) based on the N addition treatment. The specific methods are as follows: taking NO_3^- -N as a control variable, we calculated the amount of pH variation explained by LMWOA. Similarly, taking LMWOA as a control variable, we calculated the amount of pH variation explained by NO_3^- -N. The results are shown in the following figure: the total amount of pH variation explained by LMWOA is 69% (13.4% + 55.6%), among which the net amount of pH variation explained by LMWOA is 13.4% (in the figure, L represents LMWOA, and N represents NO_3^- -N). These results have been added to the revised manuscript. Please refer to Fig. S4 in Supplementary Materials.

➤ **Comment 8:**

8. Lines 257-258: The claim of “a large number of C fixation genes” lacks comparative data, making it difficult to assess the validity of this statement.

Reply:

Thank you very much for your insightful comment. We greatly appreciate your attention to the details of our manuscript and the valuable feedback you have provided. You are absolutely right that the claim of “a large number of C fixation genes” in the original statement lacked comparative data, which could make it difficult for readers to assess the validity of this statement.

Consequently, we have revised the statement to provide more specific and comparable data (see line 401-402 in Manuscript text (Revised Document with Track Changes)). The revised statement is as follows: Additionally, a considerable amount of the *cbbL* gene involved in carbon fixation from microorganisms (3.4×10^4 – 1.1×10^6 copies/g in Fig. 2B, compared to the study by Tahon et al.) was detected in our soil samples. We believe that this revision provides a clearer and more accurate description of the findings in our study. Once again, thank you for your valuable comment, which has helped us improve the quality of our manuscript.

References:

Tahon et al.. Analysis of *cbbL*, *nifH*, and *pufLM* in Soils from the Sør Rondane Mountains, Antarctica, Reveals a Large Diversity of Autotrophic and Phototrophic Bacteria

➤ **Comment 9:**

9. Lines 261-262: The manuscript incorrectly states that C fixation provides energy for N₂ fixation. Photosynthesis converts light energy into chemical energy, which can support cellular processes, including N₂ fixation. However, C fixation itself refers to the conversion of inorganic carbon (e.g., CO₂) to organic carbon. This distinction should be clarified.

Reply:

We sincerely thank you for your valuable comments on the manuscript. The issue you raised is critically important, and we fully acknowledge that the original wording lacked precision. We sincerely apologize for the ambiguity in describing the essential nature of energy sources, which may have led to misinterpretation. To ensure scientific accuracy, we have revised the relevant section as follows:

Original statement:

"Another important reason for this opinion is that the N fixation process requires a large amount of energy, which mainly comes from C fixation processes such as photosynthesis."

Revised statement:

"Another important reason supporting this perspective is that N fixation requires substantial energy input. As demonstrated in previous studies(Dixon and Kahn, 2004; Wang et al., 2021; Zheng et al., 2020), organic carbon serves as a crucial energy source for nitrogen fixation processes, and this organic carbon is primarily derived from carbon fixation pathways, such as photosynthesis."

Rationale for revision:

According to previous research(Dixon and Kahn, 2004; Zheng et al., 2020; Wang et al., 2021), organic carbon serves as a crucial energy source for N fixation processes. For instance, Wang et al.(2021)proposed that "C availability alone may be a major predictor of free-living N fixation (FLNF) rates." This is because, unlike symbiotic organisms, heterotrophic diazotrophs must independently acquire energy, typically by oxidizing organic molecules released during organic matter decomposition or excreted by other organisms, to sustain the energy-intensive process of N fixation¹. Additionally, Dixon and Kahn(2004) showed that the N status-sensing PII proteins and the *nif*-gene regulation systems of some diazotrophs are coregulated by C and N signals (e.g., 2-oxoglutarate and glutamine) and that N signals override C signals to depress N fixation only in situations of N excess. Conversely, this finding implies that N fixation is promoted when C signals outweigh N signals. This conclusion is further supported by the work of Zheng et al. (2020), whose research demonstrated "increased FLNF rates in N-rich late-successional ecosystems compared with relatively N-poor early-successional ecosystems," and identified the C:N ratio as a key factor in explaining variations in FLNF rates.

Thank you once again for your meticulous attention to detail, which has significantly strengthened the clarity and rigor of our work.

References:

- Dixon, R., Kahn, D., 2004. Genetic regulation of biological nitrogen fixation. *Nature Reviews Microbiology* 2, 621–631. doi:10.1038/nrmicro954
- Wang, J., Wu, Y., Li, J., He, Q., Zhu, H., Bing, H., 2021. Energetic supply regulates heterotrophic nitrogen fixation along a glacial chronosequence. *Soil Biology and Biochemistry* 154, 108150. doi:10.1016/j.soilbio.2021.108150

Zheng, M., Chen, H., Li, D., Luo, Y., Mo, J., 2020. Substrate stoichiometry determines nitrogen fixation throughout succession in southern Chinese forests. *Ecology Letters* 23, 336–347. doi:10.1111/ele.13437

➤ **Comment 10:**

10. Lines 267-276: The proposed link between AP and available N via C fixation may not be valid. As noted earlier, C fixation does not provide energy for N₂ fixation, and the evidence for increased C fixation is weak (see comments on *cbbL* gene copy numbers and TOC). Additionally, P addition might promote N₂-fixing microorganisms that do not depend on available N, while other microorganisms remain N-limited. It should be noted that some N₂-fixing microorganisms, such as cyanobacteria, can perform both photosynthesis and N₂ fixation. Therefore, AP and directly contributes to N₂ fixation and C fixation may not be the link between AP and N₂ fixation.

Reply:

We sincerely apologize for the misunderstandings caused to you and readers by our oversimplified expressions, insufficient data, and previous inaccurate description of the relationship between carbon fixation and energy supply (which has been revised in our response to Comment 9). To rectify these errors and enhance the scientific rigor of our research, we have supplemented TOC measurements and *cbbL* RNA reverse-transcribed qPCR data (as detailed in our responses to Comments 2 and 3). Accordingly, we have rewritten and modified the content from Lines 417-432 (in Manuscript text (Revised Document with Track Changes)) as follows:

To elucidate the mechanism by which P drives N availability in early-stage soils, we constructed a structural equation model (SEM) (Fig. 3B). In this model, whereas no significant relationship was observed between available phosphorus (AP) and *nifH* gene abundance, several significant pathways (such as the associations between AP and nitrogenase activity, AP and DOC, DOC and the *nifH* gene, DOC and nitrogenase activity) combined with the findings that AP addition treatment (Fig. 2B) significantly increased DOC concentration, TOC concentration, and *cbbL* RNA reverse-transcribed copy numbers, collectively indicate that organic C derived from C fixation processes serves as the critical link between AP and N availability. Besides, these significant pathways also indicate that AP may preferentially regulate nitrogenase activity and energy acquisition (i.e., organic C production through C fixation) rather than simply increasing *nifH* gene copy numbers. Moreover, previous studies (Dixon and Kahn, 2004; Wang et al., 2021; Zheng et al., 2020) have established organic carbon as a crucial energy source for N fixation. Therefore, our findings emphasize that AP in early-stage soils enhances N fixation by stimulating organic C production through C fixation processes, thereby promoting N availability. The results and analyses above indicate that integrating C-related processes into the interaction framework of the N-P cycle plays a key role in refining the existing N-P coupling theory.

If our current revisions still fail to meet your requirements, we sincerely hope that you can provide us with further feedback and guidance.

References:

- Wang, J. et al. Energetic supply regulates heterotrophic nitrogen fixation along a glacial chronosequence. *Soil Biology and Biochemistry* 154, 108150 (2021)..
- Dixon, R. & Kahn, D. Genetic regulation of biological nitrogen fixation. *Nature Reviews Microbiology* 2, 621–631 (2004).

Zheng, M., Chen, H., Li, D., Luo, Y. & Mo, J. Substrate stoichiometry determines nitrogen fixation throughout succession in southern Chinese forests. *Ecology Letters* 23, 336–347 (2020).

Reply to Reviewer #4:

➤ Comment 1:

Sun et al. perform a nutrient addition experiment (N,P) in early seral habitat in an attempt to understand available N & P dynamics. If I understand their experimental design correctly then they tested the effect of three treatments on various soil properties and N cycling functional genes with 30 randomly assigned plots per treatment. Each plot replication contained a soil core incubated in a plastic bag to ensure that additions of nutrients were kept in contact with the soil and didn't leach away. Post-incubation the soil cores were assayed. N&P additions were provided as a solution periodically sprayed onto the top of each bagged soil core. It isn't clear that a similar amount of water was sprayed onto the control cores to control for moisture additions (which might possibly affect organic matter if water limited). The authors probably need to elaborate on how the nutrient addition amounts were carefully controlled with a sprayer (e.g. make and model of sprayer, calibration info. etc.) in place of the more conventional approach of pouring a fixed amounts of solution into bags.

Reply:

Thank you for your comments. Your feedback has helped us recognize that our description of the research methodology for "Field in-situ experiments and soil sample collection" (Lines 329-356) was unclear. For instance, while we allocated 90 plots per treatment as replicates, our original wording inadvertently created the impression that only 30 plots were assigned to each treatment. Additionally, our description lacked critical details you highlighted, such as the brand and model of the sprayer used, calibration protocols, and related specifications. To ensure methodological clarity, we have comprehensively revised this section in accordance with your suggestions as follows (see line 535-582 in Manuscript text (Revised Document with Track Changes)):

We conducted a large-scale, three-year in situ nitrogen (N) and phosphorus (P) addition experiment on early-stage soils in the Hailuoguo glacier retreat area, with over 270 plots established. This experiment is particularly significant (beyond its focus on "early-stage soils") because the early soils in this study area can develop into a soil ecosystem with approximately 25% pioneer plant cover within about six years. Consequently, for the early-stage soil ecosystems in this region—characterized by the absence of vegetation and rapid development—a three-year study period can be regarded as "long-term" study. Thus, our experiment can be defined as a "long-term in situ nitrogen and phosphorus addition experiment targeting early-stage soils."

In May 2019, a substantial number of plots (>270) were established in the study area to conduct the in situ N and P addition experiments. Each plot measured 0.3 m × 0.3 m, with spacing between plots exceeding 4 m. The experiment included three treatments: nitrogen addition (+N), phosphorus addition (+P), and a control (CK). Given the complex microtopography, frequent geological disturbances (e.g., small-scale debris flows and landslides), and significant microenvironmental heterogeneity in the study area, to enhance the accuracy of experimental data and mitigate data loss from potential plot damage caused by natural disasters, we randomly allocated over 90 plots per treatment as replicates (i.e., the total number of plots assigned to the three treatments exceeded 270). This design also ensured the collection of 90 replicate samples per treatment (a total of 270 samples for the three treatments) by the end of the experiment.

For each of the over 270 plots, the experimental setup involved the following steps:

Soil column extraction: Using an undisturbed soil sampler (inner diameter: 10 cm), we extracted an intact soil column (dimensions: 15 cm height × 10 cm diameter, preserving its natural structure) from each plot. The soil column was placed into a tubular polyethylene bag to ensure thorough interaction between subsequently added N (or P) and the soil column, prevent lateral leaching of N(or P) and avoid cross-contamination between plots.

Reinstallation: The soil column was then returned to its original location (Fig. S1).

Nutrient addition protocol:

Given the high permeability of early-stage soils and referencing prior studies, N and P were not applied in a single dose. Instead, the following protocol was implemented:

N addition: Starting in May 2019, N was applied to +N treatment plots at a rate of 10 g N m⁻² year⁻¹. Specifically, NH₄Cl solution (concentration: 10.00 mg/ml) was evenly sprayed onto the soil column surface (area: 78.5 cm²) using a sprayer (brand: Lizhen; capacity: 100 ml; spray volume per press: 0.132 ml). Each application involved 5 ml of solution (achieved via 38 presses), repeated every two months.

P addition: Similarly, +P treatment plots received 10 g P m⁻² year⁻¹ via NaH₂PO₄ solution (concentration: 10.13 mg/ml), applied identically using the same sprayer (5 ml per application, 38 presses).

CK treatment: For CK plots, pure water was sprayed using the same method.

In summary, the above-mentioned in-situ experimental method ensured that in July 2022, three years later, we collected a sufficient number of soil samples (i.e., 90 replicates for N addition treatments, 90 replicates for P addition treatments, and 90 replicates for CK treatments, totaling 270 soil samples). The in-situ experiment was concluded in July 2022, and soil columns (0 - 5 cm) in each plot were collected as soil samples. The collection method was as follows: For each plot, the tubular polyethylene bag enclosing the intact soil core was excavated as a whole, and the soil at a depth of 0 - 5 cm of this core was collected and placed into a sterile sample bag as one soil sample. Altogether, we collected 270 fresh soil samples. Each of them was sieved through a 2-mm mesh before being divided into two subsamples. One subsample was stored at 4 °C for the measurement of soil physicochemical characteristics, and the other subsample was stored at -80 °C for genes analysis.

➤ **Comment 2:**

In the Introduction the authors cite numerous meta-studies on related topics and appear to use these meta-studies as motivation for the current study. This is somewhat confusing. A central problem with the study's presentation is that the authors don't clearly distinguish their study from prior work or adequately highlight the novelty of their study relative to prior work. Their study is likely somewhat novel because they applied nutrients to an area undergoing primary succession unlike most studies but exactly why that is needed or necessary isn't clear. Their study appeared to mainly be motivated by the fact that three similar meta-studies reached divergent conclusions. However, this isn't convincing motivation for doing yet another nutrient addition field experiment. In fact, the reality that there are now multiple meta-studies on the topic is an indication that the topic is well studied. So the motivation for the present study was a bit unclear and was not framed well. Further, this had me questioning why the authors submitted the current study to a journal that expects studies to make important advances within a field. Nutrient addition studies are possibly the most common field manipulation in ecological studies.

Reply:

We sincerely appreciate your constructive feedback. We agree that the original introduction inadequately highlighted the unique rationale for our study. In accordance with your comment, we have revised and rewritten the introduction (see line 146-250 in Manuscript text (Revised Document with Track Changes)). In the revised version, we have:

Clarified the unresolved scientific gap: Emphasized that prior meta-analyses reached conflicting conclusions due to oversimplified single-process interpretations (e.g., focusing solely on phosphatase activity or N fixation without integrating cross-process interactions and energy constraints).

Explicitly justified the experimental context: Highlighted how early-stage primary succession soils (1) minimize confounding biotic/abiotic complexities (e.g., lack of plant dominance, simplified microbial communities), enabling clearer mechanistic insights into N-P coupling, and (2) represent a critical yet understudied phase where traditional stoichiometric theories may behave differently due to extreme nutrient limitation.

Reframed novelty: Stressed that our study uniquely integrates C process and nitrification process into the N-P coupling framework—a dimension overlooked in prior nutrient addition experiments but critical for explaining context-dependent responses.

These revisions aim to demonstrate that our work advances ecological theory by addressing mechanistic ambiguities in a distinct environmental context, rather than merely replicating prior nutrient addition approaches. Especially, despite the presence of multiple meta-analysis studies on this topic of N and P, their inconsistent conclusions indicate that this topic has not been well studied, to the extent that its essential laws have not been delved into deeply.

➤ Comment 3:

Many of the topics presented in the paper on N and especially P cycling were only superficially explored in the Introduction. I expected these topics to be well developed and that they would lead the reader toward a researchable problem. Sentences often lacked citations or mainly cited meta-studies. Overall, their study appeared to be mainly inspired by meta-studies.

Reply:

We sincerely appreciate the reviewer's insightful feedback regarding the development of key concepts in the Introduction. We have substantially revised this section through the following specific improvements:

Revised mechanistic discussion paragraphs (now spanning Lines 185-204 and 215-226 in Manuscript text (Revised Document with Track Changes)) detailing:

- a) The mediation of carbon processes in N-P interactions
- b) The possibility of inhibition of phosphorus activation processes by nitrification processes
- c) The limitations of using a single process indicator for exploring the complete mechanism of N-P coupling processes
- d) The necessity of integrating other soil processes into the N-P coupling framework

Incorporated 16 new citations spanning:

- a) The concept of nitrogen-phosphorus co-limitation and coupling (Du et al., 2020; Zhou et al., 2021, Elser et al., 2007; Čapek et al., 2018, and so on)
- b) Carbon-driven studies of phosphorus availability processes (Spohn and Kuzyakov, 2013;

Heuck et al., 2015)

- c) The acidification effects of nitrogen (Zhou et al., 2014)
- d) Energy sources for nitrogen fixation processes (Davies and Friedlingstein, 2020)

Problem Articulation

Developed a dedicated gap analysis framework through:

- (i) Triangulation of conflicting meta-analysis findings (Lines 179-185 contrasting Marklein and Houlton, 2012 vs. Gou et al., 2024 vs. Chen et al., 2023)
- (ii) Identification of missing mechanistic processes (C-P-N cross-talk in Lines 185-192; energy mediation in 215-226)
- (iii) Context specificity of early soil ecosystem (Lines 230-243 contrasting mature vs. pioneer ecosystems)

Citation Strategy Overhaul

In the introduction, we have supplemented and provided corresponding citations for each sentence, except for those expressing the authors' viewpoints and logical arguments.

We have also increased the proportion of experimental and case study investigations, including:

- a) three studies on nitrogen-phosphorus co-limitation;
- b) five studies on nitrogen-phosphorus processes;
- c) three studies on the linkage of carbon processes with nitrogen-phosphorus processes;
- d) two studies on the energy sources for nitrogen fixation processes.

For detailed modifications, please see the introduction section (Line 146-250 in Manuscript text (Revised Document with Track Changes)) of the revised manuscript.

References:

- Davies, B.T., Friedlingstein, P., 2020. The Global Distribution of Biological Nitrogen Fixation in Terrestrial Natural Ecosystems. *Global Biogeochemical Cycles* 34, e2019GB006387. doi:10.1029/2019GB006387
- Du, E., Terrer, C., Pellegrini, A.F.A., Ahlström, A., Van Lissa, C.J., Zhao, X., Xia, N., Wu, X., Jackson, R.B., 2020. Global patterns of terrestrial nitrogen and phosphorus limitation. *Nature Geoscience* 13, 221–226. doi:10.1038/s41561-019-0530-4
- Chen, Y., Xia, A., Zhang, Z., Wang, F., Chen, J., Hao, Y., Cui, X., 2023. Extracellular enzyme activities response to nitrogen addition in the rhizosphere and bulk soil: A global meta-analysis. *Agriculture, Ecosystems & Environment* 356, 108630. doi:10.1016/j.agee.2023.108630
- Čapek, P., Manzoni, S., Kaštovská, E., Wild, B., Diáková, K., Bárta, J., Schneckner, J., Biasi, C., Martikainen, P.J., Alves, R.J.E., Guggenberger, G., Gentsch, N., Hugelius, G., Palmtag, J., Mikutta, R., Shibistova, O., Urich, T., Schleper, C., Richter, A., Šantrůčková, H., 2018. A plant–microbe interaction framework explaining nutrient effects on primary production. *Nature Ecology & Evolution* 2, 1588–1596. doi:10.1038/s41559-018-0662-8
- Elser, J.J., Bracken, M.E., Cleland, E.E., Gruner, D.S., Harpole, W.S., Hillebrand, H., Ngai, J.T., Seabloom, E.W., Shurin, J.B., Smith, J.E., 2007. Global analysis of nitrogen and phosphorus limitation of primary producers in freshwater, marine and terrestrial ecosystems. *Ecology Letters* 10, 1135–1142.
- Gou, X., Ren, Y., Qin, X., Wei, X., Wang, J., 2024. Global patterns of soil phosphatase responses to nitrogen and phosphorus fertilization. *Pedosphere* 34, 200–210. doi:10.1016/j.pedsph.2023.06.011
- Heuck, C., Weig, A., Spohn, M., 2015. Soil microbial biomass C:N:P stoichiometry and microbial use of organic phosphorus. *Soil Biology and Biochemistry* 85, 119–129. doi:10.1016/j.soilbio.2015.02.029
- Marklein, A.R., Houlton, B.Z., 2012. Nitrogen inputs accelerate phosphorus cycling rates across a wide variety of

terrestrial ecosystems. *New Phytologist* 193, 696–704. doi:10.1111/j.1469-8137.2011.03967.x

Spohn, M., Kuzyakov, Y., 2013. Phosphorus mineralization can be driven by microbial need for carbon. *Soil Biology and Biochemistry* 61, 69–75. doi:10.1016/j.soilbio.2013.02.013

Zhou, J., Li, X., Peng, F., Li, C., Lai, C., You, Q., Xue, X., Wu, Y., Sun, H., Chen, Y., Zhong, H., Lambers, H., 2021. Mobilization of soil phosphate after 8 years of warming is linked to plant phosphorus - acquisition strategies in an alpine meadow on the Qinghai - Tibetan Plateau. *Global Change Biology* 27, 6578 - 6591. doi:10.1111/gcb.15914

Zhou, J., Xia, F., Liu, X., He, Y., Xu, J., Brookes, P.C., 2014. Effects of nitrogen fertilizer on the acidification of two typical acid soils in South China. *Journal of Soils and Sediments* 14, 415–422. doi:10.1007/s11368-013-0695-1

➤ **Comment 4:**

The writing was often vague and topics were not well developed. Below are many suggestions to help improve the writing.

L1 “Insights into the driving effects between nitrogen and phosphorus”- wordy & vague.

Reply:

Based on your feedback and the full revised manuscript, we have updated the title to: "Carbon-mediated co-amplification of N and P mobilization in proglacial soils: Large-scale experimental evidence of nutrient synergy". We believe this revision achieves a standard of conciseness and clarity.

➤ **Comment 5:**

L7 “But, due to the complexity of processes in mature ecosystems, a variety of inconsistent conclusions have been drawn.” – Authors then go on to mention how studying primary successional systems may help to improve understanding of mature ecosystems. This seems like a long shot. I think the authors are trying to say, “look scientists have spent a lot of time studying mature ecosystems and barely understand what is going on. So we are motivated to study these processes during primary succession.” However, it seems more direct to say while mature ecosystems have been extensively studied little is known about X in newly developing ecosystems. [or something similar] I think you are mostly trying to identify a gap in understanding. And primarily citing meta-studies fails to show that problem.

Reply:

We sincerely appreciate your insightful comments and fully agree with your assessment. We apologize for the inadequate articulation of research rationale in our original introduction. Following your suggestions, we have thoroughly restructured the connections between previous studies and our research motivations, while explicitly clarifying the scientific novelty of this work.

The revised introduction now follows this logical framework:

- 1) We first identify the conceptual incompleteness in current understanding of nitrogen-phosphorus (N-P) interaction mechanisms within biogeochemical cycles.
- 2) Through systematic analysis of conflicting conclusions in existing literature (examining both N→P and P→N perspectives), we highlight persistent discrepancies in reported N-P coupling patterns.
- 3) Critical evaluation of methodological variations across studies reveals that failure to integrate ancillary soil processes (particularly carbon dynamics) into conventional N-P

coupling frameworks likely drives these inconsistencies.

- 4) Building on the unique advantages of studying early-stage soil ecosystems - their simplified processes minimize confounding factors while maintaining evolutionary trajectories toward mature systems - we establish our research objective: To decode fundamental N-P interaction mechanisms through carbon-mediated pathways using glacial forefield succession as a model system.

Regarding meta-analysis citations:

We deliberately incorporated meta-analytic approaches because they synthesize numerous localized case studies, providing conclusions with broader generalizability and enhanced credibility compared to individual case studies. This aligns with our goal of establishing universally applicable principles for N-P interaction mechanisms.

For detailed modifications, please see the introduction section of the Manuscript text (Revised Document with Track Changes).

➤ **Comment 6:**

L10 “initial processes of nitrogen and phosphorus and their interactions”- unclear. N & P aren't processes... Also, it isn't clear what you measured or why?.

Reply:

We concur with your assessment. Based on this feedback of yours and other comments, we have rewritten the abstract. Please refer to the abstract in the revised manuscript text.

➤ **Comment 7:**

L12 “N & P interactions”- This sounds like you want to study how the biogeochemistry of these two things interact. I suspect you actually have a different aim that isn't being described well here.

Reply:

Thank you for this astute observation. We acknowledge that the phrasing "N and P interactions" in its original form lacked precision in conveying the specific mechanistic focus of our study.

We understand that it might give the initial impression that the focus is solely on the general biogeochemical interaction of nitrogen and phosphorus. However, my actual aim is to delve into the specific and fundamental pathways through which these two elements interact in early - stage soils. In these soils, the lack of vegetation and simpler soil processes provide a unique and less - complicated environment. This allows for a more clear - cut investigation of how N and P influence each other's cycling and availability, as well as their combined effects on the initial development of soil properties and potential for supporting early plant colonization. The simplicity of the soil processes in the early stage acts as a kind of "baseline" that can help us better understand the core mechanisms of their interaction without the interference of too many other complex factors that are present in more developed soils.

We realize now that the original phrasing might not have fully conveyed this specific research goal. Based on this feedback of yours and other comments, we have rewritten the abstract. Please refer to the abstract in the revised manuscript text.

Thank you again for your helpful feedback. It is greatly appreciated and will definitely

contribute to improving the quality of my paper.

➤ **Comment 8:**

L13 Another nutrient addition experiment... Is there a hypothesis?.

Reply:

Regarding the "Another nutrient addition experiment" mentioned in the comment, we searched the original manuscript five times and unfortunately did not find this content. We guess that you may be referring to the nitrogen addition experiment and phosphorus addition experiment conducted in our study. If that's the case, the hypotheses corresponding to these two experiments are as follows:

1. An increase in N availability will increase the ecosystem's demand for P, leading to a decrease in available P and thereby exacerbating P limitation.

2. Similarly, based on elemental stoichiometric theory, an increase in P availability will increase the ecosystem's demand for N, leading to a decrease in available N and thereby exacerbating N limitation.

However, after analyzing and reflecting on your comments and those of other reviewers, we have recondensed and summarized the purpose of this study: Our aim is to attempt to integrate additional soil processes such as the carbon process and the nitrification process into the theoretical framework of nitrogen-phosphorus coupling, rather than focusing solely on the nitrogen fixation and phosphorus solubilization processes. Based on this approach, we seek to explore and reveal the positive feedback mechanisms among nutrient elements during the early stages of soil formation, providing empirical insights for the development of a unified theory of nitrogen-phosphorus coupling.

➤ **Comment 9:**

L14 This sounds like a fishing expedition without a testable hypothesis.

Reply:

Thank you for your comment. We recognize that the expression was inappropriate and have removed it in the revised manuscript.

➤ **Comment 10:**

L15-16 This seems detached from logic (add stuff there should be more not less right) and readers aren't presented with counter point for mature systems in Abstract introduction... Please rework.

Reply:

Yes, you are right. Based on this feedback of yours and other comments, we have rewritten the abstract. Please refer to the abstract in the revised manuscript text..

➤ **Comment 11:**

L17 "soil carbon (C) fixation processes" do you mean "soil carbon sequestration"?.

Reply:

In this study, the "soil carbon (C) fixation processes" do not mean "soil carbon sequestration". These terms are defined as follows: Carbon fixation denotes the process by which organic molecules are formed from inorganic carbon (Lawson et al., 2022), while carbon sequestration

refers to the capture and ensuing storage of carbon in order to prevent it from being released into the atmosphere (Duku et al., 2011; Zhou et al., 2021). Some scholars further emphasize that carbon sequestration specifically involves capturing atmospheric carbon dioxide and storing it long-term in plants, soils, and oceans (Sedjo and Sohngen, 2012; Perez-Verdin et al., 2016). From these definitions, it is evident that carbon fixation constitutes merely a non-essential step within carbon sequestration. The carbon fixation process emphasizes the transformation of inorganic carbon to organic carbon, without emphasizing long-term storage to prevent atmospheric release. In this research, carbon fixation processes are characterized by the increase in organic carbon, which serves as both carbon and energy sources for biological metabolic processes such as nitrogen fixation and phosphorus solubilization, ultimately being released as carbon dioxide. Therefore, the carbon fixation processes discussed in this study are distinct from carbon sequestration processes.

References:

- Duku, M.H., Gu, S., Hagan, E.B., 2011. Biochar production potential in Ghana—A review. *Renewable and Sustainable Energy Reviews* 15, 3539–3551. doi:<https://doi.org/10.1016/j.rser.2011.05.010>
- Lawson, T., Emmerson, R., Battle, M., Pullin, J., Wall, S., Hofmann, T.A., 2022. Chapter 3 - Carbon fixation, in: Ruban, A., Foyer, C.H., Murchie, E.H. (Eds.), *Photosynthesis in Action*. Academic Press, pp. 31–58. doi:<https://doi.org/10.1016/B978-0-12-823781-6.00008-3>
- Perez-Verdin, G., Sanjurjo-Rivera, E., Galicia, L., Ciro Hernandez-Diaz, J., Hernandez-Trejo, V., Antonio Marquez-Linares, M., 2016. Economic valuation of ecosystem services in Mexico: Current status and trends. *Ecosystem Services* 21, 6–19. doi:10.1016/j.ecoser.2016.07.003
- Sedjo, R., Sohngen, B., 2012. Carbon Sequestration in Forests and Soils. *Annual Review of Resource Economics*. doi:<https://doi.org/10.1146/annurev-resource-083110-115941>
- Zhou, Y., Qin, S., Verma, S., Sar, T., Sarsaiya, S., Ravindran, B., Liu, T., Sindhu, R., Patel, A.K., Binod, P., Varjani, S., Singhania, R.R., Zhang, Z., Awasthi, M.K., 2021. Production and beneficial impact of biochar for environmental application: A comprehensive review. *Bioresource Technology* 337. doi:10.1016/j.biortech.2021.125451

➤ **Comment 12:**

L20 “P solubilization” do you mean “P addition”? Also, I wonder if this is backwards. You added nutrients so isn't possible that nutrient additions accelerate various processes that might have been limited by initial soil nutrient levels? What about co-limitation?.

Reply:

We apologize for any lack of clarity in our manuscript that may have led to your questions. Please allow us to address your concerns as follows:

Firstly, in our study, P solubilization does not equate to P addition. In fact, P addition simulates the outcome of the P solubilization process, that is, it improves the availability of P. In order to highlight and amplify the effects caused by the outcome of the P solubilization process (such as the effects on the N fixation process and the C fixation process) for the purpose of measurement, we carried out the operation of P addition. Of course, the added bioavailable P may accelerate or inhibit other processes. According to our experimental data, the added bioavailable P promoted both N fixation and C fixation, thereby confirming one aspect of nitrogen-phosphorus

co-limitation: P limitation.

We would also like to add that our N addition experiments revealed increased abundance of phosphate solubilization-related genes, elevated concentrations of organic acids, and enhanced C fixation. Thus, the experiment confirmed other aspect of nitrogen-phosphorus co-limitation: N limitation.

These experimental designs allowed us to isolate mechanistic drivers while still capturing the overall patterns of co-limitation.

At this point, you might be wondering: Why not conduct experiments with simultaneous N and P addition? To this, our response is that we did perform experiments with simultaneous N and P additions (though the data were not shown in the original manuscript), because in these experiments, since N and P were added simultaneously, we found it impossible to distinguish between the individual effects of N or P, and thus could not convincingly identify the pathways through which N affects P (or vice versa). However, the results indicated positive effects such as enhanced P availability, increased N availability, increased P solubilization-related gene abundances, higher concentrations of organic acids, and so forth, supporting the nitrogen-phosphorus co-limitation hypothesis. Nevertheless, proving the nitrogen-phosphorus co-limitation was not the focus of our study; rather, our interest lay in understanding the pathways through which N and P interact. Therefore, the data from the simultaneous N and P addition experiments were not included in the original manuscript.

To address your concern and fully substantiate our findings, the results of experiments with simultaneous N and P addition have been added to the supplementary materials (see Table S7 source data for the treatment with combined N and P addition.xlsx and Figure S2 in Supplementary materials).

We appreciate your valuable feedback and will incorporate these clarifications into the revised manuscript.

➤ **Comment 13:**

L22 Unclear the basis for this interpretation or modes of what? Passage feels like a word salad.

L23 same.

Reply:

We sincerely appreciate your critique regarding the lack of clarity in this section. We fully acknowledge that the original passage suffered from undefined terminology (e.g., ambiguous use of "modes") and insufficient linkage between assertions and supporting evidence, resulting in unintended obfuscation. Please accept our apologies for this oversight.

To address your concerns, we will implement the following revisions.

As a brief illustration, the restructured paragraph will now read:

Furthermore, based on previous research findings and the experimental data from this study, we explored and established a model depicting the driving pattern of P availability on N availability in early soils. Similarly, another mode was constructed for the driving pattern of N availability on P availability. These results emphasized the central role of carbon-related processes in mediating nitrogen and phosphorus coupling, thereby enhancing their availability in early soil environments. These models also reveal a pathway in early soil development where enrichment of one nutrient enhances the co-availability of another rather than intensifying nutrient limitation,

suggesting that synergistic N-P coupling dominates interactions between N and P bioavailability during early soil development. This provides an explanation for the drivers of positive succession in early soil ecosystems.

To incorporate all the modifications regarding the abstract, we have rewritten the abstract (see line 30-51 in Manuscript text (Revised Document with Track Changes)).

We are grateful for your rigorous scrutiny, which has significantly strengthened this manuscript. Should further clarifications be needed, we eagerly await your guidance.

➤ **Comment 14:**

L29 “The mobilization of nitrogen (N) and phosphorus (P) constitutes a fundamental and critical soil process, with the bioavailability dynamics of these elements being among the most ecologically significant outcomes in this process.” Wordy and redundant.

Reply:

In accordance with your comment, we have revised and rewritten the introduction (see line 146-250 in Manuscript text (Revised Document with Track Changes)).

➤ **Comment 15:**

L36 I would probably replace citations 1,2 with (Fay et al. 2015, Du et al. 2020)..

Reply:

Thank you very much for recommending these two papers. After carefully reading the two papers you recommended, we feel that they are more suitable for supporting the viewpoints in our paper than the originally cited ones. We have replaced the original references with the ones you recommended(see 711-712 in Manuscript text (Revised Document with Track Changes)).

References

- Fay, P.A., Prober, S.M., Harpole, W.S., Knops, J.M.H., Bakker, J.D., Borer, E.T., et al., 2015. Grassland productivity limited by multiple nutrients. *Nature Plants* 1. doi:10.1038/NPLANTS.2015.80
- Du, E., Terrer, C., Pellegrini, A.F.A., Ahlstrom, A., van Lissa, C.J., Zhao, X., Xia, N., Wu, X., Jackson, R.B., 2020. Global patterns of terrestrial nitrogen and phosphorus limitation. *Nature Geoscience*. doi:10.1038/s41561-019-0530-4

➤ **Comment 16:**

L36 “Most notably, in terms of mechanism, the prevalence of N and P co-limitation hinted strongly the couplings between these two nutrients in ecosystems².”- wordy & confusing.

Reply:

Thank you for your feedback. Based on your suggestions, we have revised and rewritten the expression to improve clarity and conciseness. The updated version (see line 149-155 in Manuscript text (Revised Document with Track Changes)), along with its context, is provided below:

Empirical evidence demonstrates that across various ecosystems, plant growth exhibits comparable increments following the addition of either N or P, which suggests the prevalence of N and P co-limitation^{5,8}. Moreover, given this fact that organisms require elements in stoichiometric proportion and are expected to take up and recycle nutrients in ratios that maintain balanced nutrition^{9,10}, some scholars² considered that such co-limitation profoundly suggests an intrinsic

coupling of N and P through soil biological processes, highlighting the interconnected dynamics of these nutrient cycles within terrestrial environments.

➤ **Comment 17:**

L38 “The establishment and maintenance of this coupling relationship cannot be separated from the most active components (i.e., available N and P).” – This doesn’t make sense to me. My impression is that N is highly labile while available P is not and sorbs to numerous soil components (clay, Ca, Calcium carbonate, Fe, Al). The next sentence hints at some of these issues but seems to ignore how fundamentally different these two nutrient behave in the soil solution and interact with other things.

Reply:

We sincerely appreciate your insightful comment regarding the distinct behaviors of nitrogen (N) and phosphorus (P) in soil systems. You is absolutely correct in highlighting the fundamental differences in their mobility and interaction with soil components. Below, we clarify our original intent and propose revisions to address this concern:

We agree that the original statement oversimplified the dynamics of available N and P. As you rightly point out, mineral N (e.g., NH_4^+ , NO_3^-) is highly labile and prone to leaching or volatilization, while available P (e.g., H_2PO_4^- , HPO_4^{2-}) exhibits strong sorption to soil constituents (e.g., Fe/Al oxides, carbonates, clays) and forms stable complexes. This distinction is critical to their roles in nutrient coupling.

Our intention was to emphasize that both nutrients—despite their contrasting behaviors—are essential drivers of biogeochemical coupling in the studied system. For instance, N availability regulates microbial P mineralization (via phosphatase production), while P limitation can constrain N fixation (Vitousek et al., 2010). Although available N and available P have many differences, as you mentioned that available N is highly labile while available P is stable and easily fixed by soil components, they share a common characteristic, which is that they are easily utilized by organisms (i.e., they have high biological availability), and they can affect each other's biological availability when they are absorbed by organisms and participate in biological processes. For example, the addition of available N can enhance biological processes such as nitrification, acidify the soil, and solubilize the available P fixed by soil components; the addition of available P can promote biological N fixation and increase the level of available N in the soil. Obviously, these processes illustrate the coupling effect between available N and available P. However, we recognize that the original phrasing failed to explicitly address their divergent retention mechanisms.

Therefore, considering your comments and the logical context as a whole, we have decided to remove this expression. we have revised and rewritten the introduction (see line 146-250 in Manuscript text (Revised Document with Track Changes))

References

Vitousek, P.M., Porder, S., Houlton, B.Z., Chadwick, O.A., 2010. Terrestrial phosphorus limitation: mechanisms, implications, and nitrogen–phosphorus interactions. *Ecological Applications* 20, 5–15. doi:10.1890/08-0127.1

➤ **Comment 18:**

L41 This statement feels a bit superficial and doesn't include citations. P is incredibly immobile (Jungk 2002) and that should probably be noted and possibly differentiated from N. Also, it seems like 10x more research has been conducted on N..

Reply:

Through your comments, we have recognized that our statement did not effectively convey our research intent to the readers. Therefore, we have removed this statement and reorganized the content to rewrite this section(see line 146-167 in Manuscript text (Revised Document with Track Changes)).

➤ **Comment 19:**

L44 phosphatase activity concerns mineralization of organic matter. In early seral environments, there is little OM. I would suspect that other processes (weathering, P solubilizing bacteria) are likely more important for P mobilization.

Reply:

You are correct. In our research area, due to the lack of vegetation producing organic matter, the organic matter content in these early soils is very low. Previous research findings(Prietz et al., 2013; Zhou et al., 2018) in this area indicate that mineral phosphorus (i.e., inorganic phosphorus) is the absolute primary component of total phosphorus in these early soils (i.e., inorganic phosphorus is almost equal to total soil phosphorus, while the concentration of organic phosphorus is below the detection limit). Therefore, the mineralization of organic phosphorus is not a major pathway for phosphorus mobilization in these early soils. Instead, bioweathering involving phosphate-solubilizing bacteria is the main pathway for phosphorus mobilization in these early soils. Accordingly, in this study, we use indicators related to the dissolution of mineral phosphorus through bioweathering (such as pH, organic acids, available phosphorus concentration) to investigate the process of phosphorus mobilization.

References:

- Prietz, J., Duemig, A., Wu, Y., Zhou, J., Klysubun, W., 2013. Synchrotron-based P K-edge XANES spectroscopy reveals rapid changes of phosphorus speciation in the topsoil of two glacier foreland chronosequences. *Geochimica ET Cosmochimica Acta* 108, 154–171. doi:10.1016/j.gca.2013.01.029
- Zhou, J., Bing, H., Wu, Y., Sun, H., Wang, J., 2018. Weathering of primary mineral phosphate in the early stages of ecosystem development in the Hailuogou Glacier foreland chronosequence. *European Journal of Soil Science* 69, 450–461. doi:10.1111/ejss.12536

➤ **Comment 20:**

L44-45 It seems most of your citations are for nutrient additions. Are these realistic and/or good ways for understanding processing affecting nutrient availability in most natural systems? It seems like a crude tool, especially for understanding P availability which can be sorbed to various soil particles (Jones et al. 2013b).

Reply:

We sincerely thank the reviewer for raising this critical point regarding the ecological realism of nutrient addition experiments, particularly in the context of phosphorus (P) dynamics influenced by soil-particle interactions (Jones et al. 2013b). We fully acknowledge that nutrient additions represent a simplified experimental approach and agree that sorption

processes—especially for P—may limit direct extrapolation to highly weathered or mineralogically complex soils. However, we propose that our methodology is both appropriate and informative for the specific context of early-stage soil ecosystems. Below, we clarify the rationale for our methodology while integrating the reviewer's concerns into our revised discussion.

1) Unique substrate characteristics:

The study area consists of freshly exposed rock following glacial retreat. Strictly speaking, the early-stage soils studied are composed of rock debris particles, which are predominantly made up of primary minerals (with minimal clay minerals) (Yang et al., 2015; Sun et al., 2022) and exhibit very weak adsorption of available phosphorus. It is important to emphasize that these early-stage soils are very young soils, not highly weathered soils with strong adsorption capacities.

2) Methodological superiority:

Although the nutrient addition method is simple, it allows for minimal introduction of confounding factors other than the variables under investigation. This ensures that the factors set in the experiment can independently explain the results. Additionally, the method is straightforward to implement and has low maintenance costs, making it well-suited for in situ field studies on nutrient availability effects.

3) Scientific novelty:

Currently, the nutrient addition method has been widely applied in various ecological environments (over 7000 studies since 2015, Web of Science search: "nutrient addition" AND "experiment"). This not only indicates that the nutrient addition method has a comprehensive dataset for generalization and synthesis, but also demonstrates its effectiveness as a research method—otherwise, it would not have been adopted by so many studies. However, despite the widespread use of nutrient addition in research, very few studies have applied this method to early-stage soil ecosystems, highlighting one aspect of the significance of our work.

References:

- Jones, M.P., Webb, B.L., Jolley, V.D., Vickery, M.D., Buck, R.L., Hopkins, B.G., 2013b. Evaluating Nutrient Availability in Semi-arid Soils with Resin Capsules and Conventional Soil Tests, II: Field Studies. *Communications in Soil Science and Plant Analysis* 44, 1764–1775. doi:10.1080/00103624.2013.769564
- Sun, H., Wu, Y., Zhou, J., Yu, D., Chen, Y., 2022. Microorganisms drive stabilization and accumulation of organic phosphorus: An incubation experiment. *Soil Biology and Biochemistry* 172, 108750. doi:10.1016/j.soilbio.2022.108750
- Yang, Z., Bing, H., Zhou, J., Wu, Y., Sun, H., Luo, J., Sun, S., Wang, J., 2015. Variation of mineral composition along the soil chronosequence at the Hailuoguo Glacier foreland of Gongga Mountain. *Acta Pedologica Sinica* 52, 507–516. (Published in Chinese)

➤ **Comment 21:**

L47 I think only one of the supporting sentences in this paragraph included citations....

Reply:

According to your other comments and guidance mentioned above, we have rewritten this paragraph. Please see lines 146 to 167 in Manuscript text (Revised Document with Track Changes).

➤ **Comment 22:**

L48 “interrelated”- I don’t follow how N and P availability is interrelated.

Reply:

We sincerely thank you for your feedback and apologize for any lack of clarity in our original text. We have removed the problematic sentence and rewritten the section for improved precision. In the original manuscript, when we referred to 'the interaction between nitrogen (N) and phosphorus (P) availability,' we meant that N availability (or P availability) can influence P availability (or N availability) through soil biological processes. This perspective is grounded in the well-established understanding, as noted in our revised Introduction (see line 146-250 in Manuscript text (Revised Document with Track Changes)):

“The interaction between nitrogen (N) and phosphorus (P) cycling processes has long been a focal point in the field of biogeochemistry. It is universally acknowledged that global terrestrial ecosystems are constrained by both N and P availability. Empirical evidence demonstrates that across various ecosystems, plant growth exhibits comparable increments following the addition of either N or P, which suggests the prevalence of N and P co-limitation. Moreover, given this fact that organisms require elements in stoichiometric proportion and are expected to take up and recycle nutrients in ratios that maintain balanced nutrition, some scholars considered that such co-limitation profoundly suggests an intrinsic coupling of N and P through soil biological processes, highlighting the interconnected dynamics of these nutrient cycles within terrestrial environments.”

➤ **Comment 23:**

L53-55 I don’t follow the logic of this statement. Occurring plants are known to use different startagies to acquire P (e.g. Raven et al. 2018).

Reply:

We have deleted this sentence and reorganized the logic in the Introduction section. Please review the revised Introduction section of the manuscript.

➤ **Comment 24:**

L55-57 I don’t follow the logic of this statement either. Poor word choice. What two analytical perspectives?.

Reply:

We have reorganized the logic for clarity. The two analytical perspectives are discussed in the immediately following text, namely:

1) “One analytical perspective focuses on the impact of N on indicators related to P processes, usually through N amendment experiments.”

2) “The other analytical perspective examines the influence of P on indicators related to N processes, typically through P addition experiments.”

➤ **Comment 25:**

L58 perspective on what? Poor word choice and poor starting sentence of a paragraph (which is often the paragraph’s topical sentence)..

Reply:

This perspective concerns the impact of N on indicators related to P processes and the impact

of P on indicators related to N processes. The two perspectives are expressed in the revised manuscript as follows:

1) “One analytical perspective focuses on the impact of N on indicators related to P processes, usually through N amendment experiments.”

2) “The other analytical perspective examines the influence of P on indicators related to N processes, typically through P addition experiments.”

➤ **Comment 26:**

L59 Again, nutrient addition seems like a rather crude tool to build understanding.

Reply:

This comment addresses a similar issue as the one raised in 'Comment 20' above. Please refer to our response to 'Comment 20' for a detailed explanation.

➤ **Comment 27:**

L60 Again, this feels superficial as there is no mention of the conditions that might cause one to predominate over the other. These are some relevant studies to look over (e.g. Ryan et al. 2012, Reinhart et al. 2024 and citations therein).

Reply:

We are extremely grateful for the valuable comments you have provided on our paper. You pointed out that our description of the soil phosphorus process is rather superficial, lacking a specific discussion on which of the mineralization of organic phosphorus and the dissolution of inorganic phosphorus is more dominant under different conditions. This is a very important perspective, and we fully agree with it.

In accordance with your suggestion, we have revised this part. The revised content explores which of the mineralization of organic phosphorus and the dissolution of inorganic phosphorus is more dominant under different conditions. For the detailed content, please see lines 169 to 175 in Manuscript text (Revised Document with Track Changes)

➤ **Comment 28:**

L62 Don't all enzymes require N?.

Reply:

Thank you for raising this question. Indeed, many enzymes contain nitrogen as a part of their structure, so your point is correct. However, in our statement, what I intended to emphasize was the role of nitrogen in phosphatases during the phosphorus cycle, especially in the process of organic phosphorus mineralization. Some studies have shown that the nitrogen content in phosphatases ranges from 8% to 32% (Treseder and Vitousek, 2001), suggesting that nitrogen is an unavoidable factor in terms of its importance for phosphatases when studying the process of organic phosphorus mineralization. Indeed, the influence of nitrogen has been taken into account in many studies that focus on phosphatases (Godin et al., 2015; Ratliff and Fisk, 2016; Chen et al., 2020; Mori et al., 2024; Tian et al., 2025).

References:

Chen, J., van Groenigen, K.J., Hungate, B.A., Terrer, C., van Groenigen, J.-W., Maestre, F.T., Ying, S.C., Luo, Y., Jorgensen, U., Sinsabaugh, R.L., Olesen, J.E., Elsgaard, L., 2020. Long-term nitrogen loading alleviates

- phosphorus limitation in terrestrial ecosystems. *Global Change Biology* 26, 5077–5086. doi:10.1111/gcb.15218
- Godin, A.M., Lidher, K.K., Whiteside, M.D., Jones, M.D., 2015. Control of soil phosphatase activities at millimeter scales in a mixed paper birch - Douglas-fir forest: The importance of carbon and nitrogen. *Soil Biology & Biochemistry* 80, 62–69. doi:10.1016/j.soilbio.2014.09.022
- Mori, T., Wang, S., Wang, C., Zhang, W., Mo, J., 2024. Effect of long-term nitrogen addition on the kinetics of phosphatases in a subtropical forest in southern China. *Applied Soil Ecology* 202. doi:10.1016/j.apsoil.2024.105589
- Ratliff, T.J., Fisk, M.C., 2016. Phosphatase activity is related to N availability but not P availability across hardwood forests in the northeastern United States. *Soil Biology & Biochemistry* 94, 61–69. doi:10.1016/j.soilbio.2015.11.009
- Tian, M., Jiang, N., Chen, Zhenhua, Zhang, Y., Jiang, D., Wu, C., Chen, Zhuoran, Qiu, W., Wang, J., 2025. Nitrogen and carbon addition mediate phosphorus cycling in grassland ecosystems: Insights from *phoD* gene abundance and community diversity. *Applied Soil Ecology* 206. doi:10.1016/j.apsoil.2025.105896
- Treseder, K.K., Vitousek, P.M., 2001. Effects of soil nutrient availability on investment in acquisition of N and P in Hawaiian rain forests. *Ecology* 82, 946–954. doi:10.1890/0012-9658(2001)082[0946:EOSNAO]2.0.CO;2

➤ **Comment 29:**

L64 “conclusions” on what?.

Reply:

Thank you for your comment. Upon reading this comment, we understand that although we elaborated on the "conclusions" in the following text, the way we expressed our ideas hindered readers from effectively accessing the information. Specifically, the conclusions referred to are those from previous studies discussed later in the text, which address whether nitrogen addition indeed enhances phosphatase activity.

In light of your feedback, we have revised the text to ensure that the subject of our conclusions is clear and that the discussion is more accessible to the reader. For the specific revised text, please see lines 179 to 185 in Manuscript text (Revised Document with Track Changes).

➤ **Comment 30:**

L63-7 wordy.

Reply:

Thank you for your feedback. According to your comments, we have rewritten this section to ensure it is concise and clear. For the specific revised text, please see lines 179 to 185 in Manuscript text (Revised Document with Track Changes).

➤ **Comment 31:**

L77 Meta-analysis tailor their study one questions with sufficient data to justify tests. They are effectively hamstrung by the prelevance of data (often collected by other people).

Reply:

We quite agree with the viewpoints in your comment. The reason why we cited these meta - analysis studies is that these studies have synthesized a large number of case studies on the relationship between nitrogen addition and phosphatase activity. Logically, the conclusions drawn

from these meta - studies through comprehensive analysis are more likely to be more universal and credible than those of individual case studies. Nevertheless, inconsistent conclusions have also been reached among these meta - analysis studies. This largely indicates that in - depth research on the relationship between nitrogen and phosphorus is still needed.

To highlight the necessity of integrating other processes apart from the N and P processes in the study of the N-P relationship, based on revealing the limitations of current research (including the case studies in meta-research), we added citations to demonstrate the importance of the C process in the study of the nitrogen-phosphorus relationship. Based on this, we improved this part of the text. For the specific revised text, please see lines 185 to 192 in Manuscript text (Revised Document with Track Changes).

➤ **Comment 32:**

L79 Good point but content seems like something that should have been mentioned earlier..

Reply:

. We are very glad to receive your appreciation. According to your comments, we have adjusted and revised the text. For the specific revised text, please see lines 189 to 192 in Manuscript text (Revised Document with Track Changes).

➤ **Comment 33:**

Comment- The authors seem to be using the inconsistent results of three global meta-analyses as motivation for the current study. This seems like a weak motivation for an empirical study. For example, the presence of meta-analyses alone indicate that there is already a lot of work on the topic. How is your work unique. If you are simply conducting a replication of prior studies then that work should be targeted to journals like PLoS One; Agrosystems, Geosciences & Environment; etc..

Reply:

Thank you for your comment. Due to the limitations of our writing skills, we failed to fully explain the unique motivation of our study in the introduction section of the original manuscript. Based on your comments and guidance, as well as those from other reviewers, we have revised and rewritten the introduction section.

In the revised content, we have placed particular emphasis on elaborating the necessity of integrating relevant processes other than the nitrogen and phosphorus processes into the nitrogen-phosphorus coupling framework. Additionally, we have highlighted that our study uniquely integrates the carbon process and the nitrification process into the nitrogen-phosphorus coupling framework - a dimension that has been overlooked in previous nutrient addition experiments but is crucial for explaining context-dependent responses.

For the specific revised text, please see lines 168 to 226 in Manuscript text (Revised Document with Track Changes).

➤ **Comment 34:**

L80 This seems unlikely. Many soil scientists have studied the effects of liming (to counter act soil acidification with fertilization) on available P and effects of fertilizer on soil pH (which can regulate available P). It is well known that N additions acidify soil and this can increase soil P depending on the soil's starting pH.

Reply:

Thank you for your feedback. We are sorry for the hasty conclusion written in that manner. According to your comments, we have deleted that text.

➤ Comment 35:

L88 “promoting effect of P is always significant.” – vague & poor word choice. “This” refers to what?.

Reply:

The term 'this' refers to the previously mentioned statement that 'exogenous P inputs often enhance diazotrophic activity and N fixation rate.' According to your comments, we have revised this text. For the specific revised text, please see lines 206 to 209 in Manuscript text (Revised Document with Track Changes).

➤ Comment 36:

L98 good point.

Reply:

We are genuinely gratified by your favorable assessment, which provides compelling affirmation of the validity embedded within our conceptual framework. It is noteworthy that your commentary has not only informed but also catalyzed substantive refinements across multiple manuscript sections.

➤ Comment 37:

L109 You should clearly (re-)state the research gap that is being addressed.

Reply:

In accordance with your comments, we have rewritten this text and clearly stated the research gap, that is, integrating related processes such as the carbon process and the nitrification process into a nitrogen-phosphorus coupling framework is crucial for interpreting environment-dependent responses and resolving contradictions in the conclusions, and this concept has been overlooked in previous nutrient addition experiments.

For the specific revised text, please see lines 189-192, 202-204, and 224-226 in Manuscript text (Revised Document with Track Changes).

➤ Comment 38:

L113 citation #2- I think you mean optimal allocation theory (Bloom et al. 1985)..

Reply:

Yes, this perspective incorporates the optimal allocation theory proposed by Bloom et al. (1985). Specifically, this view was summarized by Marklein and Houlton(2012) from the papers of Bloom et al. (1985) and Chapin et al. (2002). This perspective holds that plants (as well as microorganisms) are expected to allocate their resource reserves to strategies that can enhance the acquisition of the most limiting resources – thus moving towards a state where all resources simultaneously limit productivity and growth. Following the logic of this perspective, it can be inferred that ecosystems will degenerate and disappear because an increasing variety of nutrients become limiting factors. However, this perspective does not seem to be able to explain the phenomenon of primary succession in pioneer ecosystems (such as our study area): Pioneer

ecosystems are so nutrient-poor that plants cannot grow, but they can undergo succession to develop into forest ecosystems rich in available nutrients.

References:

- Bloom, A.J., Chapin, F.S., Mooney, H.A., 1985. Resource Limitation in Plants—An Economic Analogy. *Annual Review of Ecology and Systematics* 16, 363–392.
- Chapin, F.I., Mooney, H., Chapin, M., Matson, P., 2002. *Principles of terrestrial ecosystem ecology*. Springer, New York, NY, USA.
- Marklein, A.R., Houlton, B.Z., 2012. Nitrogen inputs accelerate phosphorus cycling rates across a wide variety of terrestrial ecosystems. *New Phytologist* 193, 696–704. doi:10.1111/j.1469-8137.2011.03967.x

➤ **Comment 39:**

L116 sentence has similar problem to L109..

Reply:

According to your comments, we have rewritten the text. For the specific revised text, please see lines 244 to 250 in Manuscript text (Revised Document with Track Changes).

➤ **Comment 40:**

L146 poorly worded sentence. Try- X positively affected Y and negatively affected Z..

Reply:

It is our great honor to receive your guidance. According to your instructions, we have revised the sentence to clarify the contrasting effects of N addition as follows (see line 277-280 in Manuscript text (Revised Document with Track Changes)):

N addition treatments negatively affected pH (causing a significant decrease), but positively affected AP concentrations, low molecular weight organic acids levels, and the abundance of acid secretion genes (*gcd*), all of which showed significant increases.

➤ **Comment 41:**

L151 leaching- What about sorption? If the sorption capacity of the soil is high that there may be no (or a very temporary) increase in AP (Jones et al. 2013a, Jones et al. 2013b)..

Reply:

The adsorption capacity of this early-stage soil should be very weak. The study area is the bedrock that emerged right after the glacier retreated. The soil here is extremely young and has not undergone long-term weathering. It contains almost no secondary minerals (such as clay minerals) with strong adsorption capabilities (Yang et al., 2015). Strictly speaking, this soil is more like rock debris (Sun et al., 2022), as shown in Figure S1 in Supplementary materials. Moreover, our results indicate that, compared with the control group, the phosphorus addition treatment significantly increased the concentration of available phosphorus in the soil (Figure 2a of our manuscript). Therefore, the ability of this early-stage soil to adsorb available phosphorus is weak.

➤ **Comment 42:**

L173 Okay but pH was affected by N addition right (Fig. 2)?.

Reply:

Yes, our results indicate that the addition of nitrogen indeed affects the pH value, especially

considering that the soil in our study area is alkaline..

➤ **Comment 43:**

L340 It is somewhat curious that with 90 replicate pots (+N, +P, and control) that you didn't include an N&P addition and reduce your replication number per treatment.

Reply:

Actually, we did conduct experiments with both N and P additions (though the data were not presented in the original manuscript). Since both N and P were added simultaneously in these treatments, we found that it was impossible to distinguish the individual effects of N or P, and we were unable to convincingly identify the pathways through which N affects P (or vice versa). While the results from this treatment indicated positive effects such as increased P and N availability, enhanced abundance of nitrogen-fixing genes, elevated levels of phosphorus-solubilizing related genes, higher concentrations of organic acids, etc., illustrating the co-limitation by N and P, our study's focus was not on proving their co-limitation but rather on understanding the mechanisms of interaction between N and P. To streamline the presentation of our findings and reduce redundancy in the manuscript, the data from the simultaneous addition of N and P were omitted from the original manuscript. However, to address your queries and provide robust support for our findings, the data and analysis results from the N and P dual addition experiment have been included in the supplementary materials of the revised manuscript (see Table S7 source data for the treatment with combined N and P addition.xlsx and Figure S2 in Supplementary materials).

➤ **Comment 44:**

L348 “to prevent lateral leaching of N or P”- I sort of understand the decision to bag and reinsert cores but this also adds layers of disturbance (coring) and abstraction (bagged soil that prevents leaching) which makes the results a bit unrealistic compared to the more common practice of applying fertilizer to entire plots.

Reply:

Thank you for raising this important methodological concern. We deeply appreciate your thoughtful feedback and would like to clarify our experimental design with additional technical details to address your inquiry comprehensively.

First of all, we would like to clarify that the container we used to hold the undisturbed soil cores is not a bag with one open end and one sealed end, but a cylindrical plastic film container with both ends open. This cylindrical container not only minimizes the interference with the natural leaching phenomenon but also prevents the contamination of other surrounding experimental soil cores by lateral seepage. In fact, this experimental design is essentially a miniaturized version of a partitioned plot experiment. Secondly, if N and P were applied to the entire plot, it would cause pollution to this nature reserve (supplementary note: our study area is within a national protected glacier forest park). Compared with the plot fertilization experiment, our experimental design can limit the pollution to a very small area. Moreover, after the experiment, by digging out the entire cylindrical plastic film container, it is conducive to the removal of these treated soil cores, thus reducing the pollution to the original ecosystem. We fully understand and appreciate the importance of balancing rigorous scientific inquiry with environmental stewardship. Our experimental design reflects a conscientious effort to advance

research objectives while safeguarding the integrity of this ecologically sensitive area. We hope this explanation clarifies our methodology and addresses your concerns adequately.

To summarize, the reasons for designing our experiment in this manner are as follows:

1). Experimental Container Design

The soil containment system employs a dual-open-ended cylindrical polyethylene sleeve rather than a single-sealed bag structure. This design specifically preserves natural vertical hydrological connectivity while preventing lateral solute transport through three key mechanisms:

- a. The high-density polyethylene film creates a hydrologically isolated microenvironment
- b. Vertical alignment maintains intact preferential flow paths within soil macropores
- c. Absence of lateral permeability eliminates cross-contamination between adjacent cores

Notably, this configuration achieves functional equivalence to partitioned field plots in terms of hydrological isolation, while offering superior control over microscale biogeochemical processes.

2). Environmental Protection Considerations

Given that our study area is situated within the Gongga Mountain National Glacier Forest Park (Class I Protected Area), we implemented a rigorous hierarchical pollution mitigation protocol:

- a. Spatial confinement: Fertilization was restricted to the cylindrical microcosms (<0.1 m² footprint per unit), reducing impacted area compared to traditional plot-scale applications
- b. Temporal control: Pulse fertilization events were synchronized with natural precipitation cycles to minimize solute retention
- c. Post-experimental remediation: A complete removal protocol was executed, involving: Full excavation of sleeve-contained soil volumes and Site restoration.

➤ **Comment 45:**

L360-1 So each sample (270) were separately assayed for physicochemical properties and genes? Were any samples lost or destroyed [I think that was motivation for the large number of plots]?

Reply:

Yes, a total of 270 samples were separately assayed for physicochemical properties and genes (we have annotated the sample sizes (i.e., n values) used for calculating the respective metrics in both the tables and figures). From the description of the methods in the manuscript, you can see that the number of sample plots set up is more than 270. This was done for the following purposes:

1). To ensure that the experimental data accurately reflect the true conditions of the soil in the study area and to guarantee the robustness of the data analysis results. This is because the diverse microtopography in the study area leads to significant soil heterogeneity.

2). Since this area is a high mountain area (the highest peak in the area is 7,556 meters above sea level) and an active glacier retreat area, and small-scale geological disasters (such as small debris flows, small landslides, etc.) occur frequently. Considering the situation that the experimental sample plots may be damaged by these disasters during the experiment, in order to ensure that 90 samples can be obtained from each treatment at the end of the experiment, the number of sample plots was set to be more than 270. Such a sample size is conducive to constructing a robust structural equation model.

➤ **Comment 46:**

L390 You probably need to spell out the samples size for the various measurements (ST, BD) that were not based on the 270 soil cores..

Reply:

Based on your comments, we have annotated the sample sizes (i.e., n values) used for calculating the respective metrics in both the tables and figures.

➤ **Comment 47:**

L589 Fig. 2 header should mention the number of plot (or core) replications per treatment (e.g. n= 90 plot replications per treatment).

Reply:

Yes, according to your comment, we have annotated the sample sizes (i.e., n values) used for calculating the respective metrics in both the tables and figures.

➤ **Comment 48:**

Fig. S1 This figure is somewhat helpful but I was expecting to see something with more detail on where the plots were positioned or plot layout diagram or photos. The two pictures of the bags fail to show plots. Ex/ How big is the area where the 90 plots (per treatment) were laid out (also not shown on the map)?.

Reply:

We have redrawn "Fig. S1". Specifically, we have added a real-scene photograph of the study area and a schematic diagram of the layout of the plots to show details such as the location and size of the plots.(see Figure S1 in Supplementary materials)

➤ **Comment 49:**

Excel data file doesn't appear to include information on bulk density, soil temperature, and moisture.

Reply:

Based on your comments, we have added information regarding bulk density, soil temperature, and soil moisture in the supplementary materials (see Table S9 soil property data.xlsx).

Response to Referees Letter

Dear Referees:

We have studied the valuable comments from you carefully, and tried our best to revise the manuscript(NCOMMS-24-66832B). In response to the valuable suggestions from you, we have undertaken systematic revisions to the manuscript. In the introduction section, through a comprehensive review of research advances in the field, we identified significant discrepancies in existing literature regarding the conclusions of nitrogen-phosphorus interactions. Additionally, the long-neglected driving role of carbon in nitrogen and phosphorus cycling processes was highlighted. Based on these findings, we propose integrating carbon-related processes into the theoretical framework of nitrogen-phosphorus coupling, which is critical for elucidating the environmental-dependent response mechanisms of nitrogen-phosphorus interactions and bridging discrepancies in research conclusions across different studies.

At the experimental data level, we have supplemented thousands of key data points as recommended by reviewers, specifically including: 1) measurements of total organic carbon (TOC) content and *cbbL* gene mRNA transcription abundance in 450 soil samples; 2) experimental data from treatments with simultaneous nitrogen and phosphorus additions; 3) detailed results of physicochemical property tests for natural soils; and 4) observed data on pH dynamics in nitrification experiments with nitrogen additions. These newly added data provide more robust empirical support for our research conclusions.

We believe that these revisions—through the innovative construction of the theoretical framework and the refinement of the data system—not only strengthen the scientific rigor of our conclusions but also make a substantial contribution to the development of nitrogen-phosphorus coupling theory.

The point to point replies to the referees' comments are listed as following:

Reply to Reviewer #2

➤ Comment 1:

The work is interesting, and the manuscript is well written and structured. Overall, this is an excellent study that provides significant value in enhancing our understanding of the coupling mechanisms of nitrogen and phosphorus in soils. Firstly, I greatly appreciate the research perspective presented in this manuscript. Rather than focusing on mature ecosystems as is common in previous studies of N and P cycling mechanisms, this study explores these mechanisms from the vantage point of early soils, which are the origins of mature ecosystems and involve relatively simpler processes. This approach is particularly advantageous for elucidating the mechanisms that are obscured by complex interactions in more developed systems. For instance, the experiments conducted in this study reveal that the enrichment of one nutrient in the soil can enhance the availability of other nutrients, a finding that aligns well with my expectations and starkly contrasts with the traditional view that the abundance of one nutrient would lead to limiting effects on others. Therefore, I find this research to be highly innovative. Secondly, for a field experimental study, the volume of data utilized in this research is remarkably large, even comparable to that used in some meta-analyses. I am confident that the reliability of the data analysis results from this study surpasses that of many similar studies.

Reply:

We sincerely appreciate the insightful comments and valuable suggestions provided by you. Your recognition of the novelty and methodological rigor of our work is particularly encouraging. We have carefully addressed all points raised and believe these revisions have significantly enhanced the manuscript.

➤ Comment 2:

Line 10: "N and P" should be used as abbreviations for nitrogen and phosphorus.

Reply:

Based on your comments, we have corrected these irregular words. Moreover, by incorporating the suggestions of other reviewers, we have rewritten the abstract of this manuscript.(see line 30-51 in Manuscript text (Revised Document with Track Changes))

➤ Comment 3:

Line 40-42: "In soil environments, the bioavailability of N and P is governed by a suite of intricate soil processes, including N fixation, phosphate dissolution, organic matter mineralization, and biological uptake." These processes should also include soil leaching.

Reply:

In accordance with your comments and by incorporating the opinions of other reviewers, we have rewritten the introduction of this manuscript. (see line 146-250 in Manuscript text (Revised Document with Track Changes))

➤ Comment 4:

Line 43: "...the impacts of N or P availability..." The word "or" fits better in this context than "and".

Reply:

Based on your comments, we have corrected these irregular words. Additionally, taking into account the comments of other reviewers, we have rewritten the introduction of this manuscript. (see line 146-250 in Manuscript text (Revised Document with Track Changes))

➤ Comment 5:

Line 61: please provide a reference for "These processes are influenced by N availability".

Reply:

According to your comments, I have added the relevant references. Moreover, through your comments and the opinions of other reviewers(see line 168-204 in Manuscript text (Revised Document with Track Changes)), we realized that the introduction of the manuscript needed a thorough improvement, so we have rewritten the introduction of this manuscript. (see line 146-250)

➤ Comment 6:

Line 63: please provide a reference for "...as the N element is an important component of phosphatase proteins.

Reply:

In the revised manuscript, we have provided the references that support this argument (see

line 176-179 in Manuscript text (Revised Document with Track Changes)). This reference indicates that the nitrogen content in the phosphatase molecule is between 8% and 32%, which is sufficient to demonstrate that "the N element is an important component of phosphatase proteins". The references supporting this argument are as follows:

Treseder, K.K., Vitousek, P.M., 2001. Effects of soil nutrient availability on investment in acquisition of N and P in Hawaiian rain forests. *Ecology* 82, 946–954.

➤ **Comment 7:**

Line67, 74: nitrogen > N

Reply:

Based on your comments, we have corrected these irregular words.

➤ **Comment 8:**

Line64-76: In fact, these meta-analyses with large datasets also highlight the importance of case studies that utilize large data volumes.

Reply:

Yes, we highly agree with your comment. The viewpoint of this comment is also one of the bases for our research work to conduct experiments with a large sample size.

➤ **Comment 9:**

Line112-115: The authors have not fully elucidated why 'the nutrient limitation in early soil ecosystems is distinctive.

Reply:

The elaboration on this has been added to the revised manuscript text(see line 236-243 in Manuscript text (Revised Document with Track Changes)), as follows:

Traditional theory(Marklein and Houlton, 2012) posits that organisms are expected to allocate their resource reserves toward strategies that increase the acquisition of the most limiting resources — thus moving toward a state where all resources simultaneously limit productivity and growth. The observed N-P co-limitation patterns in developed ecosystems appear consistent with this framework. However, this paradigm fails to explain the successional trajectory of nascent soil systems: Despite an initial severe lack of available nutrients, these systems progressively evolve into productive forests(Zhou et al., 2013; Buma et al., 2017). This evolution suggests the presence of positive feedback mechanisms, indicating a directional development that is incompatible with static co-limitation models.

Based on your comments and in combination with the opinions of other reviewers, we have improved the writing idea of this introduction and rewritten this part of the manuscript.

References:

- Buma, B., Bisbing, S., Krapek, J., Wright, G., 2017. A foundation of ecology rediscovered: 100 years of succession on the William S. Cooper plots in Glacier Bay, Alaska. *Ecology* 98, 1513–1523. doi:10.1002/ecy.1848
- Marklein, A.R., Houlton, B.Z., 2012. Nitrogen inputs accelerate phosphorus cycling rates across a wide variety of terrestrial ecosystems. *New Phytologist* 193, 696–704. doi:10.1111/j.1469-8137.2011.03967.x
- Zhou, J., Wu, Y., Priezel, J., Bing, H., Yu, D., Sun, S., Luo, J., Sun, H., 2013. Changes of soil phosphorus speciation along a 120-year soil chronosequence in the Hailuoguo Glacier retreat area (Gongga Mountain, SW China). *Geoderma* 195–196, 251–259. doi:https://doi.org/10.1016/j.geoderma.2012.12.010

➤ **Comment 10:**

Line 353: please specify why 10 g m⁻² year⁻¹ was adopted?

Reply:

The reason for using this addition amount in the experiment is as follows: Taking into account the high permeability of the soil in the early stage and referring to the amounts of N and P added in the research conducted by Zhao et al. (2014),; Li et al. (2015),; and Tian et al.(2016).

References:

- Li, J., Li, Z., Wang, F., Zou, B., Chen, Y., Zhao, J., Mo, Q., Li, Y., Li, X., Xia, H., 2015. Effects of nitrogen and phosphorus addition on soil microbial community in a secondary tropical forest of China. *BIOLOGY AND FERTILITY OF SOILS* 51, 207–215. doi:10.1007/s00374-014-0964-1
- Tian, J., Wei, K., Condron, L.M., Chen, Z., Xu, Z., Chen, L., 2016. Impact of land use and nutrient addition on phosphatase activities and their relationships with organic phosphorus turnover in semi-arid grassland soils. *Biology and Fertility of Soils* 52, 675–683. doi:10.1007/s00374-016-1110-z
- Zhao, J., Wang, F., Li, J., Zou, B., Wang, X., Li, Z., Fu, S., 2014. Effects of experimental nitrogen and/or phosphorus additions on soil nematode communities in a secondary tropical forest. *SOIL BIOLOGY & BIOCHEMISTRY* 75, 1–10. doi:10.1016/j.soilbio.2014.03.019

➤ **Comment 11:**

Line131: AP? Write out the full name.

Reply:

AP is the abbreviation of "available phosphorus". We have supplemented the full name of AP (see line 259 in Manuscript text (Revised Document with Track Changes)).

➤ **Comment 12:**

Line125-135: The author provides a description of the soil physicochemical properties under natural conditions for the early soils in the study area, which is excellent. However, I believe that information on phosphorus minerals in the soil of the study area (such as apatite) should also be provided. This information is crucial for extending the research findings to mature ecosystems.

Reply:

We would like to express our sincere gratitude for your guidance. Based on your instructions, we have incorporated information on phosphorus minerals in the research area's soil into the manuscript (see line 503-504 in Manuscript text (Revised Document with Track Changes)), which are detailed as follows: According to previous studies conducted by our team, the dominant phosphorus mineral in the study area is apatite, accounting for approximately 2% of the total soil mineral mass(Zhou et al., 2016, 2018).

References:

- Zhou, J., Bing, H., Wu, Y., Sun, H., Wang, J., 2018. Weathering of primary mineral phosphate in the early stages of ecosystem development in the Hailuoguo Glacier foreland chronosequence. *European Journal of Soil Science* 69, 450–461. doi:10.1111/ejss.12536
- Zhou, J., Bing, H., Wu, Y., Yang, Z., Wang, J., Sun, H., Luo, J., Liang, J., 2016. Rapid weathering processes of a 120-year-old chronosequence in the Hailuoguo Glacier foreland, Mt. Gongga, SW China. *GEODERMA* 267, 78–91. doi:10.1016/j.geoderma.2015.12.024

➤ **Comment 13:**

Line136: “Highly significant?” Other sections also contain similar phrasing. I believe it is not appropriate to describe the statistical p-values of research results from highly variable environments using binary terms such as 'significant' or 'not significant.' This approach can mislead readers about the precision of the study results. I recommend adopting the 'Evidence language' method proposed by Stefanie Muff et al., 2022.

Reply:

We sincerely appreciate your suggestions. Following your advice, we have reformulated the degree of difference between the datasets according to the method proposed by Stefanie Muff et al.(2022). (see line 265-267 in Manuscript text (Revised Document with Track Changes))

References:

Muff, S., Nilsen, E.B., O’Hara, R.B., Nater, C.R., 2022. Rewriting results sections in the language of evidence. *Trends in Ecology & Evolution* 37, 203–210. doi:10.1016/j.tree.2021.10.009

➤ **Comment 14:**

Line140-141: In this study, each experimental treatment includes 90 replicates, which is quite remarkable for a field experiment. To justify the necessity of such a large number of replicates, I recommend demonstrating the variability of the data in the supplementary materials. This would provide readers with a reference for evaluating the credibility of previous field experiments, as many typically do not exceed 10 replicates. I believe this approach would greatly assist readers in assessing the reliability of past studies.

Reply:

Thank you for your appreciation of our experimental method. We also believe that this experiment will greatly assist readers in evaluating the reliability of previous studies in terms of sample size. According to your suggestion, we have included the coefficient of variation (CV) for the dataset in the supplementary materials (see Table S8 the statistical characteristics of the dataset.xlsx).

➤ **Comment 15:**

Line153: DOC? Write out the full name

Line192:LMWOA? Write out the full name

Reply:

We have supplemented these full name (see line 286, line 279 in Manuscript text (Revised Document with Track Changes)).

➤ **Comment 16:**

Line194: The Discussion section could be organized into several subsections with headings for clarity. For example, lines 195-236 could be headed 'Mechanism Underlying Nitrogen-Driven Phosphorus Availability in Early-Stage Soils'. Additionally, in the discussion, I recommend extending the discussion to speculate on the mechanisms related to nitrogen and phosphorus in mature soils (or vegetation-dominated soils) based on your data results. This would significantly enhance the depth of the discussion in this manuscript.

Reply:

Following your advice, we have organized the Discussion section into several subsections

with headings. In fact, we have extended the discussion to speculate on the mechanisms related to nitrogen and phosphorus in mature soils (see line 339-349, line 382-398, line 460-467 in Manuscript text (Revised Document with Track Changes)).

➤ **Comment 17:**

Line223:CK? Write out the full name

Line 240, 290, 296: mechanism > mechanisms; nitrogen and phosphorus > N and P; nitrogen addition > N addition.

Reply:

Based on your comments, we have corrected these irregular words.

➤ **Comment 18:**

Line314-316: The content of this sentence should be moved to the Discussion section. It should correspond with the description of phosphorus minerals in the Supplementary Materials.

Reply:

This sentence have been moved to the Discussion section.

➤ **Comment 19:**

Line350-356: Present the experimental design in tabular form and include it in the Supplementary Materials.

Reply:

Following your advice, we have showed the experimental design in tabular form and include it in the Supplementary Materials (see Table S1 in Supplementary materials).

Reply to Reviewer #3:

➤ Comment 1:

1. Lines 125-135 and Table 1: This section describes the natural state of the early soils. However, key details are missing, such as when and how the soils were sampled, the number of replicates, and whether the sampling occurred before, during, or after the three-year experiment. It is also unclear whether the natural state of the soils remained consistent over the three-year period without N and P addition.

Reply:

Thank you for pointing out the issue of insufficient detail in the description of sample information in our manuscript. According to your comments, we have supplemented the key details you pointed out in the "Methods" section of the manuscript (see line 572-582 in Manuscript text (Revised Document with Track Changes)). The added content is as follows:

In summary, the above-mentioned in-situ experimental method ensured that in July 2022, three years later, we collected a sufficient number of soil samples (i.e., 90 replicates for N addition treatments, 90 replicates for P addition treatments, and 90 replicates for CK treatments, totaling 270 soil samples). The in-situ experiment was concluded in July 2022, and soil columns (0 - 5 cm) in each plot were collected as soil samples. The collection method was as follows: For each plot, the tubular polyethylene bag enclosing the intact soil core was excavated as a whole, and the soil at a depth of 0 - 5 cm of this core was collected and placed into a sterile sample bag as one soil sample. Altogether, we collected 270 fresh soil samples. Each of them was sieved through a 2-mm mesh before being divided into two subsamples. One subsample was stored at 4 °C for the measurement of soil physicochemical characteristics, and the other subsample was stored at -80 °C for genes analysis.

The above supplementary content indicates the following: the time of soil sampling was June 2022; the method of sampling was to excavate the tube-shaped polyethylene bag enclosing the intact soil core for each plot, and collect soil from the 0-5 cm depth of this core into a sterile sample bag as one soil sample; the number of replicates was 90 for each treatment; and the sampling occurred three years after the experiment began, i.e., in June 2022.

Regarding the comment, "It is also unclear whether the natural state of the soils remained consistent over the three-year period without N and P addition," our response is as follows: In fact, we also collected natural soil samples (soils without any treatment, with 90 replicates) corresponding to the experimental treatments. However, these natural soil samples showed no significant differences in various metrics compared to the CK treatment samples. Considering that the CK treatment samples more effectively exclude the influence of other factors and more convincingly demonstrate the effects of the experimental treatments, and to reduce data redundancy in the manuscript for a more focused presentation of the research results, we did not include the natural soil data in the main text. Nevertheless, to address your concern and fully substantiate our findings, the natural soil data have been added to the supplementary materials (see Table S6 source data for natural soil.xlsx and Figure S2).

➤ Comment 2:

2. Lines 153-156 and Figure 2B: The cbbL gene copy number showed only minor changes, with slight differences between N and P additions, despite statistical significance. However, cbbL

gene copy number does not always correlate with C fixation activity. Direct measurements of C fixation rates or qPCR analysis of *cbbL* RNA transcripts would provide more robust evidence. Given the large variation in this study, small changes in *cbbL* gene abundance, and the lack of C fixation rate measurements, the conclusion that N and P additions increased C fixation may not be fully supported.

Reply:

Thank you for your valuable feedback on our research. We understand your perspective regarding the magnitude of changes in the *cbbL* gene copy numbers and have conducted an in-depth analysis accordingly. Indeed, according to our experimental data, the average *cbbL* gene copy number increased by 90.1% under N addition treatments and by 48.2% under P addition treatments compared to the CK treatment. Such increases are statistically significant and represent relatively large changes from both biological and large sample size (90 samples per treatment) perspectives, especially concerning soil microbial activity.

However, we also acknowledge that the actual environmental impacts and ecological significance of these increases may require more direct evidence for support, such as measurements of carbon fixation rates or qPCR analysis of *cbbL* RNA transcripts. We agree with your viewpoint that changes in gene copy numbers alone are insufficient to fully assess their potential impact on carbon fixation. To more accurately reflect the importance of this finding, we will further discuss the implications of these data and conduct additional experiments, namely measuring TOC and performing qPCR analysis of *cbbL* RNA transcripts (see Figure 2B, Table S3 source data for N addition treatment, Table S4 source data for CK treatment, Table S5 source data for P addition treatment), to confirm the significance of these changes.

➤ **Comment 3:**

Organic Carbon Data: Figure 2 and many places in the manuscript present only dissolved organic carbon (DOC) data, excluding total organic carbon (TOC). Since DOC represents only a fraction of the organic carbon fixed by microorganisms, TOC is a critical parameter. Without both TOC and DOC data, it is difficult to draw right conclusions.

Reply:

Thank you for your valuable feedback on our research. In accordance with your comments, we have measured the soil total organic carbon (TOC) and included the results in the revised manuscript(see Figure 2B, Table S3 source data for N addition treatment, Table S4 source data for CK treatment, Table S5 source data for P addition treatment).

➤ **Comment 4:**

4. Lines 157-167: Nitrogenase activity is a more reliable indicator of N₂ fixation than *nifH* gene copy number. It is well known that many bacteria/proteobacteria, possess the *nifH* gene but do not actively fix N₂. Some lack the complete set of genes required for nitrogenase function, while others, despite having the full gene set, exhibit no detectable N₂ fixation activity. I would suggest that gene abundance, such *cbbL* and *nifH* gene, may not be good indicators in this particular study.

Reply:

Thank you for your valuable comments and insightful suggestions on our paper. We fully agree with your viewpoint that nitrogenase activity is a more reliable indicator of nitrogen fixation

compared to the *nifH* gene copy number. We also concur with your opinion that not all bacteria/proteobacteria possessing the *nifH* gene can effectively carry out nitrogen fixation, which may be due to either lacking the complete set of genes required for nitrogenase function or showing no detectable nitrogen fixation activity despite having the full gene set. Indeed, our data (Figure 2C)—showing no significant changes in nitrogenase activity or *nifH* gene abundance under nitrogen addition treatments compared to the CK treatment—further support this viewpoint.

Regarding your suggestions, we would like to provide some clarification: In the research design of this study, we did adopt a multi-level assessment system for nitrogen fixation activity. In addition to analyzing the abundance of the *nifH* gene, we characterized nitrogen fixation activity through direct indicators such as ammonium nitrogen concentration and nitrogenase activity.

The combined use of these indicators ensures the reliability of our research findings. We acknowledge the limitations associated with *nifH* gene abundance, and thus, in our data analysis, we mainly used it as a supplementary reference indicator while focusing on the results obtained from direct measurements. We will further emphasize this point in the revised manuscript (see line 292-295 in Manuscript text (Revised Document with Track Changes)), clearly stating that Our study does not merely rely on the abundance data of the nitrogenase gene (*nifH*), but also makes use of ammonium nitrogen ($\text{NH}_4^+\text{-N}$) and nitrogenase activity to more accurately reflect the characteristics of the actual nitrogen fixation process (Fig. 2 C). Thank you again for your suggestions, which will help us improve the quality of our paper..

➤ **Comment 5:**

5. Lines 168-170, 219-220: The data show a decrease in pH alongside increased nitrate concentration and *amoA* gene abundance. To conclude that nitrification significantly contributed to the pH decrease in ammonium addition groups, additional experiments are needed to quantify the pH change due to nitrate generation from ammonium via nitrification in this study.

Reply:

In another manuscript of ours focusing on the soil nitrification processes in this study area (not yet published), we have already quantified the changes in soil pH resulting from the conversion of ammonium to nitrate via nitrification through incubation experiments. Please refer to the supplementary material (“Nitrification experiment” section) of the revised manuscript for details.

➤ **Comment 6:**

6. Lines 174-193: The two models presented are somewhat confusing due to numerous negative correlations. Ammonium-N in the N addition group represents residual ammonium in the soil, and available P (AP) represents residual P. Including these residual concentrations in the models may not be appropriate and could complicate interpretation.

Reply:

We are extremely grateful that you have carefully reviewed our paper in the midst of your busy schedule and put forward highly valuable comments. We attach great importance to the two key issues you pointed out: the confusion in understanding caused by the numerous negative correlations in the structural equation model, and the possible complication of interpretation due to the inclusion of the “residual” concentrations of ammonium-N and available phosphorus (AP) in the model. Here, we would like to provide you with a detailed and in - depth explanation.

First of all, regarding the numerous negative correlations in the model, we fully understand your doubts. In the general perception, if the absorption and release of available nutrient elements (e.g. ammonium-N and AP) in soil processes reach a dynamic equilibrium state, the relationships in the model are likely to be positive correlations. However, our research system has its uniqueness: From a data perspective, the results of sample analysis clearly show that the concentration of the added elements in the addition treatment is much higher than that in the control (CK) treatment (Figure 2A, C), indicating that the absorption and release of the added elements (ammonium-N or AP) by the soil processes have not reached a dynamic equilibrium. In terms of the experimental method, we adopt the approach of periodically adding equal amounts of nutrient elements to each replicate plot, which makes it highly unlikely for the soil processes in the plots to reach the above - mentioned dynamic equilibrium state. Since equal amounts of corresponding nutrient elements are added to the corresponding plots, and there is heterogeneity among the replicate plots (the reason why we set 90 replicates for each experimental treatment is to make full use of this heterogeneity), the rates at which the soil processes in each plot absorb and assimilate the added nutrient elements to produce "functional components" (such as the *cbbL* gene, nitrogenase, etc.) cannot be the same. Specifically, the soil in the plots with faster assimilation will produce more "functional components", and correspondingly, the remaining nutrient elements will be less; while the soil in the plots with slower assimilation will produce fewer "functional components", and the remaining nutrient elements will be relatively more. This difference is ultimately manifested as a negative correlation in the structural equation model we constructed, which is actually a scientific reflection of the real experimental situation. Therefore, in the context of our experiment, these negative correlations do not imply an inhibitory relationship between the indicators.

In addition, if we measure the relevant indicators after the soil processes in our experiment reach a dynamic equilibrium state, although the positive correlations presented between the indicators seem intuitive, when analyzing the driving effects of available nutrients on soil processes, it will become more complex and difficult to distinguish. Because a positive correlation means that the indicators drive each other, and it is very difficult to clearly identify which one drives the other under the background of dynamic equilibrium, which is extremely unfavorable for our in - depth exploration of the nitrogen - phosphorus coupling mechanism in the soil and the driving effects of available nutrients.

Secondly, regarding the issue of including the "residual" concentrations of ammonium nitrogen and available phosphorus in the model. In the field of soil ecosystem research, these "residual" concentrations are key indicators for measuring the soil nutrient status, and they intuitively reflect the nutrient content in the soil that is actually available for organisms to absorb and utilize and participate in soil biochemical reactions. In our research, the "residual" ammonium nitrogen and available phosphorus in the soil are not only the result manifestations after the experimental treatment but also important driving factors for subsequent soil processes. For example, the structural equation model based on the phosphorus - addition experiment shows that the ammonium nitrogen concentration is mainly directly affected by nitrogenase activity, indicating that the "residual" ammonium nitrogen in the soil is closely related to the nitrogen - fixation process in the soil; while the structural equation model based on the nitrogen - addition experiment shows that the available phosphorus concentration is mainly directly affected by pH and low-molecular-weight organic acids (LMWOA), which reflects the internal connection

between the “residual” available phosphorus in the soil and the soil chemical properties. Therefore, including them in the model helps us to more comprehensively and accurately reveal the coupling pathways between nitrogen and phosphorus in the soil and the driving mechanisms of soil processes.

We always regard the scientific nature and rigor of the paper as of utmost importance. Once again, thank you for your attention and suggestions. Your comments have been of great help to us in further improving the scientific nature and readability of the paper. We sincerely look forward to your further guidance so that we can present our research results to readers more accurately and clearly.

➤ **Comment 7:**

7. Lines 227-230: The manuscript discusses organic acids but does not quantify their contribution to pH reduction.

Reply:

According to your comment, we quantified the contributions of low molecular weight organic acids (LMWOA) and the ammonia - oxidation process (measured by NO_3^- -N, the product of the ammonia - oxidation process) to pH using partial Redundancy Analysis (partial RDA) based on the N addition treatment. The specific methods are as follows: taking NO_3^- -N as a control variable, we calculated the amount of pH variation explained by LMWOA. Similarly, taking LMWOA as a control variable, we calculated the amount of pH variation explained by NO_3^- -N. The results are shown in the following figure: the total amount of pH variation explained by LMWOA is 69% (13.4% + 55.6%), among which the net amount of pH variation explained by LMWOA is 13.4% (in the figure, L represents LMWOA, and N represents NO_3^- -N). These results have been added to the revised manuscript. Please refer to Fig. S4 in Supplementary Materials.

➤ **Comment 8:**

8. Lines 257-258: The claim of “a large number of C fixation genes” lacks comparative data, making it difficult to assess the validity of this statement.

Reply:

Thank you very much for your insightful comment. We greatly appreciate your attention to the details of our manuscript and the valuable feedback you have provided. You are absolutely right that the claim of “a large number of C fixation genes” in the original statement lacked comparative data, which could make it difficult for readers to assess the validity of this statement.

Consequently, we have revised the statement to provide more specific and comparable data (see line 401-402 in Manuscript text (Revised Document with Track Changes)). The revised statement is as follows: Additionally, a considerable amount of the *cbbL* gene involved in carbon fixation from microorganisms ($3.4 \times 10^4 - 1.1 \times 10^6$ copies/g in Fig. 2B, compared to the study by Tahon et al.) was detected in our soil samples. We believe that this revision provides a clearer and more accurate description of the findings in our study. Once again, thank you for your valuable comment, which has helped us improve the quality of our manuscript.

References:

Tahon et al.. Analysis of *cbbL*, *nifH*, and *pufLM* in Soils from the Sør Rondane Mountains, Antarctica, Reveals a Large Diversity of Autotrophic and Phototrophic Bacteria

➤ **Comment 9:**

9. Lines 261-262: The manuscript incorrectly states that C fixation provides energy for N₂ fixation. Photosynthesis converts light energy into chemical energy, which can support cellular processes, including N₂ fixation. However, C fixation itself refers to the conversion of inorganic carbon (e.g., CO₂) to organic carbon. This distinction should be clarified.

Reply:

We sincerely thank you for your valuable comments on the manuscript. The issue you raised is critically important, and we fully acknowledge that the original wording lacked precision. We sincerely apologize for the ambiguity in describing the essential nature of energy sources, which may have led to misinterpretation. To ensure scientific accuracy, we have revised the relevant section as follows:

Original statement:

"Another important reason for this opinion is that the N fixation process requires a large amount of energy, which mainly comes from C fixation processes such as photosynthesis."

Revised statement:

"Another important reason supporting this perspective is that N fixation requires substantial energy input. As demonstrated in previous studies (Dixon and Kahn, 2004; Wang et al., 2021; Zheng et al., 2020), organic carbon serves as a crucial energy source for nitrogen fixation processes, and this organic carbon is primarily derived from carbon fixation pathways, such as photosynthesis."

Rationale for revision:

According to previous research (Dixon and Kahn, 2004; Zheng et al., 2020; Wang et al., 2021), organic carbon serves as a crucial energy source for N fixation processes. For instance, Wang et al. (2021) proposed that "C availability alone may be a major predictor of free-living N fixation (FLNF) rates." This is because, unlike symbiotic organisms, heterotrophic diazotrophs must independently acquire energy, typically by oxidizing organic molecules released during organic matter decomposition or excreted by other organisms, to sustain the energy-intensive process of N fixation¹. Additionally, Dixon and Kahn (2004) showed that the N status-sensing PII proteins and the *nif*-gene regulation systems of some diazotrophs are coregulated by C and N signals (e.g., 2-oxoglutarate and glutamine) and that N signals override C signals to depress N fixation only in situations of N excess. Conversely, this finding implies that N fixation is promoted when C signals outweigh N signals. This conclusion is further supported by the work of Zheng et al. (2020), whose research demonstrated "increased FLNF rates in N-rich late-successional ecosystems compared with relatively N-poor early-successional ecosystems," and identified the C:N ratio as a key factor in explaining variations in FLNF rates.

Thank you once again for your meticulous attention to detail, which has significantly strengthened the clarity and rigor of our work.

References:

- Dixon, R., Kahn, D., 2004. Genetic regulation of biological nitrogen fixation. *Nature Reviews Microbiology* 2, 621–631. doi:10.1038/nrmicro954
- Wang, J., Wu, Y., Li, J., He, Q., Zhu, H., Bing, H., 2021. Energetic supply regulates heterotrophic nitrogen fixation along a glacial chronosequence. *Soil Biology and Biochemistry* 154, 108150. doi:10.1016/j.soilbio.2021.108150

Zheng, M., Chen, H., Li, D., Luo, Y., Mo, J., 2020. Substrate stoichiometry determines nitrogen fixation throughout succession in southern Chinese forests. *Ecology Letters* 23, 336–347. doi:10.1111/ele.13437

➤ **Comment 10:**

10. Lines 267-276: The proposed link between AP and available N via C fixation may not be valid. As noted earlier, C fixation does not provide energy for N₂ fixation, and the evidence for increased C fixation is weak (see comments on *cbbL* gene copy numbers and TOC). Additionally, P addition might promote N₂-fixing microorganisms that do not depend on available N, while other microorganisms remain N-limited. It should be noted that some N₂-fixing microorganisms, such as cyanobacteria, can perform both photosynthesis and N₂ fixation. Therefore, AP and directly contributes to N₂ fixation and C fixation may not be the link between AP and N₂ fixation.

Reply:

We sincerely apologize for the misunderstandings caused to you and readers by our oversimplified expressions, insufficient data, and previous inaccurate description of the relationship between carbon fixation and energy supply (which has been revised in our response to Comment 9). To rectify these errors and enhance the scientific rigor of our research, we have supplemented TOC measurements and *cbbL* RNA reverse-transcribed qPCR data (as detailed in our responses to Comments 2 and 3). Accordingly, we have rewritten and modified the content from Lines 417-432 (in Manuscript text (Revised Document with Track Changes)) as follows:

To elucidate the mechanism by which P drives N availability in early-stage soils, we constructed a structural equation model (SEM) (Fig. 3B). In this model, whereas no significant relationship was observed between available phosphorus (AP) and *nifH* gene abundance, several significant pathways (such as the associations between AP and nitrogenase activity, AP and DOC, DOC and the *nifH* gene, DOC and nitrogenase activity) combined with the findings that AP addition treatment (Fig. 2B) significantly increased DOC concentration, TOC concentration, and *cbbL* RNA reverse-transcribed copy numbers, collectively indicate that organic C derived from C fixation processes serves as the critical link between AP and N availability. Besides, these significant pathways also indicate that AP may preferentially regulate nitrogenase activity and energy acquisition (i.e., organic C production through C fixation) rather than simply increasing *nifH* gene copy numbers. Moreover, previous studies (Dixon and Kahn, 2004; Wang et al., 2021; Zheng et al., 2020) have established organic carbon as a crucial energy source for N fixation. Therefore, our findings emphasize that AP in early-stage soils enhances N fixation by stimulating organic C production through C fixation processes, thereby promoting N availability. The results and analyses above indicate that integrating C-related processes into the interaction framework of the N-P cycle plays a key role in refining the existing N-P coupling theory.

If our current revisions still fail to meet your requirements, we sincerely hope that you can provide us with further feedback and guidance.

References:

- Wang, J. et al. Energetic supply regulates heterotrophic nitrogen fixation along a glacial chronosequence. *Soil Biology and Biochemistry* 154, 108150 (2021)..
- Dixon, R. & Kahn, D. Genetic regulation of biological nitrogen fixation. *Nature Reviews Microbiology* 2, 621–631 (2004).

Zheng, M., Chen, H., Li, D., Luo, Y. & Mo, J. Substrate stoichiometry determines nitrogen fixation throughout succession in southern Chinese forests. *Ecology Letters* 23, 336–347 (2020).

Reply to Reviewer #4:

➤ Comment 1:

Sun et al. perform a nutrient addition experiment (N,P) in early seral habitat in an attempt to understand available N & P dynamics. If I understand their experimental design correctly then they tested the effect of three treatments on various soil properties and N cycling functional genes with 30 randomly assigned plots per treatment. Each plot replication contained a soil core incubated in a plastic bag to ensure that additions of nutrients were kept in contact with the soil and didn't leach away. Post-incubation the soil cores were assayed. N&P additions were provided as a solution periodically sprayed onto the top of each bagged soil core. It isn't clear that a similar amount of water was sprayed onto the control cores to control for moisture additions (which might possibly affect organic matter if water limited). The authors probably need to elaborate on how the nutrient addition amounts were carefully controlled with a sprayer (e.g. make and model of sprayer, calibration info. etc.) in place of the more conventional approach of pouring a fixed amounts of solution into bags.

Reply:

Thank you for your comments. Your feedback has helped us recognize that our description of the research methodology for "Field in-situ experiments and soil sample collection" (Lines 329-356) was unclear. For instance, while we allocated 90 plots per treatment as replicates, our original wording inadvertently created the impression that only 30 plots were assigned to each treatment. Additionally, our description lacked critical details you highlighted, such as the brand and model of the sprayer used, calibration protocols, and related specifications. To ensure methodological clarity, we have comprehensively revised this section in accordance with your suggestions as follows (see line 535-582 in Manuscript text (Revised Document with Track Changes)):

We conducted a large-scale, three-year in situ nitrogen (N) and phosphorus (P) addition experiment on early-stage soils in the Hailuoguo glacier retreat area, with over 270 plots established. This experiment is particularly significant (beyond its focus on "early-stage soils") because the early soils in this study area can develop into a soil ecosystem with approximately 25% pioneer plant cover within about six years. Consequently, for the early-stage soil ecosystems in this region—characterized by the absence of vegetation and rapid development—a three-year study period can be regarded as "long-term" study. Thus, our experiment can be defined as a "long-term in situ nitrogen and phosphorus addition experiment targeting early-stage soils."

In May 2019, a substantial number of plots (>270) were established in the study area to conduct the in situ N and P addition experiments. Each plot measured 0.3 m × 0.3 m, with spacing between plots exceeding 4 m. The experiment included three treatments: nitrogen addition (+N), phosphorus addition (+P), and a control (CK). Given the complex microtopography, frequent geological disturbances (e.g., small-scale debris flows and landslides), and significant microenvironmental heterogeneity in the study area, to enhance the accuracy of experimental data and mitigate data loss from potential plot damage caused by natural disasters, we randomly allocated over 90 plots per treatment as replicates (i.e., the total number of plots assigned to the three treatments exceeded 270). This design also ensured the collection of 90 replicate samples per treatment (a total of 270 samples for the three treatments) by the end of the experiment.

For each of the over 270 plots, the experimental setup involved the following steps:

Soil column extraction: Using an undisturbed soil sampler (inner diameter: 10 cm), we extracted an intact soil column (dimensions: 15 cm height × 10 cm diameter, preserving its natural structure) from each plot. The soil column was placed into a tubular polyethylene bag to ensure thorough interaction between subsequently added N (or P) and the soil column, prevent lateral leaching of N(or P) and avoid cross-contamination between plots.

Reinstallation: The soil column was then returned to its original location (Fig. S1).

Nutrient addition protocol:

Given the high permeability of early-stage soils and referencing prior studies, N and P were not applied in a single dose. Instead, the following protocol was implemented:

N addition: Starting in May 2019, N was applied to +N treatment plots at a rate of 10 g N m⁻² year⁻¹. Specifically, NH₄Cl solution (concentration: 10.00 mg/ml) was evenly sprayed onto the soil column surface (area: 78.5 cm²) using a sprayer (brand: Lizhen; capacity: 100 ml; spray volume per press: 0.132 ml). Each application involved 5 ml of solution (achieved via 38 presses), repeated every two months.

P addition: Similarly, +P treatment plots received 10 g P m⁻² year⁻¹ via NaH₂PO₄ solution (concentration: 10.13 mg/ml), applied identically using the same sprayer (5 ml per application, 38 presses).

CK treatment: For CK plots, pure water was sprayed using the same method.

In summary, the above-mentioned in-situ experimental method ensured that in July 2022, three years later, we collected a sufficient number of soil samples (i.e., 90 replicates for N addition treatments, 90 replicates for P addition treatments, and 90 replicates for CK treatments, totaling 270 soil samples). The in-situ experiment was concluded in July 2022, and soil columns (0 - 5 cm) in each plot were collected as soil samples. The collection method was as follows: For each plot, the tubular polyethylene bag enclosing the intact soil core was excavated as a whole, and the soil at a depth of 0 - 5 cm of this core was collected and placed into a sterile sample bag as one soil sample. Altogether, we collected 270 fresh soil samples. Each of them was sieved through a 2-mm mesh before being divided into two subsamples. One subsample was stored at 4 °C for the measurement of soil physicochemical characteristics, and the other subsample was stored at -80 °C for genes analysis.

➤ **Comment 2:**

In the Introduction the authors cite numerous meta-studies on related topics and appear to use these meta-studies as motivation for the current study. This is somewhat confusing. A central problem with the study's presentation is that the authors don't clearly distinguish their study from prior work or adequately highlight the novelty of their study relative to prior work. Their study is likely somewhat novel because they applied nutrients to an area undergoing primary succession unlike most studies but exactly why that is needed or necessary isn't clear. Their study appeared to mainly be motivated by the fact that three similar meta-studies reached divergent conclusions. However, this isn't convincing motivation for doing yet another nutrient addition field experiment. In fact, the reality that there are now multiple meta-studies on the topic is an indication that the topic is well studied. So the motivation for the present study was a bit unclear and was not framed well. Further, this had me questioning why the authors submitted the current study to a journal that expects studies to make important advances within a field. Nutrient addition studies are possibly the most common field manipulation in ecological studies.

Reply:

We sincerely appreciate your constructive feedback. We agree that the original introduction inadequately highlighted the unique rationale for our study. In accordance with your comment, we have revised and rewritten the introduction (see line 146-250 in Manuscript text (Revised Document with Track Changes)). In the revised version, we have:

Clarified the unresolved scientific gap: Emphasized that prior meta-analyses reached conflicting conclusions due to oversimplified single-process interpretations (e.g., focusing solely on phosphatase activity or N fixation without integrating cross-process interactions and energy constraints).

Explicitly justified the experimental context: Highlighted how early-stage primary succession soils (1) minimize confounding biotic/abiotic complexities (e.g., lack of plant dominance, simplified microbial communities), enabling clearer mechanistic insights into N-P coupling, and (2) represent a critical yet understudied phase where traditional stoichiometric theories may behave differently due to extreme nutrient limitation.

Reframed novelty: Stressed that our study uniquely integrates C process and nitrification process into the N-P coupling framework—a dimension overlooked in prior nutrient addition experiments but critical for explaining context-dependent responses.

These revisions aim to demonstrate that our work advances ecological theory by addressing mechanistic ambiguities in a distinct environmental context, rather than merely replicating prior nutrient addition approaches. Especially, despite the presence of multiple meta-analysis studies on this topic of N and P, their inconsistent conclusions indicate that this topic has not been well studied, to the extent that its essential laws have not been delved into deeply.

➤ Comment 3:

Many of the topics presented in the paper on N and especially P cycling were only superficially explored in the Introduction. I expected these topics to be well developed and that they would lead the reader toward a researchable problem. Sentences often lacked citations or mainly cited meta-studies. Overall, their study appeared to be mainly inspired by meta-studies.

Reply:

We sincerely appreciate the reviewer's insightful feedback regarding the development of key concepts in the Introduction. We have substantially revised this section through the following specific improvements:

Revised mechanistic discussion paragraphs (now spanning Lines 185-204 and 215-226 in Manuscript text (Revised Document with Track Changes)) detailing:

- a) The mediation of carbon processes in N-P interactions
- b) The possibility of inhibition of phosphorus activation processes by nitrification processes
- c) The limitations of using a single process indicator for exploring the complete mechanism of N-P coupling processes
- d) The necessity of integrating other soil processes into the N-P coupling framework

Incorporated 16 new citations spanning:

- a) The concept of nitrogen-phosphorus co-limitation and coupling (Du et al., 2020; Zhou et al., 2021, Elser et al., 2007; Čapek et al., 2018, and so on)
- b) Carbon-driven studies of phosphorus availability processes (Spohn and Kuzyakov, 2013;

Heuck et al., 2015)

- c) The acidification effects of nitrogen (Zhou et al., 2014)
- d) Energy sources for nitrogen fixation processes (Davies and Friedlingstein, 2020)

Problem Articulation

Developed a dedicated gap analysis framework through:

- (i) Triangulation of conflicting meta-analysis findings (Lines 179-185 contrasting Marklein and Houlton, 2012 vs. Gou et al., 2024 vs. Chen et al., 2023)
- (ii) Identification of missing mechanistic processes (C-P-N cross-talk in Lines 185-192; energy mediation in 215-226)
- (iii) Context specificity of early soil ecosystem (Lines 230-243 contrasting mature vs. pioneer ecosystems)

Citation Strategy Overhaul

In the introduction, we have supplemented and provided corresponding citations for each sentence, except for those expressing the authors' viewpoints and logical arguments.

We have also increased the proportion of experimental and case study investigations, including:

- a) three studies on nitrogen-phosphorus co-limitation;
- b) five studies on nitrogen-phosphorus processes;
- c) three studies on the linkage of carbon processes with nitrogen-phosphorus processes;
- d) two studies on the energy sources for nitrogen fixation processes.

For detailed modifications, please see the introduction section (Line 146-250 in Manuscript text (Revised Document with Track Changes)) of the revised manuscript.

References:

- Davies, B.T., Friedlingstein, P., 2020. The Global Distribution of Biological Nitrogen Fixation in Terrestrial Natural Ecosystems. *Global Biogeochemical Cycles* 34, e2019GB006387. doi:10.1029/2019GB006387
- Du, E., Terrer, C., Pellegrini, A.F.A., Ahlström, A., Van Lissa, C.J., Zhao, X., Xia, N., Wu, X., Jackson, R.B., 2020. Global patterns of terrestrial nitrogen and phosphorus limitation. *Nature Geoscience* 13, 221–226. doi:10.1038/s41561-019-0530-4
- Chen, Y., Xia, A., Zhang, Z., Wang, F., Chen, J., Hao, Y., Cui, X., 2023. Extracellular enzyme activities response to nitrogen addition in the rhizosphere and bulk soil: A global meta-analysis. *Agriculture, Ecosystems & Environment* 356, 108630. doi:10.1016/j.agee.2023.108630
- Čapek, P., Manzoni, S., Kaštovská, E., Wild, B., Diáková, K., Bárta, J., Schneckner, J., Biasi, C., Martikainen, P.J., Alves, R.J.E., Guggenberger, G., Gentsch, N., Hugelius, G., Palmtag, J., Mikutta, R., Shibistova, O., Urich, T., Schleper, C., Richter, A., Šantrůčková, H., 2018. A plant–microbe interaction framework explaining nutrient effects on primary production. *Nature Ecology & Evolution* 2, 1588–1596. doi:10.1038/s41559-018-0662-8
- Elser, J.J., Bracken, M.E., Cleland, E.E., Gruner, D.S., Harpole, W.S., Hillebrand, H., Ngai, J.T., Seabloom, E.W., Shurin, J.B., Smith, J.E., 2007. Global analysis of nitrogen and phosphorus limitation of primary producers in freshwater, marine and terrestrial ecosystems. *Ecology Letters* 10, 1135–1142.
- Gou, X., Ren, Y., Qin, X., Wei, X., Wang, J., 2024. Global patterns of soil phosphatase responses to nitrogen and phosphorus fertilization. *Pedosphere* 34, 200–210. doi:10.1016/j.pedsph.2023.06.011
- Heuck, C., Weig, A., Spohn, M., 2015. Soil microbial biomass C:N:P stoichiometry and microbial use of organic phosphorus. *Soil Biology and Biochemistry* 85, 119–129. doi:10.1016/j.soilbio.2015.02.029
- Marklein, A.R., Houlton, B.Z., 2012. Nitrogen inputs accelerate phosphorus cycling rates across a wide variety of

terrestrial ecosystems. *New Phytologist* 193, 696–704. doi:10.1111/j.1469-8137.2011.03967.x

Spohn, M., Kuzyakov, Y., 2013. Phosphorus mineralization can be driven by microbial need for carbon. *Soil Biology and Biochemistry* 61, 69–75. doi:10.1016/j.soilbio.2013.02.013

Zhou, J., Li, X., Peng, F., Li, C., Lai, C., You, Q., Xue, X., Wu, Y., Sun, H., Chen, Y., Zhong, H., Lambers, H., 2021. Mobilization of soil phosphate after 8 years of warming is linked to plant phosphorus - acquisition strategies in an alpine meadow on the Qinghai - Tibetan Plateau. *Global Change Biology* 27, 6578 - 6591. doi:10.1111/gcb.15914

Zhou, J., Xia, F., Liu, X., He, Y., Xu, J., Brookes, P.C., 2014. Effects of nitrogen fertilizer on the acidification of two typical acid soils in South China. *Journal of Soils and Sediments* 14, 415–422. doi:10.1007/s11368-013-0695-1

➤ **Comment 4:**

The writing was often vague and topics were not well developed. Below are many suggestions to help improve the writing.

L1 “Insights into the driving effects between nitrogen and phosphorus”- wordy & vague.

Reply:

Based on your feedback and the full revised manuscript, we have updated the title to: "Carbon-mediated co-amplification of N and P mobilization in proglacial soils: Large-scale experimental evidence of nutrient synergy". We believe this revision achieves a standard of conciseness and clarity.

➤ **Comment 5:**

L7 “But, due to the complexity of processes in mature ecosystems, a variety of inconsistent conclusions have been drawn.” – Authors then go on to mention how studying primary successional systems may help to improve understanding of mature ecosystems. This seems like a long shot. I think the authors are trying to say, “look scientists have spent a lot of time studying mature ecosystems and barely understand what is going on. So we are motivated to study these processes during primary succession.” However, it seems more direct to say while mature ecosystems have been extensively studied little is known about X in newly developing ecosystems. [or something similar] I think you are mostly trying to identify a gap in understanding. And primarily citing meta-studies fails to show that problem.

Reply:

We sincerely appreciate your insightful comments and fully agree with your assessment. We apologize for the inadequate articulation of research rationale in our original introduction. Following your suggestions, we have thoroughly restructured the connections between previous studies and our research motivations, while explicitly clarifying the scientific novelty of this work.

The revised introduction now follows this logical framework:

- 1) We first identify the conceptual incompleteness in current understanding of nitrogen-phosphorus (N-P) interaction mechanisms within biogeochemical cycles.
- 2) Through systematic analysis of conflicting conclusions in existing literature (examining both N→P and P→N perspectives), we highlight persistent discrepancies in reported N-P coupling patterns.
- 3) Critical evaluation of methodological variations across studies reveals that failure to integrate ancillary soil processes (particularly carbon dynamics) into conventional N-P

coupling frameworks likely drives these inconsistencies.

- 4) Building on the unique advantages of studying early-stage soil ecosystems - their simplified processes minimize confounding factors while maintaining evolutionary trajectories toward mature systems - we establish our research objective: To decode fundamental N-P interaction mechanisms through carbon-mediated pathways using glacial forefield succession as a model system.

Regarding meta-analysis citations:

We deliberately incorporated meta-analytic approaches because they synthesize numerous localized case studies, providing conclusions with broader generalizability and enhanced credibility compared to individual case studies. This aligns with our goal of establishing universally applicable principles for N-P interaction mechanisms.

For detailed modifications, please see the introduction section of the Manuscript text (Revised Document with Track Changes).

➤ **Comment 6:**

L10 “initial processes of nitrogen and phosphorus and their interactions”- unclear. N & P aren't processes... Also, it isn't clear what you measured or why?.

Reply:

We concur with your assessment. Based on this feedback of yours and other comments, we have rewritten the abstract. Please refer to the abstract in the revised manuscript text.

➤ **Comment 7:**

L12 “N & P interactions”- This sounds like you want to study how the biogeochemistry of these two things interact. I suspect you actually have a different aim that isn't being described well here.

Reply:

Thank you for this astute observation. We acknowledge that the phrasing "N and P interactions" in its original form lacked precision in conveying the specific mechanistic focus of our study.

We understand that it might give the initial impression that the focus is solely on the general biogeochemical interaction of nitrogen and phosphorus. However, my actual aim is to delve into the specific and fundamental pathways through which these two elements interact in early - stage soils. In these soils, the lack of vegetation and simpler soil processes provide a unique and less - complicated environment. This allows for a more clear - cut investigation of how N and P influence each other's cycling and availability, as well as their combined effects on the initial development of soil properties and potential for supporting early plant colonization. The simplicity of the soil processes in the early stage acts as a kind of "baseline" that can help us better understand the core mechanisms of their interaction without the interference of too many other complex factors that are present in more developed soils.

We realize now that the original phrasing might not have fully conveyed this specific research goal. Based on this feedback of yours and other comments, we have rewritten the abstract. Please refer to the abstract in the revised manuscript text.

Thank you again for your helpful feedback. It is greatly appreciated and will definitely

contribute to improving the quality of my paper.

➤ **Comment 8:**

L13 Another nutrient addition experiment... Is there a hypothesis?.

Reply:

Regarding the "Another nutrient addition experiment" mentioned in the comment, we searched the original manuscript five times and unfortunately did not find this content. We guess that you may be referring to the nitrogen addition experiment and phosphorus addition experiment conducted in our study. If that's the case, the hypotheses corresponding to these two experiments are as follows:

1. An increase in N availability will increase the ecosystem's demand for P, leading to a decrease in available P and thereby exacerbating P limitation.

2. Similarly, based on elemental stoichiometric theory, an increase in P availability will increase the ecosystem's demand for N, leading to a decrease in available N and thereby exacerbating N limitation.

However, after analyzing and reflecting on your comments and those of other reviewers, we have recondensed and summarized the purpose of this study: Our aim is to attempt to integrate additional soil processes such as the carbon process and the nitrification process into the theoretical framework of nitrogen-phosphorus coupling, rather than focusing solely on the nitrogen fixation and phosphorus solubilization processes. Based on this approach, we seek to explore and reveal the positive feedback mechanisms among nutrient elements during the early stages of soil formation, providing empirical insights for the development of a unified theory of nitrogen-phosphorus coupling.

➤ **Comment 9:**

L14 This sounds like a fishing expedition without a testable hypothesis.

Reply:

Thank you for your comment. We recognize that the expression was inappropriate and have removed it in the revised manuscript.

➤ **Comment 10:**

L15-16 This seems detached from logic (add stuff there should be more not less right) and readers aren't presented with counter point for mature systems in Abstract introduction... Please rework.

Reply:

Yes, you are right. Based on this feedback of yours and other comments, we have rewritten the abstract. Please refer to the abstract in the revised manuscript text..

➤ **Comment 11:**

L17 "soil carbon (C) fixation processes" do you mean "soil carbon sequestration"?.

Reply:

In this study, the "soil carbon (C) fixation processes" do not mean "soil carbon sequestration". These terms are defined as follows: Carbon fixation denotes the process by which organic molecules are formed from inorganic carbon (Lawson et al., 2022), while carbon sequestration

refers to the capture and ensuing storage of carbon in order to prevent it from being released into the atmosphere (Duku et al., 2011; Zhou et al., 2021). Some scholars further emphasize that carbon sequestration specifically involves capturing atmospheric carbon dioxide and storing it long-term in plants, soils, and oceans (Sedjo and Sohngen, 2012; Perez-Verdin et al., 2016). From these definitions, it is evident that carbon fixation constitutes merely a non-essential step within carbon sequestration. The carbon fixation process emphasizes the transformation of inorganic carbon to organic carbon, without emphasizing long-term storage to prevent atmospheric release. In this research, carbon fixation processes are characterized by the increase in organic carbon, which serves as both carbon and energy sources for biological metabolic processes such as nitrogen fixation and phosphorus solubilization, ultimately being released as carbon dioxide. Therefore, the carbon fixation processes discussed in this study are distinct from carbon sequestration processes.

References:

- Duku, M.H., Gu, S., Hagan, E.B., 2011. Biochar production potential in Ghana—A review. *Renewable and Sustainable Energy Reviews* 15, 3539–3551. doi:<https://doi.org/10.1016/j.rser.2011.05.010>
- Lawson, T., Emmerson, R., Battle, M., Pullin, J., Wall, S., Hofmann, T.A., 2022. Chapter 3 - Carbon fixation, in: Ruban, A., Foyer, C.H., Murchie, E.H. (Eds.), *Photosynthesis in Action*. Academic Press, pp. 31–58. doi:<https://doi.org/10.1016/B978-0-12-823781-6.00008-3>
- Perez-Verdin, G., Sanjurjo-Rivera, E., Galicia, L., Ciro Hernandez-Diaz, J., Hernandez-Trejo, V., Antonio Marquez-Linares, M., 2016. Economic valuation of ecosystem services in Mexico: Current status and trends. *Ecosystem Services* 21, 6–19. doi:[10.1016/j.ecoser.2016.07.003](https://doi.org/10.1016/j.ecoser.2016.07.003)
- Sedjo, R., Sohngen, B., 2012. Carbon Sequestration in Forests and Soils. *Annual Review of Resource Economics*. doi:<https://doi.org/10.1146/annurev-resource-083110-115941>
- Zhou, Y., Qin, S., Verma, S., Sar, T., Sarsaiya, S., Ravindran, B., Liu, T., Sindhu, R., Patel, A.K., Binod, P., Varjani, S., Singhania, R.R., Zhang, Z., Awasthi, M.K., 2021. Production and beneficial impact of biochar for environmental application: A comprehensive review. *Bioresource Technology* 337. doi:[10.1016/j.biortech.2021.125451](https://doi.org/10.1016/j.biortech.2021.125451)

➤ **Comment 12:**

L20 “P solubilization” do you mean “P addition”? Also, I wonder if this is backwards. You added nutrients so isn't possible that nutrient additions accelerate various processes that might have been limited by initial soil nutrient levels? What about co-limitation?.

Reply:

We apologize for any lack of clarity in our manuscript that may have led to your questions. Please allow us to address your concerns as follows:

Firstly, in our study, P solubilization does not equate to P addition. In fact, P addition simulates the outcome of the P solubilization process, that is, it improves the availability of P. In order to highlight and amplify the effects caused by the outcome of the P solubilization process (such as the effects on the N fixation process and the C fixation process) for the purpose of measurement, we carried out the operation of P addition. Of course, the added bioavailable P may accelerate or inhibit other processes. According to our experimental data, the added bioavailable P promoted both N fixation and C fixation, thereby confirming one aspect of nitrogen-phosphorus

co-limitation: P limitation.

We would also like to add that our N addition experiments revealed increased abundance of phosphate solubilization-related genes, elevated concentrations of organic acids, and enhanced C fixation. Thus, the experiment confirmed other aspect of nitrogen-phosphorus co-limitation: N limitation.

These experimental designs allowed us to isolate mechanistic drivers while still capturing the overall patterns of co-limitation.

At this point, you might be wondering: Why not conduct experiments with simultaneous N and P addition? To this, our response is that we did perform experiments with simultaneous N and P additions (though the data were not shown in the original manuscript), because in these experiments, since N and P were added simultaneously, we found it impossible to distinguish between the individual effects of N or P, and thus could not convincingly identify the pathways through which N affects P (or vice versa). However, the results indicated positive effects such as enhanced P availability, increased N availability, increased P solubilization-related gene abundances, higher concentrations of organic acids, and so forth, supporting the nitrogen-phosphorus co-limitation hypothesis. Nevertheless, proving the nitrogen-phosphorus co-limitation was not the focus of our study; rather, our interest lay in understanding the pathways through which N and P interact. Therefore, the data from the simultaneous N and P addition experiments were not included in the original manuscript.

To address your concern and fully substantiate our findings, the results of experiments with simultaneous N and P addition have been added to the supplementary materials (see Table S7 source data for the treatment with combined N and P addition.xlsx and Figure S2 in Supplementary materials).

We appreciate your valuable feedback and will incorporate these clarifications into the revised manuscript.

➤ **Comment 13:**

L22 Unclear the basis for this interpretation or modes of what? Passage feels like a word salad.

L23 same.

Reply:

We sincerely appreciate your critique regarding the lack of clarity in this section. We fully acknowledge that the original passage suffered from undefined terminology (e.g., ambiguous use of "modes") and insufficient linkage between assertions and supporting evidence, resulting in unintended obfuscation. Please accept our apologies for this oversight.

To address your concerns, we will implement the following revisions.

As a brief illustration, the restructured paragraph will now read:

Furthermore, based on previous research findings and the experimental data from this study, we explored and established a model depicting the driving pattern of P availability on N availability in early soils. Similarly, another mode was constructed for the driving pattern of N availability on P availability. These results emphasized the central role of carbon-related processes in mediating nitrogen and phosphorus coupling, thereby enhancing their availability in early soil environments. These models also reveal a pathway in early soil development where enrichment of one nutrient enhances the co-availability of another rather than intensifying nutrient limitation,

suggesting that synergistic N-P coupling dominates interactions between N and P bioavailability during early soil development. This provides an explanation for the drivers of positive succession in early soil ecosystems.

To incorporate all the modifications regarding the abstract, we have rewritten the abstract (see line 30-51 in Manuscript text (Revised Document with Track Changes)).

We are grateful for your rigorous scrutiny, which has significantly strengthened this manuscript. Should further clarifications be needed, we eagerly await your guidance.

➤ **Comment 14:**

L29 “The mobilization of nitrogen (N) and phosphorus (P) constitutes a fundamental and critical soil process, with the bioavailability dynamics of these elements being among the most ecologically significant outcomes in this process.” Wordy and redundant.

Reply:

In accordance with your comment, we have revised and rewritten the introduction (see line 146-250 in Manuscript text (Revised Document with Track Changes)).

➤ **Comment 15:**

L36 I would probably replace citations 1,2 with (Fay et al. 2015, Du et al. 2020)..

Reply:

Thank you very much for recommending these two papers. After carefully reading the two papers you recommended, we feel that they are more suitable for supporting the viewpoints in our paper than the originally cited ones. We have replaced the original references with the ones you recommended(see 711-712 in Manuscript text (Revised Document with Track Changes)).

References

Fay, P.A., Prober, S.M., Harpole, W.S., Knops, J.M.H., Bakker, J.D., Borer, E.T., et al., 2015. Grassland productivity limited by multiple nutrients. *Nature Plants* 1. doi:10.1038/NPLANTS.2015.80

Du, E., Terrer, C., Pellegrini, A.F.A., Ahlstrom, A., van Lissa, C.J., Zhao, X., Xia, N., Wu, X., Jackson, R.B., 2020. Global patterns of terrestrial nitrogen and phosphorus limitation. *Nature Geoscience*. doi:10.1038/s41561-019-0530-4

➤ **Comment 16:**

L36 “Most notably, in terms of mechanism, the prevalence of N and P co-limitation hinted strongly the couplings between these two nutrients in ecosystems².”- wordy & confusing.

Reply:

Thank you for your feedback. Based on your suggestions, we have revised and rewritten the expression to improve clarity and conciseness. The updated version (see line 149-155 in Manuscript text (Revised Document with Track Changes)), along with its context, is provided below:

Empirical evidence demonstrates that across various ecosystems, plant growth exhibits comparable increments following the addition of either N or P, which suggests the prevalence of N and P co-limitation^{5,8}. Moreover, given this fact that organisms require elements in stoichiometric proportion and are expected to take up and recycle nutrients in ratios that maintain balanced nutrition^{9,10}, some scholars² considered that such co-limitation profoundly suggests an intrinsic

coupling of N and P through soil biological processes, highlighting the interconnected dynamics of these nutrient cycles within terrestrial environments.

➤ **Comment 17:**

L38 “The establishment and maintenance of this coupling relationship cannot be separated from the most active components (i.e., available N and P).” – This doesn’t make sense to me. My impression is that N is highly labile while available P is not and sorbs to numerous soil components (clay, Ca, Calcium carbonate, Fe, Al). The next sentence hints at some of these issues but seems to ignore how fundamentally different these two nutrient behave in the soil solution and interact with other things.

Reply:

We sincerely appreciate your insightful comment regarding the distinct behaviors of nitrogen (N) and phosphorus (P) in soil systems. You is absolutely correct in highlighting the fundamental differences in their mobility and interaction with soil components. Below, we clarify our original intent and propose revisions to address this concern:

We agree that the original statement oversimplified the dynamics of available N and P. As you rightly point out, mineral N (e.g., NH_4^+ , NO_3^-) is highly labile and prone to leaching or volatilization, while available P (e.g., H_2PO_4^- , HPO_4^{2-}) exhibits strong sorption to soil constituents (e.g., Fe/Al oxides, carbonates, clays) and forms stable complexes. This distinction is critical to their roles in nutrient coupling.

Our intention was to emphasize that both nutrients—despite their contrasting behaviors—are essential drivers of biogeochemical coupling in the studied system. For instance, N availability regulates microbial P mineralization (via phosphatase production), while P limitation can constrain N fixation (Vitousek et al., 2010). Although available N and available P have many differences, as you mentioned that available N is highly labile while available P is stable and easily fixed by soil components, they share a common characteristic, which is that they are easily utilized by organisms (i.e., they have high biological availability), and they can affect each other's biological availability when they are absorbed by organisms and participate in biological processes. For example, the addition of available N can enhance biological processes such as nitrification, acidify the soil, and solubilize the available P fixed by soil components; the addition of available P can promote biological N fixation and increase the level of available N in the soil. Obviously, these processes illustrate the coupling effect between available N and available P. However, we recognize that the original phrasing failed to explicitly address their divergent retention mechanisms.

Therefore, considering your comments and the logical context as a whole, we have decided to remove this expression. we have revised and rewritten the introduction (see line 146-250 in Manuscript text (Revised Document with Track Changes))

References

Vitousek, P.M., Porder, S., Houlton, B.Z., Chadwick, O.A., 2010. Terrestrial phosphorus limitation: mechanisms, implications, and nitrogen–phosphorus interactions. *Ecological Applications* 20, 5–15. doi:10.1890/08-0127.1

➤ **Comment 18:**

L41 This statement feels a bit superficial and doesn't include citations. P is incredibly immobile (Jungk 2002) and that should probably be noted and possibly differentiated from N. Also, it seems like 10x more research has been conducted on N..

Reply:

Through your comments, we have recognized that our statement did not effectively convey our research intent to the readers. Therefore, we have removed this statement and reorganized the content to rewrite this section(see line 146-167 in Manuscript text (Revised Document with Track Changes)).

➤ **Comment 19:**

L44 phosphatase activity concerns mineralization of organic matter. In early seral environments, there is little OM. I would suspect that other processes (weathering, P solubilizing bacteria) are likely more important for P mobilization.

Reply:

You are correct. In our research area, due to the lack of vegetation producing organic matter, the organic matter content in these early soils is very low. Previous research findings(Prietz et al., 2013; Zhou et al., 2018) in this area indicate that mineral phosphorus (i.e., inorganic phosphorus) is the absolute primary component of total phosphorus in these early soils (i.e., inorganic phosphorus is almost equal to total soil phosphorus, while the concentration of organic phosphorus is below the detection limit). Therefore, the mineralization of organic phosphorus is not a major pathway for phosphorus mobilization in these early soils. Instead, bioweathering involving phosphate-solubilizing bacteria is the main pathway for phosphorus mobilization in these early soils. Accordingly, in this study, we use indicators related to the dissolution of mineral phosphorus through bioweathering (such as pH, organic acids, available phosphorus concentration) to investigate the process of phosphorus mobilization.

References:

- Prietz, J., Duemig, A., Wu, Y., Zhou, J., Klysubun, W., 2013. Synchrotron-based P K-edge XANES spectroscopy reveals rapid changes of phosphorus speciation in the topsoil of two glacier foreland chronosequences. *Geochimica ET Cosmochimica Acta* 108, 154–171. doi:10.1016/j.gca.2013.01.029
- Zhou, J., Bing, H., Wu, Y., Sun, H., Wang, J., 2018. Weathering of primary mineral phosphate in the early stages of ecosystem development in the Hailuoguo Glacier foreland chronosequence. *European Journal of Soil Science* 69, 450–461. doi:10.1111/ejss.12536

➤ **Comment 20:**

L44-45 It seems most of your citations are for nutrient additions. Are these realistic and/or good ways for understanding processing affecting nutrient availability in most natural systems? It seems like a crude tool, especially for understanding P availability which can be sorbed to various soil particles (Jones et al. 2013b).

Reply:

We sincerely thank the reviewer for raising this critical point regarding the ecological realism of nutrient addition experiments, particularly in the context of phosphorus (P) dynamics influenced by soil-particle interactions (Jones et al. 2013b). We fully acknowledge that nutrient additions represent a simplified experimental approach and agree that sorption

processes—especially for P—may limit direct extrapolation to highly weathered or mineralogically complex soils. However, we propose that our methodology is both appropriate and informative for the specific context of early-stage soil ecosystems. Below, we clarify the rationale for our methodology while integrating the reviewer's concerns into our revised discussion.

1) Unique substrate characteristics:

The study area consists of freshly exposed rock following glacial retreat. Strictly speaking, the early-stage soils studied are composed of rock debris particles, which are predominantly made up of primary minerals (with minimal clay minerals) (Yang et al., 2015; Sun et al., 2022) and exhibit very weak adsorption of available phosphorus. It is important to emphasize that these early-stage soils are very young soils, not highly weathered soils with strong adsorption capacities.

2) Methodological superiority:

Although the nutrient addition method is simple, it allows for minimal introduction of confounding factors other than the variables under investigation. This ensures that the factors set in the experiment can independently explain the results. Additionally, the method is straightforward to implement and has low maintenance costs, making it well-suited for in situ field studies on nutrient availability effects.

3) Scientific novelty:

Currently, the nutrient addition method has been widely applied in various ecological environments (over 7000 studies since 2015, Web of Science search: "nutrient addition" AND "experiment"). This not only indicates that the nutrient addition method has a comprehensive dataset for generalization and synthesis, but also demonstrates its effectiveness as a research method—otherwise, it would not have been adopted by so many studies. However, despite the widespread use of nutrient addition in research, very few studies have applied this method to early-stage soil ecosystems, highlighting one aspect of the significance of our work.

References:

- Jones, M.P., Webb, B.L., Jolley, V.D., Vickery, M.D., Buck, R.L., Hopkins, B.G., 2013b. Evaluating Nutrient Availability in Semi-arid Soils with Resin Capsules and Conventional Soil Tests, II: Field Studies. *Communications in Soil Science and Plant Analysis* 44, 1764–1775. doi:10.1080/00103624.2013.769564
- Sun, H., Wu, Y., Zhou, J., Yu, D., Chen, Y., 2022. Microorganisms drive stabilization and accumulation of organic phosphorus: An incubation experiment. *Soil Biology and Biochemistry* 172, 108750. doi:10.1016/j.soilbio.2022.108750
- Yang, Z., Bing, H., Zhou, J., Wu, Y., Sun, H., Luo, J., Sun, S., Wang, J., 2015. Variation of mineral composition along the soil chronosequence at the Hailuoguo Glacier foreland of Gongga Mountain. *Acta Pedologica Sinica* 52, 507–516. (Published in Chinese)

➤ **Comment 21:**

L47 I think only one of the supporting sentences in this paragraph included citations....

Reply:

According to your other comments and guidance mentioned above, we have rewritten this paragraph. Please see lines 146 to 167 in Manuscript text (Revised Document with Track Changes).

➤ **Comment 22:**

L48 “interrelated”- I don’t follow how N and P availability is interrelated.

Reply:

We sincerely thank you for your feedback and apologize for any lack of clarity in our original text. We have removed the problematic sentence and rewritten the section for improved precision. In the original manuscript, when we referred to 'the interaction between nitrogen (N) and phosphorus (P) availability,' we meant that N availability (or P availability) can influence P availability (or N availability) through soil biological processes. This perspective is grounded in the well-established understanding, as noted in our revised Introduction (see line 146-250 in Manuscript text (Revised Document with Track Changes)):

“The interaction between nitrogen (N) and phosphorus (P) cycling processes has long been a focal point in the field of biogeochemistry. It is universally acknowledged that global terrestrial ecosystems are constrained by both N and P availability. Empirical evidence demonstrates that across various ecosystems, plant growth exhibits comparable increments following the addition of either N or P, which suggests the prevalence of N and P co-limitation. Moreover, given this fact that organisms require elements in stoichiometric proportion and are expected to take up and recycle nutrients in ratios that maintain balanced nutrition, some scholars considered that such co-limitation profoundly suggests an intrinsic coupling of N and P through soil biological processes, highlighting the interconnected dynamics of these nutrient cycles within terrestrial environments.”

➤ **Comment 23:**

L53-55 I don’t follow the logic of this statement. Occurring plants are known to use different startagies to acquire P (e.g. Raven et al. 2018).

Reply:

We have deleted this sentence and reorganized the logic in the Introduction section. Please review the revised Introduction section of the manuscript.

➤ **Comment 24:**

L55-57 I don’t follow the logic of this statement either. Poor word choice. What two analytical perspectives?.

Reply:

We have reorganized the logic for clarity. The two analytical perspectives are discussed in the immediately following text, namely:

1) “One analytical perspective focuses on the impact of N on indicators related to P processes, usually through N amendment experiments.”

2) “The other analytical perspective examines the influence of P on indicators related to N processes, typically through P addition experiments.”

➤ **Comment 25:**

L58 perspective on what? Poor word choice and poor starting sentence of a paragraph (which is often the paragraph’s topical sentence)..

Reply:

This perspective concerns the impact of N on indicators related to P processes and the impact

of P on indicators related to N processes. The two perspectives are expressed in the revised manuscript as follows:

1) “One analytical perspective focuses on the impact of N on indicators related to P processes, usually through N amendment experiments.”

2) “The other analytical perspective examines the influence of P on indicators related to N processes, typically through P addition experiments.”

➤ **Comment 26:**

L59 Again, nutrient addition seems like a rather crude tool to build understanding.

Reply:

This comment addresses a similar issue as the one raised in 'Comment 20' above. Please refer to our response to 'Comment 20' for a detailed explanation.

➤ **Comment 27:**

L60 Again, this feels superficial as there is no mention of the conditions that might cause one to predominate over the other. These are some relevant studies to look over (e.g. Ryan et al. 2012, Reinhart et al. 2024 and citations therein).

Reply:

We are extremely grateful for the valuable comments you have provided on our paper. You pointed out that our description of the soil phosphorus process is rather superficial, lacking a specific discussion on which of the mineralization of organic phosphorus and the dissolution of inorganic phosphorus is more dominant under different conditions. This is a very important perspective, and we fully agree with it.

In accordance with your suggestion, we have revised this part. The revised content explores which of the mineralization of organic phosphorus and the dissolution of inorganic phosphorus is more dominant under different conditions. For the detailed content, please see lines 169 to 175 in Manuscript text (Revised Document with Track Changes)

➤ **Comment 28:**

L62 Don't all enzymes require N?.

Reply:

Thank you for raising this question. Indeed, many enzymes contain nitrogen as a part of their structure, so your point is correct. However, in our statement, what I intended to emphasize was the role of nitrogen in phosphatases during the phosphorus cycle, especially in the process of organic phosphorus mineralization. Some studies have shown that the nitrogen content in phosphatases ranges from 8% to 32% (Treseder and Vitousek, 2001), suggesting that nitrogen is an unavoidable factor in terms of its importance for phosphatases when studying the process of organic phosphorus mineralization. Indeed, the influence of nitrogen has been taken into account in many studies that focus on phosphatases (Godin et al., 2015; Ratliff and Fisk, 2016; Chen et al., 2020; Mori et al., 2024; Tian et al., 2025).

References:

Chen, J., van Groenigen, K.J., Hungate, B.A., Terrer, C., van Groenigen, J.-W., Maestre, F.T., Ying, S.C., Luo, Y., Jorgensen, U., Sinsabaugh, R.L., Olesen, J.E., Elsgaard, L., 2020. Long-term nitrogen loading alleviates

- phosphorus limitation in terrestrial ecosystems. *Global Change Biology* 26, 5077–5086. doi:10.1111/gcb.15218
- Godin, A.M., Lidher, K.K., Whiteside, M.D., Jones, M.D., 2015. Control of soil phosphatase activities at millimeter scales in a mixed paper birch - Douglas-fir forest: The importance of carbon and nitrogen. *Soil Biology & Biochemistry* 80, 62–69. doi:10.1016/j.soilbio.2014.09.022
- Mori, T., Wang, S., Wang, C., Zhang, W., Mo, J., 2024. Effect of long-term nitrogen addition on the kinetics of phosphatases in a subtropical forest in southern China. *Applied Soil Ecology* 202. doi:10.1016/j.apsoil.2024.105589
- Ratliff, T.J., Fisk, M.C., 2016. Phosphatase activity is related to N availability but not P availability across hardwood forests in the northeastern United States. *Soil Biology & Biochemistry* 94, 61–69. doi:10.1016/j.soilbio.2015.11.009
- Tian, M., Jiang, N., Chen, Zhenhua, Zhang, Y., Jiang, D., Wu, C., Chen, Zhuoran, Qiu, W., Wang, J., 2025. Nitrogen and carbon addition mediate phosphorus cycling in grassland ecosystems: Insights from *phoD* gene abundance and community diversity. *Applied Soil Ecology* 206. doi:10.1016/j.apsoil.2025.105896
- Treseder, K.K., Vitousek, P.M., 2001. Effects of soil nutrient availability on investment in acquisition of N and P in Hawaiian rain forests. *Ecology* 82, 946–954. doi:10.1890/0012-9658(2001)082[0946:EOSNAO]2.0.CO;2

➤ **Comment 29:**

L64 “conclusions” on what?.

Reply:

Thank you for your comment. Upon reading this comment, we understand that although we elaborated on the "conclusions" in the following text, the way we expressed our ideas hindered readers from effectively accessing the information. Specifically, the conclusions referred to are those from previous studies discussed later in the text, which address whether nitrogen addition indeed enhances phosphatase activity.

In light of your feedback, we have revised the text to ensure that the subject of our conclusions is clear and that the discussion is more accessible to the reader. For the specific revised text, please see lines 179 to 185 in Manuscript text (Revised Document with Track Changes).

➤ **Comment 30:**

L63-7 wordy.

Reply:

Thank you for your feedback. According to your comments, we have rewritten this section to ensure it is concise and clear. For the specific revised text, please see lines 179 to 185 in Manuscript text (Revised Document with Track Changes).

➤ **Comment 31:**

L77 Meta-analysis tailor their study one questions with sufficient data to justify tests. They are effectively hamstrung by the prelevance of data (often collected by other people).

Reply:

We quite agree with the viewpoints in your comment. The reason why we cited these meta-analysis studies is that these studies have synthesized a large number of case studies on the relationship between nitrogen addition and phosphatase activity. Logically, the conclusions drawn

from these meta - studies through comprehensive analysis are more likely to be more universal and credible than those of individual case studies. Nevertheless, inconsistent conclusions have also been reached among these meta - analysis studies. This largely indicates that in - depth research on the relationship between nitrogen and phosphorus is still needed.

To highlight the necessity of integrating other processes apart from the N and P processes in the study of the N-P relationship, based on revealing the limitations of current research (including the case studies in meta-research), we added citations to demonstrate the importance of the C process in the study of the nitrogen-phosphorus relationship. Based on this, we improved this part of the text. For the specific revised text, please see lines 185 to 192 in Manuscript text (Revised Document with Track Changes).

➤ **Comment 32:**

L79 Good point but content seems like something that should have been mentioned earlier..

Reply:

. We are very glad to receive your appreciation. According to your comments, we have adjusted and revised the text. For the specific revised text, please see lines 189 to 192 in Manuscript text (Revised Document with Track Changes).

➤ **Comment 33:**

Comment- The authors seem to be using the inconsistent results of three global meta-analyses as motivation for the current study. This seems like a weak motivation for an empirical study. For example, the presence of meta-analyses alone indicate that there is already a lot of work on the topic. How is your work unique. If you are simply conducting a replication of prior studies then that work should be targeted to journals like PLoS One; Agrosystems, Geosciences & Environment; etc..

Reply:

Thank you for your comment. Due to the limitations of our writing skills, we failed to fully explain the unique motivation of our study in the introduction section of the original manuscript. Based on your comments and guidance, as well as those from other reviewers, we have revised and rewritten the introduction section.

In the revised content, we have placed particular emphasis on elaborating the necessity of integrating relevant processes other than the nitrogen and phosphorus processes into the nitrogen-phosphorus coupling framework. Additionally, we have highlighted that our study uniquely integrates the carbon process and the nitrification process into the nitrogen-phosphorus coupling framework - a dimension that has been overlooked in previous nutrient addition experiments but is crucial for explaining context-dependent responses.

For the specific revised text, please see lines 168 to 226 in Manuscript text (Revised Document with Track Changes).

➤ **Comment 34:**

L80 This seems unlikely. Many soil scientists have studied the effects of liming (to counteract soil acidification with fertilization) on available P and effects of fertilizer on soil pH (which can regulate available P). It is well known that N additions acidify soil and this can increase soil P depending on the soil's starting pH.

Reply:

Thank you for your feedback. We are sorry for the hasty conclusion written in that manner. According to your comments, we have deleted that text.

➤ Comment 35:

L88 “promoting effect of P is always significant.” – vague & poor word choice. “This” refers to what?.

Reply:

The term 'this' refers to the previously mentioned statement that 'exogenous P inputs often enhance diazotrophic activity and N fixation rate.' According to your comments, we have revised this text. For the specific revised text, please see lines 206 to 209 in Manuscript text (Revised Document with Track Changes).

➤ Comment 36:

L98 good point.

Reply:

We are genuinely gratified by your favorable assessment, which provides compelling affirmation of the validity embedded within our conceptual framework. It is noteworthy that your commentary has not only informed but also catalyzed substantive refinements across multiple manuscript sections.

➤ Comment 37:

L109 You should clearly (re-)state the research gap that is being addressed.

Reply:

In accordance with your comments, we have rewritten this text and clearly stated the research gap, that is, integrating related processes such as the carbon process and the nitrification process into a nitrogen-phosphorus coupling framework is crucial for interpreting environment-dependent responses and resolving contradictions in the conclusions, and this concept has been overlooked in previous nutrient addition experiments.

For the specific revised text, please see lines 189-192, 202-204, and 224-226 in Manuscript text (Revised Document with Track Changes).

➤ Comment 38:

L113 citation #2- I think you mean optimal allocation theory (Bloom et al. 1985)..

Reply:

Yes, this perspective incorporates the optimal allocation theory proposed by Bloom et al. (1985). Specifically, this view was summarized by Marklein and Houlton(2012) from the papers of Bloom et al. (1985) and Chapin et al. (2002). This perspective holds that plants (as well as microorganisms) are expected to allocate their resource reserves to strategies that can enhance the acquisition of the most limiting resources – thus moving towards a state where all resources simultaneously limit productivity and growth. Following the logic of this perspective, it can be inferred that ecosystems will degenerate and disappear because an increasing variety of nutrients become limiting factors. However, this perspective does not seem to be able to explain the phenomenon of primary succession in pioneer ecosystems (such as our study area): Pioneer

ecosystems are so nutrient-poor that plants cannot grow, but they can undergo succession to develop into forest ecosystems rich in available nutrients.

References:

- Bloom, A.J., Chapin, F.S., Mooney, H.A., 1985. Resource Limitation in Plants—An Economic Analogy. *Annual Review of Ecology and Systematics* 16, 363–392.
- Chapin, F.I., Mooney, H., Chapin, M., Matson, P., 2002. *Principles of terrestrial ecosystem ecology*. Springer, New York, NY, USA.
- Marklein, A.R., Houlton, B.Z., 2012. Nitrogen inputs accelerate phosphorus cycling rates across a wide variety of terrestrial ecosystems. *New Phytologist* 193, 696–704. doi:10.1111/j.1469-8137.2011.03967.x

➤ **Comment 39:**

L116 sentence has similar problem to L109..

Reply:

According to your comments, we have rewritten the text. For the specific revised text, please see lines 244 to 250 in Manuscript text (Revised Document with Track Changes).

➤ **Comment 40:**

L146 poorly worded sentence. Try- X positively affected Y and negatively affected Z..

Reply:

It is our great honor to receive your guidance. According to your instructions, we have revised the sentence to clarify the contrasting effects of N addition as follows (see line 277-280 in Manuscript text (Revised Document with Track Changes)):

N addition treatments negatively affected pH (causing a significant decrease), but positively affected AP concentrations, low molecular weight organic acids levels, and the abundance of acid secretion genes (*gcd*), all of which showed significant increases.

➤ **Comment 41:**

L151 leaching- What about sorption? If the sorption capacity of the soil is high that there may be no (or a very temporary) increase in AP (Jones et al. 2013a, Jones et al. 2013b)..

Reply:

The adsorption capacity of this early-stage soil should be very weak. The study area is the bedrock that emerged right after the glacier retreated. The soil here is extremely young and has not undergone long-term weathering. It contains almost no secondary minerals (such as clay minerals) with strong adsorption capabilities (Yang et al., 2015). Strictly speaking, this soil is more like rock debris (Sun et al., 2022), as shown in Figure S1 in Supplementary materials. Moreover, our results indicate that, compared with the control group, the phosphorus addition treatment significantly increased the concentration of available phosphorus in the soil (Figure 2a of our manuscript). Therefore, the ability of this early-stage soil to adsorb available phosphorus is weak.

➤ **Comment 42:**

L173 Okay but pH was affected by N addition right (Fig. 2)?.

Reply:

Yes, our results indicate that the addition of nitrogen indeed affects the pH value, especially

considering that the soil in our study area is alkaline..

➤ **Comment 43:**

L340 It is somewhat curious that with 90 replicate pots (+N, +P, and control) that you didn't include an N&P addition and reduce your replication number per treatment.

Reply:

Actually, we did conduct experiments with both N and P additions (though the data were not presented in the original manuscript). Since both N and P were added simultaneously in these treatments, we found that it was impossible to distinguish the individual effects of N or P, and we were unable to convincingly identify the pathways through which N affects P (or vice versa). While the results from this treatment indicated positive effects such as increased P and N availability, enhanced abundance of nitrogen-fixing genes, elevated levels of phosphorus-solubilizing related genes, higher concentrations of organic acids, etc., illustrating the co-limitation by N and P, our study's focus was not on proving their co-limitation but rather on understanding the mechanisms of interaction between N and P. To streamline the presentation of our findings and reduce redundancy in the manuscript, the data from the simultaneous addition of N and P were omitted from the original manuscript. However, to address your queries and provide robust support for our findings, the data and analysis results from the N and P dual addition experiment have been included in the supplementary materials of the revised manuscript (see Table S7 source data for the treatment with combined N and P addition.xlsx and Figure S2 in Supplementary materials).

➤ **Comment 44:**

L348 “to prevent lateral leaching of N or P”- I sort of understand the decision to bag and reinsert cores but this also adds layers of disturbance (coring) and abstraction (bagged soil that prevents leaching) which makes the results a bit unrealistic compared to the more common practice of applying fertilizer to entire plots.

Reply:

Thank you for raising this important methodological concern. We deeply appreciate your thoughtful feedback and would like to clarify our experimental design with additional technical details to address your inquiry comprehensively.

First of all, we would like to clarify that the container we used to hold the undisturbed soil cores is not a bag with one open end and one sealed end, but a cylindrical plastic film container with both ends open. This cylindrical container not only minimizes the interference with the natural leaching phenomenon but also prevents the contamination of other surrounding experimental soil cores by lateral seepage. In fact, this experimental design is essentially a miniaturized version of a partitioned plot experiment. Secondly, if N and P were applied to the entire plot, it would cause pollution to this nature reserve (supplementary note: our study area is within a national protected glacier forest park). Compared with the plot fertilization experiment, our experimental design can limit the pollution to a very small area. Moreover, after the experiment, by digging out the entire cylindrical plastic film container, it is conducive to the removal of these treated soil cores, thus reducing the pollution to the original ecosystem. We fully understand and appreciate the importance of balancing rigorous scientific inquiry with environmental stewardship. Our experimental design reflects a conscientious effort to advance

research objectives while safeguarding the integrity of this ecologically sensitive area. We hope this explanation clarifies our methodology and addresses your concerns adequately.

To summarize, the reasons for designing our experiment in this manner are as follows:

1). Experimental Container Design

The soil containment system employs a dual-open-ended cylindrical polyethylene sleeve rather than a single-sealed bag structure. This design specifically preserves natural vertical hydrological connectivity while preventing lateral solute transport through three key mechanisms:

- a. The high-density polyethylene film creates a hydrologically isolated microenvironment
- b. Vertical alignment maintains intact preferential flow paths within soil macropores
- c. Absence of lateral permeability eliminates cross-contamination between adjacent cores

Notably, this configuration achieves functional equivalence to partitioned field plots in terms of hydrological isolation, while offering superior control over microscale biogeochemical processes.

2). Environmental Protection Considerations

Given that our study area is situated within the Gongga Mountain National Glacier Forest Park (Class I Protected Area), we implemented a rigorous hierarchical pollution mitigation protocol:

- a. Spatial confinement: Fertilization was restricted to the cylindrical microcosms (<0.1 m² footprint per unit), reducing impacted area compared to traditional plot-scale applications
- b. Temporal control: Pulse fertilization events were synchronized with natural precipitation cycles to minimize solute retention
- c. Post-experimental remediation: A complete removal protocol was executed, involving: Full excavation of sleeve-contained soil volumes and Site restoration.

➤ **Comment 45:**

L360-1 So each sample (270) were separately assayed for physicochemical properties and genes? Were any samples lost or destroyed [I think that was motivation for the large number of plots]?

Reply:

Yes, a total of 270 samples were separately assayed for physicochemical properties and genes (we have annotated the sample sizes (i.e., n values) used for calculating the respective metrics in both the tables and figures). From the description of the methods in the manuscript, you can see that the number of sample plots set up is more than 270. This was done for the following purposes:

1). To ensure that the experimental data accurately reflect the true conditions of the soil in the study area and to guarantee the robustness of the data analysis results. This is because the diverse microtopography in the study area leads to significant soil heterogeneity.

2). Since this area is a high mountain area (the highest peak in the area is 7,556 meters above sea level) and an active glacier retreat area, and small-scale geological disasters (such as small debris flows, small landslides, etc.) occur frequently. Considering the situation that the experimental sample plots may be damaged by these disasters during the experiment, in order to ensure that 90 samples can be obtained from each treatment at the end of the experiment, the number of sample plots was set to be more than 270. Such a sample size is conducive to constructing a robust structural equation model.

➤ **Comment 46:**

L390 You probably need to spell out the samples size for the various measurements (ST, BD) that were not based on the 270 soil cores..

Reply:

Based on your comments, we have annotated the sample sizes (i.e., n values) used for calculating the respective metrics in both the tables and figures.

➤ **Comment 47:**

L589 Fig. 2 header should mention the number of plot (or core) replications per treatment (e.g. n= 90 plot replications per treatment).

Reply:

Yes, according to your comment, we have annotated the sample sizes (i.e., n values) used for calculating the respective metrics in both the tables and figures.

➤ **Comment 48:**

Fig. S1 This figure is somewhat helpful but I was expecting to see something with more detail on where the plots were positioned or plot layout diagram or photos. The two pictures of the bags fail to show plots. Ex/ How big is the area where the 90 plots (per treatment) were laid out (also not shown on the map)?.

Reply:

We have redrawn "Fig. S1". Specifically, we have added a real-scene photograph of the study area and a schematic diagram of the layout of the plots to show details such as the location and size of the plots.(see Figure S1 in Supplementary materials)

➤ **Comment 49:**

Excel data file doesn't appear to include information on bulk density, soil temperature, and moisture.

Reply:

Based on your comments, we have added information regarding bulk density, soil temperature, and soil moisture in the supplementary materials (see Table S9 soil property data.xlsx).

Reply to Reviewer #4 again:

➤ **Comment 1:**

R4: This is the second time that I have reviewed the paper. I am satisfied with the changes made by the authors. Nice experiment and interesting results that help to improve our understanding of nutrient cycling and succession is undeveloped (& probably little developed soils)!

There are a few editorial suggestions to help improve your paper.

L236 “CK plots” why not just write “control plots”?

Reply:

Thank you for your valuable suggestion. As requested, "CK plots" have been revised to "control plots" in the revised manuscript.

➤ **Comment 2:**

L238 “stimulating” to “stimulated”

Reply:

As requested, “stimulating” have been revised to “stimulated” in the revised manuscript.

➤ **Comment 3:**

L250 nice job!

Reply:

Thank you sincerely for your recognition of our work.

➤ **Comment 4:**

L258 “to produce enzymes” to “production of enzymes”

Reply:

As requested, “to produce enzymes” have been revised to “production of enzymes” in the revised manuscript.

➤ **Comment 5:**

L263 “availability;” to “availability.”

Reply:

As requested, “availability;” have been revised to “availability.” in the revised manuscript.

➤ **Comment 6:**

L279 “with in turn” to “which in turn”

Reply:

As requested, “with in turn” have been revised to “which in turn” in the revised manuscript.

➤ **Comment 7:**

L298 Good point!

Reply:

Thank you sincerely for your recognition of our work.

➤ **Comment 8:**

L325 Good point! You might want to add a citation. This is one that I can easily think of (Craine, Morrow & Stock 2008) but there might be better.

References

Craine, J.M., Morrow, C. & Stock, W.D. (2008) Nutrient concentration ratios and co-limitation in South African grasslands. *New Phytologist*, 179, 829-836.

Reply:

Thank you for highlighting this important point and for suggesting the relevant reference (Craine, Morrow & Stock 2008). We agree that contextualizing our findings within existing literature strengthens the manuscript. As requested: We have added the suggested citation (Craine et al., 2008) in the revised manuscript.

➤ **Comment 9:**

L372, 378- Did you mean to indent?

Reply:

Thank you for raising this question regarding the indentation at Lines 372 and 378. The indentation was intentionally applied to visually distinguish the experimental methodology section from surrounding text, aiming to enhance readability and clarity for readers. However, we fully respect the journal's formatting standards and are happy to adjust the indentation if it conflicts with editorial guidelines. At present, we have removed the indentation at Lines 372 and 378 as requested.

➤ **Comment 10:**

L377 extra spaces

Reply:

We have removed the extra spaces.

➤ **Comment 11:**

L738 Nice conceptual figure.

Reply:

Thank you sincerely for your recognition of our work.